# Dynamic coding and sequential integration of multiple reward attributes by primate amygdala neurons

Fabian Grabenhorst [1] ✉ & Raymundo Báez-Mendoza [2]

The value of visual stimuli guides learning, decision-making, and motivation. Although stimulus values often depend on multiple attributes, how neurons extract and integrate distinct value components from separate cues remains unclear. Here we recorded the activity of amygdala neurons while two male monkeys viewed sequential cues indicating the probability and magnitude of expected rewards. Amygdala neurons frequently signaled reward probability in an abstract, stimulus-independent code that generalized across cue formats. While some probability-coding neurons were insensitive to magnitude information, signaling 'pure' probability rather than value, many neurons showed biphasic responses that signaled probability and magnitude in a dynamic (temporally-patterned) and flexible (reversible) value code. Specific amygdala neurons integrated these reward attributes into risk signals that quantified the variance of expected rewards, distinct from value. Population codes were accurate, mutually transferable between value components, and expressed differently across amygdala nuclei. Our findings identify amygdala neurons as a substrate for the sequential integration of multiple reward attributes into value and risk.

The value associated with visual stimuli is a critical parameter that guides reinforcement learning and economic decision-making. The amygdala, a collection of nuclei in the medial temporal lobe, has long been implicated in linking visual stimuli with value and related emotional and behavioral responses[1–8]. Recent data indicate that the primate amygdala also contributes directly to decision computations that translate these values into choices[9–13]. Although value often depends on multiple attributes, including the probability and magnitude of expected rewards, how individual neurons extract such separate value components from distinct cues is poorly understood. Moreover, it remains unclear how neurons integrate different value components over time into statistical estimates that guide economic choices[14,15], including the risk (i.e., variance) of expected outcomes.

Here we investigated how primate amygdala neurons extract information about two principal reward parameters—probability and magnitude—from distinct, sequentially presented visual cues, and integrate this information into economic decision variables. Previous studies showed that primate amygdala neurons signal changing values in individual and social learning contexts[1,2,9,12,16–18], and code subjective values based on multiple attributes during immediate and longer-term economic choices[10,11,13,19]. Yet, important questions regarding the sensitivity and selectivity of single amygdala neurons to different value components remain unanswered.

First, are individual amygdala neurons sensitive to both reward probability and magnitude, consistent with value coding, or are these parameters encoded separately by different groups of neurons? Probability and magnitude reflect two fundamentally different aspects of a reward: the abstract likelihood of receiving it and its concrete, physical size. Extracting these attributes from visual cues might therefore require specialized mechanisms in distinct neurons. We reasoned that some amygdala neurons could operate analogous to sensory receptors by responding selectively to information about

[1]Department of Experimental Psychology, University of Oxford, Oxford, UK. [2]Department of Neurobiology, German Primate Center, Göttingen, Germany.
✉e-mail: fabian.grabenhorst@psy.ox.ac.uk

specific reward attributes (probability or magnitude), comparable to the transduction of different wavelengths by photoreceptors or different taste qualities by taste receptors[20,21]. By contrast, processing probability and magnitude as values would require neurons to respond similarly to cues indicating these different attributes. Specialized neurons processing distinct reward attributes have been described in select brain systems[22–27] but only a few studies tested multiple attributes with separate cues on single trials[28].

Second, do amygdala neurons also signal estimates of reward statistics including risk (measured as the variance of the current reward distribution), and are these abstract variables—which require integration of different information sources—encoded by the same or by different neurons that encode probability and magnitude? We adopt the approach common in economic and finance theory that measures risk as variance[29–32], which takes into account the variance of the statistical distribution of possible outcomes (both positive and negative), which differs from the notion of risk as 'probability of loss'. Risk-coding neurons might respond to different, separately cued reward components and update their risk signal once new information is presented. Although neuronal risk signals have been reported in various brain systems[14,15,22,24,27,33–37], how they derive from separately cued attributes is unclear.

Finally, how do single-neuron codes for reward attributes translate into population codes that enable flexible information readout? Neurons in the amygdala and other brain systems often show mixed selectivity[10,12,38–41] by encoding more than one variable. Yet, these neurons have rarely been investigated with experimental designs that aim to temporally separate the individual signal components that constitute the mixed code. Moreover, risk constitutes a particularly interesting example of mixed coding as it reflects the mathematical integration (variance calculation) of probabilities and magnitudes that specify statistical outcome distributions[15].

Here we investigated the role of primate amygdala neurons in the dynamic coding and sequential integration of distinct reward attributes to value and risk. We recorded the activity of single amygdala neurons while monkeys viewed a sequence of visual cues that separately signaled the probability and magnitude of predicted reward and whose integration over time specified the expected value (mean) and risk (variance) of the current reward distribution. We reasoned that single-neuron data recorded in such testing conditions might allow us to identify basic neural-processing elements underlying valuation and decision mechanisms. Such data could help refine current theories of value coding, e.g., by informing debates on the importance of neurons with mixed vs. highly specialized coding properties[41,42]. The findings might also advance our understanding of the amygdala's role in conditions with mental-health impairments, including depression and anxiety, which involve dysfunctional expectations of uncertain rewards and threats in which the amygdala is implicated[8,43–47].

## Results

### Experimental design
We addressed the above questions in a Pavlovian task by presenting visual cues for the basic reward attributes probability and magnitude sequentially on the same trial (Fig. 1a–c). This simple design allowed us to test the sensitivity and specificity of individual amygdala neurons to these cued parameters and their integration into more abstract properties of reward distributions: expected value (calculated as the sum of all probability-weighted values of a given reward distribution) and risk. Following previous studies, we adopted the frequently considered measure of economic risk as variance, calculated as the sum of the probability-weighted differences between outcomes and expected value[29–32]: risk defined in this way increased from $P = 0$ to $P = 0.5$ and decreased from $P = 0.5$ to $P = 1$ for a given reward magnitude, following an 'inverted-U' function, and increased across reward magnitudes for a given probability level (Fig. 1b). Notably, the sequential presentation of

probability and magnitude cues allowed us to test neuronal encoding of risk purely dependent on probability (at probability cue) and neuronal encoding of risk dependent on both probability and magnitude (at the magnitude cue that followed presentation of the probability cue).

On each trial, probability and magnitude cues were drawn pseudorandomly from a set of 15 combinations (three probability levels, five magnitude levels, Fig. 1c, d), thus requiring flexible, dynamic integration. To test the independence of visual stimulus properties, we used two distinct stimulus formats to cue probability: colored fractals and monochrome sector stimuli (Fig. 1d).

We chose a Pavlovian, non-choice task as this allows for a clearer examination of neuronal signals, i.e., a measured neuronal response would only depend on the value of the presented option rather than reflect additional decision processes involving value comparison between multiple choice options. Further, we presented information about probability and magnitude sequentially in non-overlapping cue periods to separately examine how an individual neuron responded to information pertaining to these two variables. This sequential design with non-overlapping explicit cues for reward probability and magnitude also allowed us to examine the temporal dynamics of how the abstract variables expected value and risk are encoded. Crucially, these two variables were not explicitly cued but had to be computed internally by integrating information from the sequentially presented probability and magnitude cues.

### Behavior
We assessed the influences of reward probability, magnitude, expected value and risk on behavior in a choice task, performed during the period of neuronal recordings but mostly on separate testing days (Fig. 1e). Initial tests during training confirmed that both monkeys consistently chose higher reward magnitude options in the absence of probability differences and higher probability options (cued by either sector or fractal stimuli) in the absence of magnitude differences[48].

Choices in both animals reflected the separate influences of probability and magnitude (Fig. 1f, Table 1). When testing for the explicit integration of reward probability and magnitude into expected value and risk, both animals showed significant effects of expected value on choice (Table 1). However, whereas animal B showed a consistent positive influence of risk on choice ('risk-seeking', i.e., preferring options with higher risk), animal A showed a significant positive influence of risk on choice only for low-probability trials (Fig. 1g). This effect likely reflects previously observed non-linear probability estimation[49] or risk-attitude specifically for low-probability outcomes[50]. Thus, risk had a positive weight on the animals' choices consistent with mild risk-seeking seen in previous studies[50]. Subjective values modeled as the integration of these variables, derived from the logistic regression, showed systematic relationships with choice probability (Fig. 1h). The animals' choices also conformed with first-order stochastic dominance[51]: when choosing between a safe reward and a gamble of equal magnitude (i.e., of lower expected value), the animals preferred the safe reward (Fig. 1i). (Note that the animals' risk-seeking tendencies defined in the regression in Fig. 1g suggest preferences for risk when expected value is accounted for (i.e., is included as covariate); by contrast, following first-order stochastic dominance in Fig. 1i indicates preference for higher expected value.) Thus, the animals understood the gambles' values and used this information for making meaningful choices.

We confirmed that these reward variables also influenced behavior during the Pavlovian task in which we recorded neuronal data. Error rates (trial abortion rates due to fixation breaks) suggested that the monkeys correctly processed the reward information provided by the visual cues and that this information affected the monkeys' motivation. Mixed-effects regression showed that the likelihood of aborting a trial during the cue periods in both animals reflected the

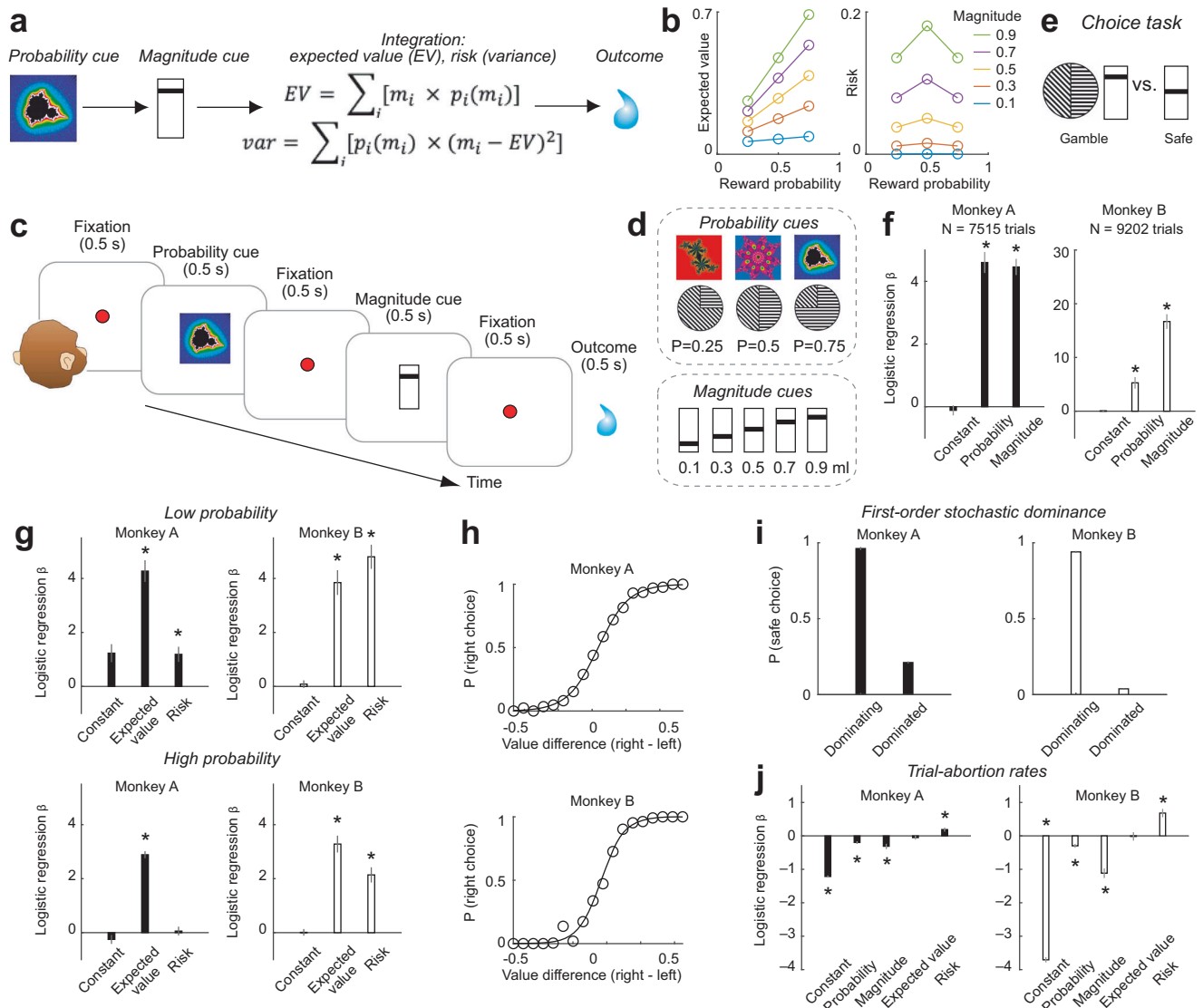

**Fig. 1 | Sequential reward-prediction task and reward-guided behavior.**
**a** Design. Sequential cues for reward probability and reward magnitude test neuronal sensitivity to these reward attributes and integration to expected value and economic risk. **b** Schematic. Illustration of how expected value and risk depend on both reward probability and magnitude. **c** Trial timeline for task used in neurophysiological recordings. **d** Stimuli used to cue probability and magnitude. **e** Schematic of choice task. **f** Logistic regression of choices on probability and magnitude across sessions. 'Constant' refers to the intercept in the regression, modeling bias for right vs. left choices. Bars in this panel and panels **g**, **j** show regression coefficients (+/− s.e.m.). Statistical significance in this panel and panels **g**, **j** assessed with two-sided t-test (no adjustment for multiple comparisons). *$P < 0.005$. **g** Logistic regression of choices on expected value and risk for low (top,

probability <=0.25; 1386/1714 trials for monkey A/B) and high (bottom, probability >0.25; 6129/6874 trials for monkey A/B) probability trials. **h** Psychometric functions illustrating the systematic relationship between subjective values (modeled from logistic regression) and choice probabilities. **i** First-order stochastic dominance. Choice probability for stochastically dominating (safe option has higher expected value than a gamble; 344/1436 trials for monkey A/B) and dominated (the safe option has lower expected value than a gamble; 1436/2519 trials for monkey A/B) options. Monkeys prefer options with higher magnitude given the dominated probability. **j** Logistic regression of trial abortions on reward and economic variables. 'Constant' refers to the intercept in the regression; negative beta for constant indicates tendency to not abort trials. 21362/34667 trials for monkey A/B.

influences of probability and magnitude, with lower abortion likelihood for higher probability and magnitude trials (Fig. 1j, Table 2). Interestingly, risk had a distinct influence on trial-abortion rate, with both animals being more likely to abort high-risk trials (Fig. 1j, Table 2). Anticipatory lick rates also reflected the influence of a combination of these variables, including risk, in a manner that differed between animals (Table 3, Fig. S1). We confirmed in a supplementary regression that both animals showed higher lick rates for probability and magnitude (Table S1), consistent with previous studies[52–54].

Thus, the animals used the cued reward attributes probability and magnitude as well as integrated variables value and risk in a subjective manner for making choices and calibrating motivational task

engagement (trial abortion rates) and reward expectation (lick rates). We next tested how amygdala neurons responded to and integrated these reward attributes.

**Amygdala neurons signal reward probability in an abstract code**
We recorded the activity of 483 amygdala neurons (176/307 neurons in animal A/B) across lateral (LA), basolateral (BL), basomedial (BM) and centromedial (Ce) nuclei (Fig. S2). We sampled activity from about 1000 amygdala neurons and typically recorded and saved the activity of those neurons that appeared to respond to any task event during online inspection of several trials. We aimed to identify task-responsive neurons but did not preselect based on particular response

**Table 1 | Mixed-effects logistic regression of choices on task variables**

| Variable | Estimate | Standard error | t-statistic | Degrees of Freedom | P-value |
|---|---|---|---|---|---|
| **Animal A** | | | | | |
| Intercept | −0.115 | 0.162 | −0.710 | 7512 | 0.477 |
| Prob left-right | 4.602 | 0.335 | 13.701 | 7512 | 3.1e−42 |
| Mag left-right | 4.459 | 0.258 | 17.268 | 7512 | 1.15e−65 |
| **Animal B** | | | | | |
| Intercept | 0.063 | 0.105 | 0.597 | 9199 | 0.549 |
| Prob left-right | 5.277 | 1.071 | 4.926 | 9199 | 8.5e−7 |
| Mag left-right | 16.689 | 1.369 | 12.189 | 9199 | 6.4e−34 |
| **Animal A** | | | | | |
| Intercept | −0.117 | 0.168 | −0.695 | 7512 | 0.486 |
| EV left-right | 2.822 | 0.132 | 21.249 | 7512 | 2.3e−97 |
| Risk left-right | 0.140 | 0.152 | 0.929 | 7512 | 0.357 |
| **Animal B** | | | | | |
| Intercept | 0.037 | 0.100 | 0.376 | 9199 | 0.706 |
| EV left-right | 13.190 | 1.292 | 10.208 | 9199 | 2.4e−24 |
| Risk left-right | 24.876 | 3.192 | 7.793 | 9199 | 7.2e−15 |
| **Animal A, low probabilities** | | | | | |
| Intercept | 1.232 | 0.333 | 3.699 | 1383 | 0.00002 |
| EV left-right | 4.272 | 0.403 | 10.590 | 1383 | 2.9e−25 |
| Risk left-right | 1.188 | 0.281 | 4.225 | 1383 | 2.5e−5 |
| **Animal B, low probabilities** | | | | | |
| Intercept | 0.084 | 0.141 | 0.596 | 1711 | 0.550 |
| EV left-right | 3.846 | 0.458 | 8.379 | 1711 | 1.0e−16 |
| Risk left-right | 4.801 | 0.447 | 10.736 | 1711 | 4.5e−26 |
| **Animal A, high probabilities** | | | | | |
| Intercept | −0.251 | 0.166 | −1.509 | 6126 | 0.131 |
| EV left-right | 2.886 | 0.130 | 22.063 | 6126 | 7.2e−104 |
| Risk left-right | 0.057 | 0.164 | 0.350 | 6126 | 0.726 |
| **Animal B, high probabilities** | | | | | |
| Intercept | 0.011 | 0.122 | 0.093 | 6871 | 0.925 |
| EV left-right | 3.281 | 0.312 | 10.506 | 6871 | 1.2e−25 |
| Risk left-right | 2.138 | 0.282 | 7.581 | 6871 | 3.8e−14 |

Statistical significance assessed with two-sided t-test (no adjustment for multiple comparisons).

characteristics. This procedure resulted in a database of 483 neurons that were recorded and analyzed statistically. In the following, we characterize amygdala neurons responding to probability, magnitude, risk, and expected value, before examining population codes.

Many amygdala neurons responded to the visual cues by signaling the cue-associated reward probability irrespective of cue format (fractals or sectors). Figure 2a illustrates an example neuron that signaled the probability of expected reward across the two cue formats.

The neuron's activity in response to the visual cues increased as a function of reward probability, both for fractal cues (left) and sector cues (right). Linear regression confirmed a significant positive relationship between neuronal activity and probability across cue formats (Fig. 2b). We identified probability-coding neurons using a multiple regression with separate probability regressors for fractal and sector trials, performed for cue-evoked activity during the first (probability) cue period. This previously established approach[12,55,56] classifies neurons based on their angle in the space of probability-regression coefficients for each cue format (Fig. 2c, d, Eq. 1, Fig. S3). The neuron in Fig. 2a was classified as coding probability on both trial types ($P = 1.1 \times 10^{-6}$; blue diamond symbol in Fig. 2c). By contrast, Fig. 2e, f show two different neurons that coded probability only in response to one specific cue format (fractals: $P = 4.3 \times 10^{-18}$; sectors: $P = 3.0 \times 10^{-19}$; represented by green and red symbols in Fig. 2c).

In total, 148 of 483 amygdala neurons (30%) encoded probability; 84 (57%) of these encoded probability across both cue formats, compared to 64 (43%) encoding probability only for a specific cue format (Fig. 2d). Of 84 neurons with significant probability coding for both cue formats, 14 neurons coded probability with different signs across cue formats. Thus, 70 of 483 neurons (15%) coded probability irrespective of cue format with matching signs. Sliding-window regression (Eq. 2) confirmed probability-coding in 124/483 neurons (26%). Graded probability signals across cue formats were prominent in the population activity of unselected amygdala neurons (Fig. 2g). Probability-coding neurons were found across different amygdala nuclei with similar prevalence (Fig. 2h; chi-square tests, $P > 0.05$).

Thus, individual amygdala neurons encoded cue-associated reward probability mostly in an abstract, cue-unspecific format, with additional neurons coding probability selectively for specific cue formats.

### Amygdala neurons sequentially code reward probability and magnitude

We found that amygdala neurons often responded to both the sequential probability and magnitude cues with graded signals that reflected the level of the currently cued reward attribute. These sequential, graded responses in the same neuron are conceptually important because they provide a single-neuron basis for signaling different types of value information, and potentially for integration over time into expected value or risk.

Figure 3a shows a neuron with responses to the probability cue that reflected the cued reward probability and subsequent responses on the same trial to the magnitude cue that reflected the cued reward magnitude. In both cases, activity increased with decreasing values of the cued reward parameter (Fig. 3b). The neuron's probability signal occurred irrespective of cue format (Fig. 3c, d). Sliding-window regression with both probability and magnitude regressors confirmed temporally specific and non-overlapping coding of these two reward parameters (Fig. 3e). Figure 3f shows a different amygdala neuron that responded only to the reward-magnitude cue but not to the preceding probability cue, indicating selective coding of reward magnitude and insensitivity to probability information.

We used the same multiple-regression approach as above for classifying neurons as coding probability only, magnitude only, or integrating both variables, performed on cue-evoked activities during the first and second cue periods (Fig. 3g, Eq. 3, Fig. S3). Of 483 neurons, 121 encoded reward magnitude (25%). Of these, 52 (43%) encoded 'pure' magnitude without encoding probability, whereas 69 (57%) encoded magnitude after encoding probability; only 19 probability-coding neurons (22%) encoded probability without subsequently encoding magnitude (Fig. 3h). Sliding-window regression (Eq. 2) across the whole trial identified 129 neurons coding magnitude but not probability, 50 neurons coding probability but not magnitude and 74 neurons coding both probability and magnitude. The magnitude

## Table 2 | Mixed-effects logistic regression of trial-abortion rates on task variables

| Variable | Estimate | Standard error | t-statistic | Degrees of Freedom | P-value |
|---|---|---|---|---|---|
| **Animal A** | | | | | |
| Intercept | −1.215 | 0.032 | −37.554 | 21,357 | 6.1e−299 |
| Probability | −0.197 | 0.037 | −5.320 | 21,357 | 1.0e−7 |
| Magnitude | −0.310 | 0.074 | −4.156 | 21,357 | 3.2e−5 |
| Expected value | −0.051 | 0.059 | −0.854 | 21,357 | 0.392 |
| Risk | 0.189 | 0.061 | 3.092 | 21,357 | 0.001 |
| **Animal B** | | | | | |
| Intercept | −3.707 | 0.076 | −48.696 | 34,662 | 0 |
| Probability | −0.297 | 0.056 | −5.294 | 34,662 | 1.2e−7 |
| Magnitude | −1.115 | 0.143 | −7.776 | 34,662 | 67.6e−15 |
| Expected value | −0.016 | 0.125 | −0.131 | 34,662 | 0.895 |
| Risk | 0.682 | 0.124 | 5.464 | 34,662 | 4.6e−8 |

Statistical significance assessed with two-sided t-test (no adjustment for multiple comparisons).

## Table 3 | Mixed-effects regression of licking behavior on task variables

| Variable | Estimate | Standard error | t-statistic | Degrees of Freedom | P-value |
|---|---|---|---|---|---|
| **Animal A** | | | | | |
| Intercept | −0.032 | 0.041 | −0.779 | 12,242 | 0.435 |
| Probability | −0.137 | 0.076 | −1.798 | 12,242 | 0.072 |
| Magnitude | −0.573 | 0.121 | −4.704 | 12,242 | 2.5e−6 |
| Expected value | 0.791 | 0.153 | −5.146 | 12,242 | 2.6e−7 |
| Risk | 3.038 | 0.513 | 5.923 | 12,242 | 3.2e−9 |
| **Animal B** | | | | | |
| Intercept | −0.259 | 0.055 | −4.689 | 6471 | 2.7e−6 |
| Probability | 0.225 | 0.102 | 2.195 | 6471 | 0.028 |
| Magnitude | 0.609 | 0.165 | 3.674 | 6471 | 0.0002 |
| Expected value | −0.123 | 0.208 | −0.591 | 6471 | 0.554 |
| Risk | −2.207 | 0.713 | −3.091 | 6471 | 0.001 |

Statistical significance assessed with two-sided t-test (no adjustment for multiple comparisons).

coding of individual neurons translated into a prominent, graded reward-magnitude signal in the population activity of all recorded amygdala neurons (Fig. 3i). Neurons coding pure magnitude or probability-to-magnitude transitions were prevalent across amygdala neurons, without detectable regional differences (Fig. 3j, chi-square tests, $P > 0.05$). The dynamic-coding pattern illustrated by the neuron in Fig. 3e with sequential probability and magnitude coding was prominent at the population level across all recorded neurons and was flexible, as it reversed in a subset of neurons tested with alternative magnitude-to-probability presentation sequences (Fig. 3k). The population signals for probability and magnitude in Fig. 3k peaked at 161 and 181 ms, respectively, following the onset of the relevant cue (probability-coding latency in specific nuclei LA/BL/BM/Ce: 121/161/121/201 ms; magnitude-coding latency in specific nuclei LA/BL/BM/Ce: 161/201/221/161 ms). (Note that signals from different nuclei were not simultaneously recorded).

Thus, individual amygdala neurons sequentially and flexibly encoded probability and magnitude in response to sequential cues,

providing a single-neuron basis for integrating multiple reward attributes to value and risk.

## Amygdala neurons signal risk as the variance of expected rewards

A substantial number of amygdala neurons (99/483, 20%; Eq. 4) carried signals that reflected the risk of expected rewards, defined by the variance of the current-trial statistical reward distribution derived from the integration of probability and magnitude information (cf. Figure 1a, b; Fig. S4). The neuron in Fig. 4a encoded risk (i.e., variance as calculated in Fig. 1a) in response to the reward-magnitude cue with increasing activity reflecting increased risk (Fig. 4a, b). The neuron's response was better explained by risk than by reward magnitude, probability or expected value, which were multiple-regression covariates (risk: $P = 0.014$, magnitude: $P = 0.90$, probability: $P = 0.11$, expected value: $P = 0.31$; Eq. 4). Interestingly, the risk signal was preceded by a reward-probability signal ($P = 0.013$, Fig. 4a–c); thus, the neuron integrated sequentially viewed probability and magnitude into risk ('probability-to-risk transition').

Risk signals were present in the average population activity of all recorded amygdala neurons (Fig. 4d). There was no clear relation in the population signal between probability coding in the first cue period and subsequent risk coding when referencing these signals based on each neuron's positive vs. negative risk-coding scheme (Fig. 4d, left and right panels). A sliding-window regression approach (Eq. 4) identified 99 neurons encoding risk derived from the integration of probability and magnitude information that was not explained by alternative variables probability, magnitude, or expected value (which were regression covariates). The population signal for risk identified in this sliding-window regression peaked at 221 ms following cue onset (probability in specific nuclei LA/BL/BM/Ce: 261/141/161/241 ms). We also tested an alternative, dynamic risk regressor (Eq. 5) that initially reflected risk only from probability (at the time of the first cue, with high risk for $P = 0.5$ and low risk for $P = 0.25$ and $P = 0.75$, cf. Fig. 1b) before reflecting risk from probability-magnitude integration (from the time of the second cue). This dynamic risk regressor identified 106 risk-coding neurons that provided a prominent biphasic risk signal in the population of all recorded amygdala neurons that was distinct from probability, magnitude, and expected value signals (Fig. 4e). Indeed, the majority of these risk-coding neurons (93/106, 87%) did not also show significant coding of probability. In a further model, we included two variants of the risk regressor to model risk derived solely from probability and risk derived from the integration of probability and magnitude (in addition to probability, magnitude, and expected value). This model identified 103 risk neurons (21%) for risk derived from the integration of probability and magnitude and 65 risk neurons (13%) for risk derived from probability; 27 of these neurons were significant for both regressors. The time-courses of these two types of risk signals showed that risk from probability was coded prominently during the first-cue period but also during the second-cue period, whereas risk from probability and magnitude was coded only during the second-cue period (Fig. 4f). Importantly, a subset of 33 individual amygdala neurons showed a particularly unambiguous risk-coding pattern (Fig. 4g): these neurons showed risk coding both at the first, probability cue (coding risk derived from probability only, by responding differently to the highest risk indicated by probability of $P = 0.5$, compared to low risk indicated by $P = 0.25$ and $P = 0.75$), followed by a risk signal at the second, magnitude cue (coding risk derived from both probability and magnitude, according to Eq. 1). These risk signals in amygdala neurons cannot be explained in terms of either probability or magnitude coding because the initial risk signal was based purely on probability and occurred before the magnitude cue, while the second risk signal reflected the integrated probability and magnitude information.

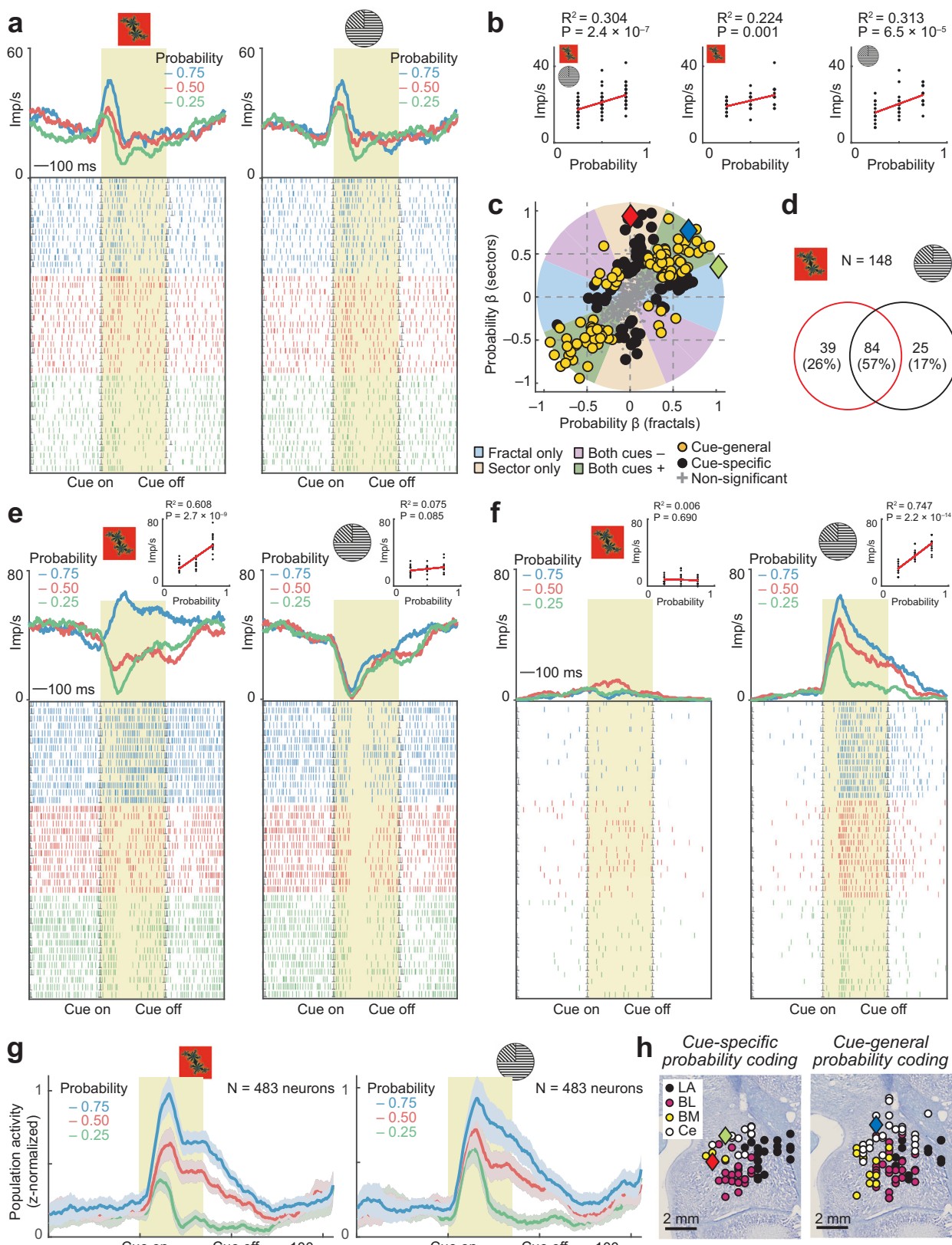

Risk neurons were similarly prevalent across amygdala nuclei (chi-square tests, $P > 0.05$; Fig. 4h). Additional examples of amygdala risk-coding neurons are shown in Fig. 4i–k (Fig. S5), illustrating risk signals with a negative coding scheme (Fig. 4i), sequential risk signals derived from probability only and the integration of probability and magnitude (Fig. 4j) and a selective response to the highest risk level (Fig. 4k).

To evaluate the robustness of our classification of neurons coding the two correlated variables magnitude and risk, we performed a bootstrap analysis (Fig. S6). Of 99 risk-coding neurons (Eq. 4), 95 showed robust risk-coding in the bootstrap (significant in more than 500/1000 bootstrap iterations; 43 neurons were significant in more than 850/1000 iterations); further, 38 of these neurons were

**Fig. 2 | Abstract and cue-specific coding of reward probability by amygdala neurons. a** Amygdala neuron encoding probability in response to both fractal cues and sector cues. Peri-event time histogram sorted by probability. Raster display: ticks indicate impulses, rows indicate trials. Yellow area: analysis period. **b** Linear regression of neuron's activity on probability across cue formats (left), and for fractal (middle) and sector (right) cues. Statistical significance in this panel and panels (**e**), (**f**) assessed with two-sided t-test (no adjustment for multiple comparisons). **c** Categorizing neurons as coding probability only for fractal or sector cues (black) or across cues (orange) from the angle in the space of probability-regression coefficients ($N = 483$; analyzing probability-cue period using fixed-window

analysis). Blue, red, green diamonds: neurons in (**a**), (**e**), (**f**). **d** Summary of results from classification of probability-coding neurons, based on fixed-window analysis (**c**). Circle size not proportional to group sizes. **e** Amygdala neuron encoding probability for fractal but not sector cues. **f** Amygdala neuron encoding probability for sector but not fractal cues. **g** Population activity of all recorded amygdala neurons (mean ± s.e.m.) in response to fractal and sector cues, sorted by cued reward probability. **h** Histologically reconstructed recording sites for probability-coding neurons. LA: lateral nucleus; BL: basolateral nucleus; BM: basomedial nucleus; Ce: central nucleus. Blue, red, green diamonds: neurons in (**a**), (**e**), (**f**). Source data are provided as a Source Data file.

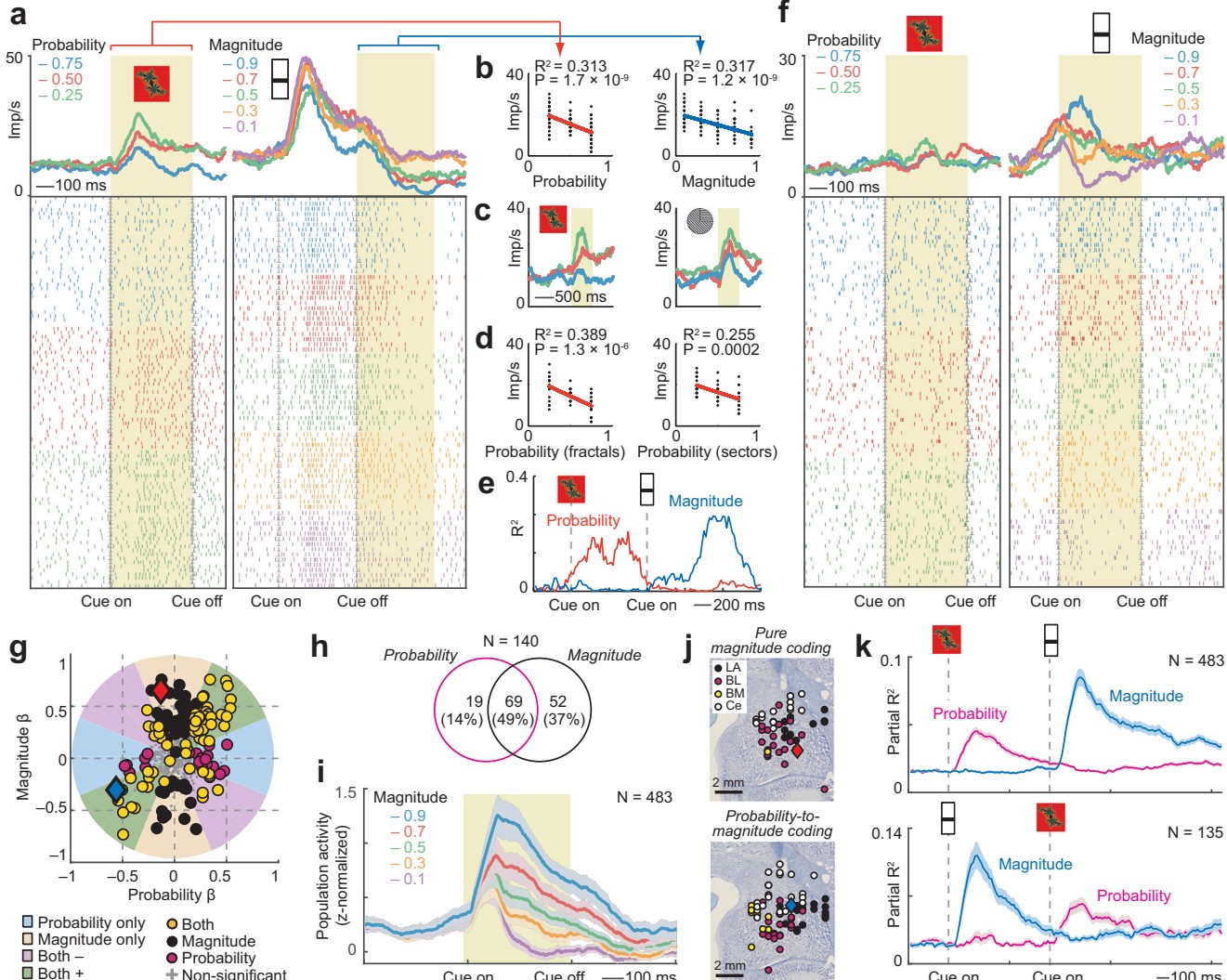

**Fig. 3 | Sequential coding of probability and reward magnitude. a** Dynamic probability-to-magnitude coding in an amygdala neuron: probability- and magnitude-coding in response to sequential cues. Peri-event time histograms sorted by probability (left) and magnitude (right). Raster display: ticks indicate impulses, rows indicate trials. Yellow areas: analysis periods. **b** Linear regression of the neuron's activity on probability and magnitude. Statistical significance in this panel and (**d**) assessed with two-sided t-test (no adjustment for multiple comparisons). **c** The neuron's activity in response to fractal and sector probability cues, sorted by probability. **d** Linear regression of the neuron's activity on probability for fractal and sector cues. **e** Partial $R^2$ for probability and magnitude for the same neuron, obtained from sliding-window multiple regression. **f** A different amygdala neuron coding only magnitude but not probability. **g** Categorizing neurons as coding magnitude (black), probability (magenta), or both (orange) from the angle in the space of value-regression coefficients ($N = 483$; analyzing first cue period for

probability and second cue period for magnitude using fixed-window analysis). Blue, red diamonds: neurons in (**a**), (**f**). **h** Summary of results from classification of probability-magnitude coding neurons, based on fixed-window analysis (**g**). Circle size not proportional to group sizes. **i** Magnitude-coding in population activity (unselected neurons; mean ± s.e.m.). **j** Histologically reconstructed recording sites for neurons coding pure magnitude (top) or probability followed by magnitude (bottom). Blue, red diamonds: neurons in (**a**), (**f**). **k** Flexible (reversible) probability-to-magnitude transitions in population activity. Top: Partial $R^2$ (sliding-window regression; mean ± s.e.m.) for probability and magnitude regressors across all recorded neurons in the main task (first cue: probability, second cue: magnitude). Bottom: Partial $R^2$ across all recorded neurons in control task with reversed cue order (first cue: magnitude, second cue: probability). Source data are provided as a Source Data file.

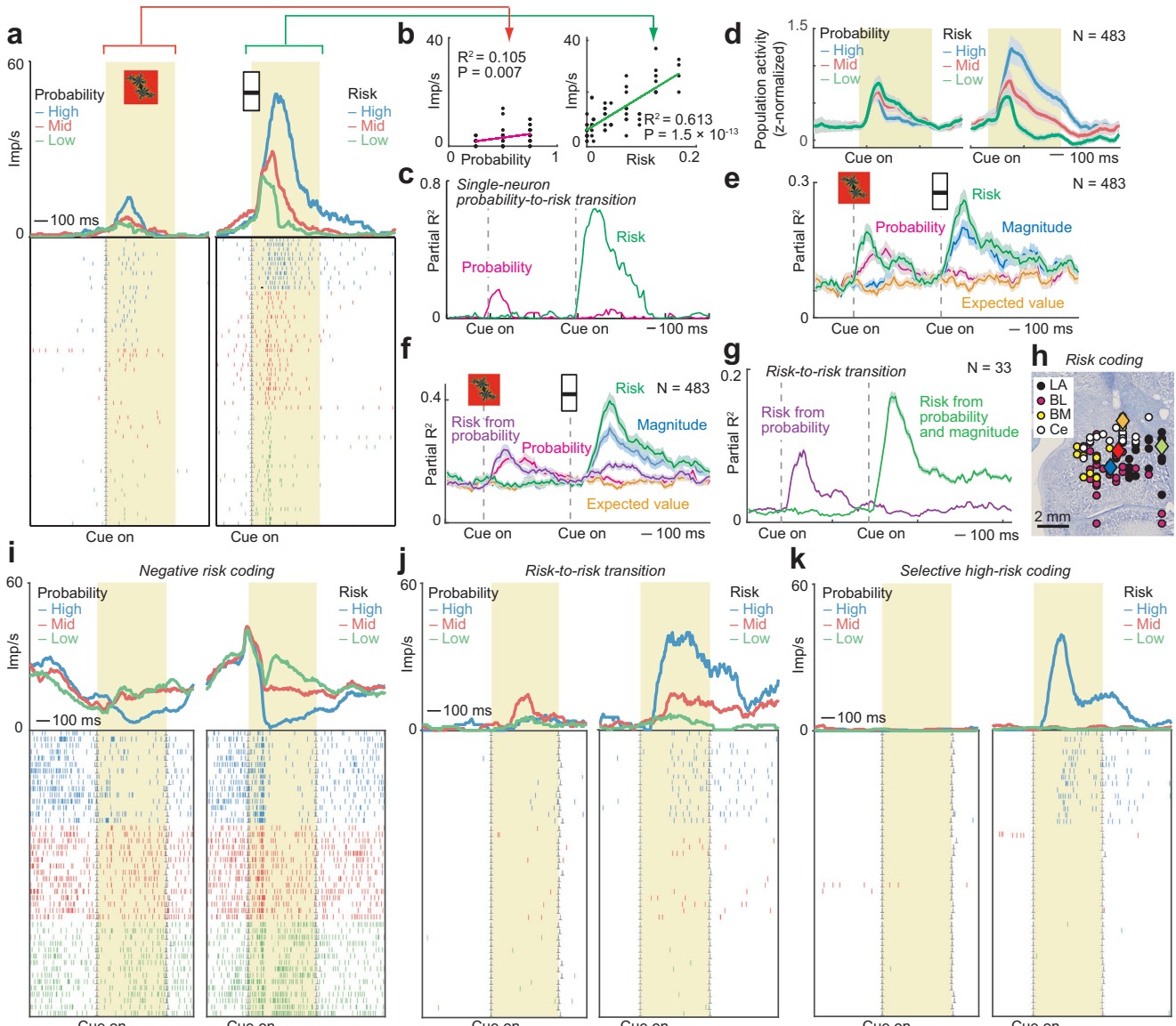

**Fig. 4 | Risk neurons in amygdala. a** Dynamic probability-to-risk coding in an amygdala neuron. Peri-event time histogram sorted by risk (variance) terciles. Raster display: ticks indicate impulses, rows indicate trials. Yellow area: analysis period. **b** Linear regression of cue responses on probability and risk. Statistical significance was assessed with a two-sided t-test (no adjustment for multiple comparisons). **c** Dynamics of the neuron's probability-to-risk transition assessed by sliding-window regression. **d** Population risk signal (mean +/− s.e.m.) in all recorded amygdala neurons. Peri-event time histogram sorted by risk (variance) terciles. Probability-related activity in left panel sorted according to each neuron's risk-coding scheme in right panel. **e** Partial $R^2$ (mean +/− s.e.m.) from a sliding-window multiple regression analysis across all recorded neurons with a dynamic risk regressor (reflecting both risk from probability only at first cue and risk from probability-magnitude integration at second cue) and other variables as covariates.

**f** Partial $R^2$ (mean +/− s.e.m.) from a sliding-window multiple regression analysis across all recorded neurons with a separate regressors for risk from probability and risk from integrated probability and magnitude. **g** Subset of amygdala neurons coding risk from probability at first cue and risk from integrated probability and magnitude at second cue (sliding-window regression). **h** Histologically reconstructed position of recorded risk neurons. Blue, red, green, orange diamonds: neurons in (**a**), (**i**)–(**k**). **i** Probability-to-risk transition in an amygdala neuron with negative coding scheme. **j** Risk-to-risk transition in an amygdala neuron. At the first, probability cue, the neuron responded strongest to the intermediate probability ($P = 0.5$), corresponding to the highest risk level. At the second, magnitude cue, the neuron reflected risk from probability and magnitude information. **k** Amygdala neuron responding selectively to high-risk at the second cue (integrated risk from probability and magnitude). Source data are provided as a Source Data file.

significant for risk but insignificant for magnitude (significant in less than 500/1000 iterations). Similarly, of 94 magnitude-coding neurons (Eq. 4), 81 showed robust magnitude-coding in the bootstrap (significant in more than 500/1000 bootstrap iterations; 48 neurons significant in more than 850/1000 iterations); 25 of these neurons were significant for magnitude but insignificant for risk. Further, among neurons classified as risk-coding, the largest fraction of bootstrap samples was also classified as risk-coding rather than magnitude-coding, and vice versa for magnitude-coding neurons (Fig. S6). Accordingly, although risk and magnitude were correlated in our dataset, substantial numbers of neurons coded these variables robustly and distinctly.

Thus, many amygdala neurons encoded economic risk as the variance of the cued statistical reward distribution on a given trial, thereby reflecting the dynamic and internal (non-cued) integration of probability and magnitude information into risk.

## Amygdala coding of expected value
Single-neuron coding of expected value (defined as the multiplicative integration of probability and magnitude) was rare, compared to the

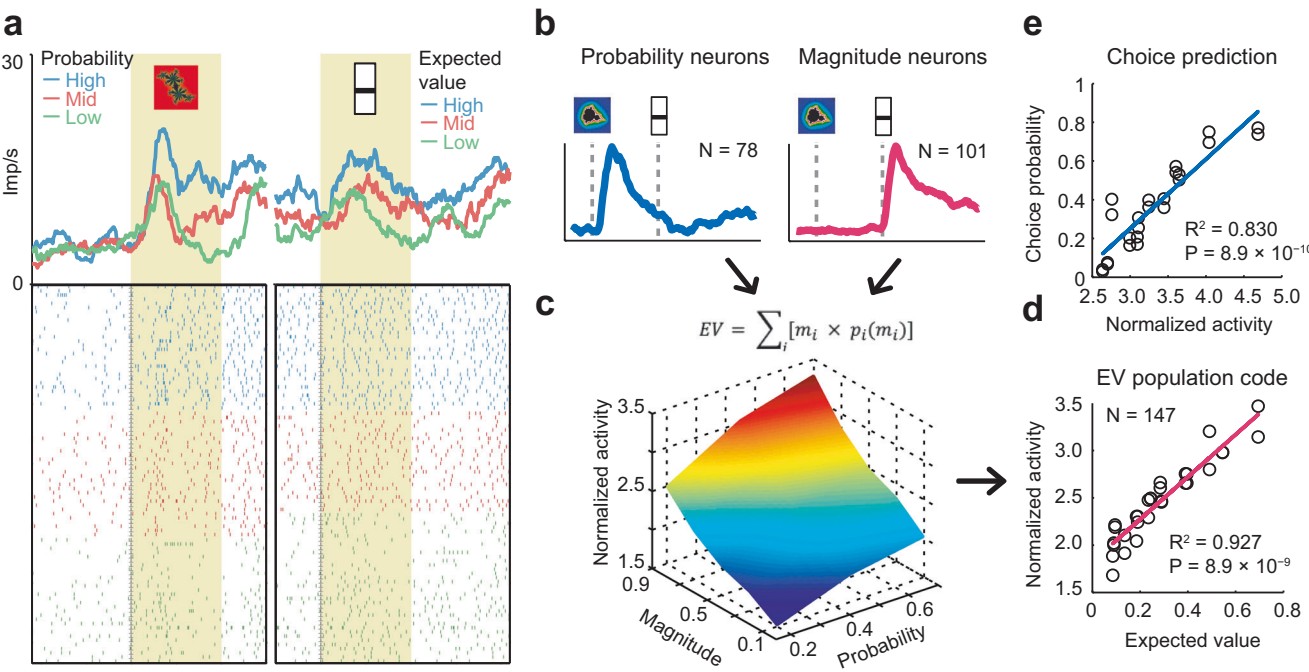

**Fig. 5 | Population coding of expected value. a** Amygdala neuron encoding expected value at the time of the magnitude cue. Peri-event time histogram sorted by value terciles. Raster display: ticks indicate impulses, and rows indicate trials. Yellow area: analysis period. **b** Population responses (mean partial $R^2 \pm$ s.e.m.), of probability-coding neurons (left) and magnitude-coding neurons (right) provides input for neuronal construction of expected-value population signal. Neurons selected using fixed-window analysis. **c** Neural expected-value signal constructed from the normalized neural responses of probability-coding neurons to different probability levels multiplied with the normalized neural responses of magnitude-coding neurons to different magnitude levels. **d** Linear regression of neuronal expected-value signal from (**c**) on objective expected value derived from cued probability and magnitude levels. Statistical significance in this panel and panel e assessed with a two-sided t-test (no adjustment for multiple comparisons). **e** Linear regression of monkeys' choice probabilities for specific expected value levels (measured in the separate choice task) on neuronal expected-value signal for corresponding levels of expected value (measured in the main recording task).

more frequent probability, magnitude, and risk signals described above. Sliding-window analysis in which the expected value regressor was included among probability, magnitude, and risk regressors identified 30 amygdala neurons with significant coding of expected value (6%, Eq. 4), which was not different from chance ($P > 0.05$, binomial test). Figure 5a (Fig. S7) shows a neuron with expected-value coding at the second cue (reflecting the integration of probability and magnitude information) that was preceded by probability coding at the first cue. (Expected value at the first cue was identical to the probability regressor, and these variables could thus not be distinguished in the first-cue period). Although single-neuron expected-value signals were rare, Fig. 5b–d illustrates how separate probability and magnitude signals, measured in amygdala single-neuron activity at the first and second cue, respectively (Fig. 5b), could be translated into an accurate population code of expected value through a multiplicate integration of the separate signals (Fig. 5c, d). The population signal of expected value constructed neuronally in this way from probability and magnitude signals tracked the objective expected value derived from cued probability and magnitude levels (Fig. 5d), indicating that neuronal probability and magnitude signals were suitable for integration into expected value. Notably, such expected-value population signals could be used to guide the monkeys' choices, as the population signals correlated with the choice probability (measured in the separate choice task) that monkeys exhibited for given expected-value levels (Fig. 5e).

We also tested whether amygdala neurons at the time of the second cue were sensitive to a change in value relative to the first cue; such value change in a trial has previously been conceptualized as 'belief confirmation'[28] or 'state value'[1]. We performed a sliding-window regression in which we included a regressor that encoded whether the

expected value at the time of the second cue was higher than the expected value of the first cue. We found that 36/483 amygdala neurons (7%) showed a significant effect for this value-change regressor. We interpret these responses as reflecting previously reported state-value coding in amygdala[1].

Thus, expected value was most prominent in the population activity rather than in individual amygdala neurons.

## Population codes for probability, magnitude, value, and risk
We used a biologically plausible nearest-neighbor classifier as in previous studies[9,12,56] to decode the main reward variables from the activity of (pseudo-)populations of amygdala neurons. We used this approach to quantify the accuracy of population codes for probability, magnitude, expected value, and risk, to relate this accuracy to the number and coding-properties of single neurons included in the decoding sample, and to examine the extent to which amygdala neurons used a shared (transferable) code for probability and magnitude. Our main results were confirmed using support-vector machine decoding (Fig. S8).

At the first (probability) cue, the classifier decoded low vs. high probability with good accuracy and expected value with lower but above-chance accuracy, while decoding of (not-yet cued) magnitude and risk was at chance level (Fig. 6a). Above-chance decoding of probability and expected value persisted after cue-offset before the second cue, although at reduced accuracy. At the second (magnitude) cue, decoding of low vs. high risk and magnitude was highly accurate while expected-value decoding was relatively lower but still above chance; these decoding accuracies persisted in the following trial epochs leading up to and including the reward period (Fig. 6a). By contrast, probability decoding was no longer significant at the second

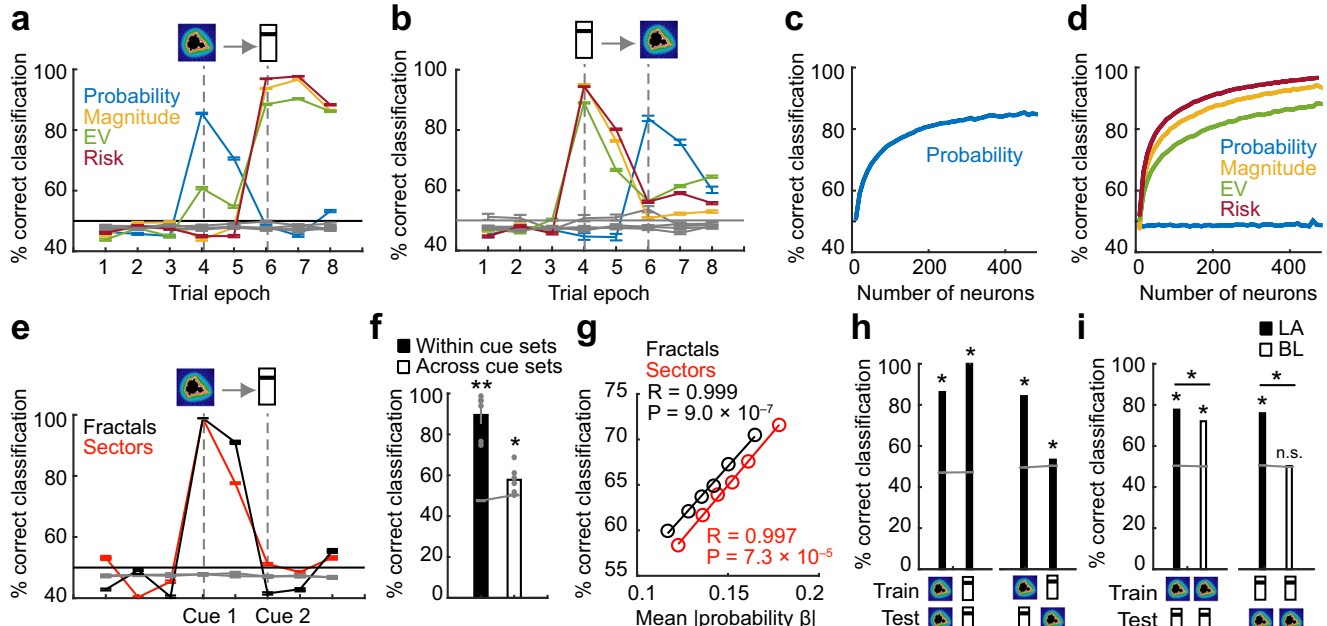

**Fig. 6 | Population decoding of value components and risk from amygdala neurons. a** Accuracy of a nearest-neighbor decoder (mean +/− s.e.m.) classifying high-vs.-low levels of probability, magnitude, expected value (EV), and risk from unselected amygdala neurons ($N = 483$) in specific trial epochs (reward delivery in period 8). Gray lines: decoding based on data using trial-shuffled group labels. **b** Decoding accuracy (mean +/− s.e.m.) in control task with reversed cue order. (One outlier neuron excluded; see Methods) **c** Decoding accuracy (mean +/− s.e.m.) for randomly selected neuro-subsets of different sizes in first-cue period. **d** Decoding accuracy (mean +/− s.e.m.) for randomly selected neuron-subsets of different sizes in second-cue period. **e** Decoding accuracy (mean +/− s.e.m.) for probability based on fractal or sector cue trials. Gray lines: decoding based on data using trial-shuffled group labels. **f** Decoding accuracy (mean +/− s.e.m.) for probability when training/testing within cue set (fractal or sector, black bar) and training/testing across cue sets (averaged over fractal-to-sector and sector-to-fractal decoding, white bar). Gray lines: decoding using trial-shuffled group labels. Gray

dots: accuracies for each decoding condition ($N = 6$; 2 cue sets, 3 low-vs.-high, low-vs.-mid, mid-vs.-high probability comparisons). Statistical significance in this panel and panels g-i assessed with two-sided t-test (no adjustment for multiple comparisons). **g** Linear regression of probability-decoding accuracy on the mean single-neuron probability sensitivity of the decoding sample, when decoding repeatedly from randomly selected small ($N = 20$) subsets of amygdala neurons. 5,000 random samples, without replacement of neurons within each sample. **h** Decoding and cross-decoding (mean +/− s.e.m.) of probability and magnitude information. Decoders were trained to discriminate low-vs.-high probability (first cue period, indicated by fractal symbol, though the analysis included both fractal and sector trials) or magnitude (second cue period, indicated by bar symbol) and tested within or across periods and reward variables. Gray lines: decoding based on data using trial-shuffled group labels. **i** Decoding and cross-decoding (mean +/- s.e.m.) performed separately for LA ($N = 113$) and BL ($N = 172$) neurons. Gray lines: decoding based on data using trial-shuffled group labels. *$P < 0.001$; n.s.: non-significant.

cue. Decoding from neuronal data recorded in the task with reversed cue order showed largely the opposite pattern (Fig. 6b). Notably, while magnitude decoding fell back to chance level at the second (probability) cue, decoding accuracy for risk and expected value remained significantly above chance, consistent with the notion that these two variables integrated probability and magnitude information.

Decoding accuracy increased with the number of amygdala neurons that were included in the decoding sample for probability at the first cue (Fig. 6c) and for magnitude, expected value and risk at the second cue (Fig. 6d); the highest decoding accuracies were obtained when all neurons were included in the decoding sample, indicating that also individually non-significant neurons contributed to population decoding, although steepest increases in decoding accuracy were observed between 1 to approximately 100 neurons.

Probability decoding remained highly accurate when decoding separately from trials in which probability was cued using fractal or sector stimuli (Fig. 6e). When training the classifier on one cue type (e.g., fractal) to decode probability from the other cue type (e.g., sector), probability decoding was significantly above chance, although accuracy was considerably lower than decoding within cue type (Fig. 6f). This reduced accuracy was largely due to low cross-cue decoding between the mid and high probability levels. Although these probabilities were distinct behaviorally (Fig. 1), discriminability across cue sets was low (52%, compared to 60% and 63% for discrimination between low and medium and low and high probabilities, respectively,

for nearest-neighbor decoder; for support-vector machine: 52% compared to 71% and 62%) perhaps related to properties of the images used to cue these probabilities. To investigate the contributions of individual neurons' probability coding to population decoding, we repeatedly ($N = 5000$ iterations) performed classification on small ($N = 20$ neurons) randomly selected subsets of neurons and related the decoding accuracy to the mean single-neuron regression coefficients of the decoding sample. This analysis confirmed that population decoding depended on individual neurons with strong probability coding and that single-neuron coding of probability for both fractal and sector trial types contributed similarly (Fig. 6g). Extending this approach to our four main reward variables and using multiple regression to determine the factors influencing decoding accuracy, we found that decoding from even small ($N = 20$) neuronal subsets was significantly above chance for all variables (Fig. S9), that probability decoding depended on both fractal- and sector-related coding of single neurons, that magnitude decoding depended strongly on single-cell magnitude coding, that risk decoding depended exclusively on risk but not probability or magnitude single-cell coding, and that expected-value decoding was explained largely by magnitude coding of single-neurons.

If sequential probability and magnitude coding reflected neural integration to value in a shared (transferable) code, then it should be possible to train the classifier on one variable (e.g., probability, using data from the first cue period) to decode the alternative variable (e.g.,

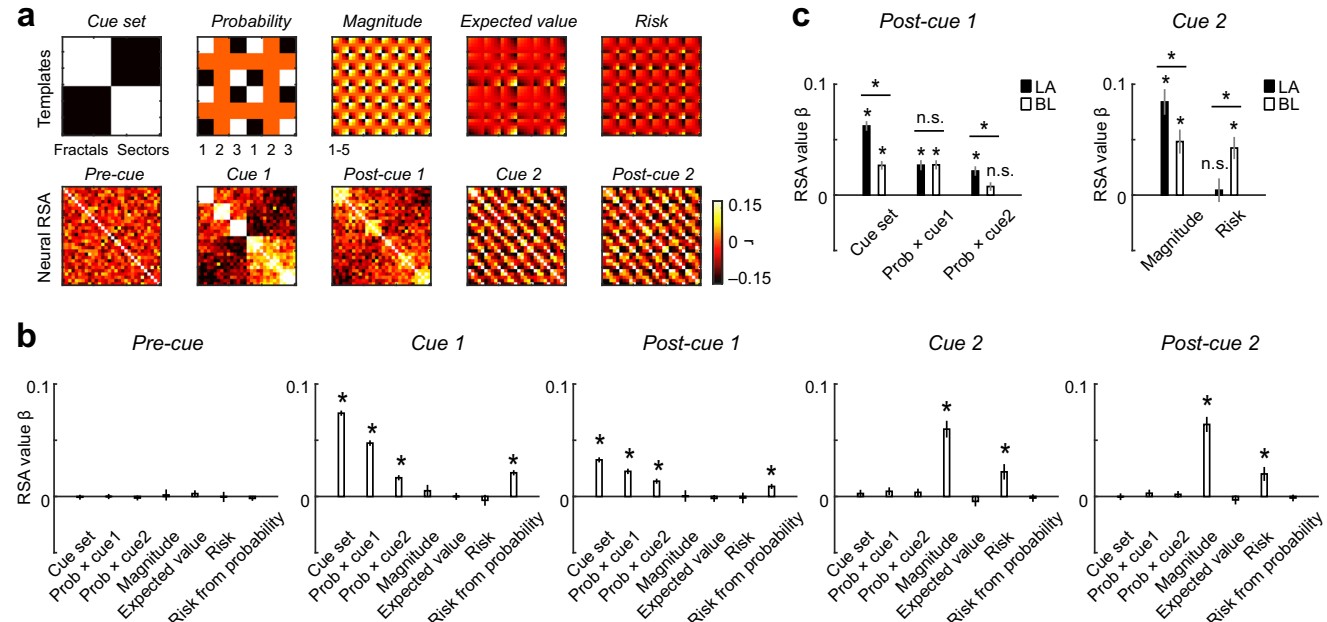

**Fig. 7 | Structure of population codes in amygdala and its subdivisions assessed with Representational Similarity Analysis (RSA). a** RSA definition of key variables and neuronal RSA matrices. Top: Templates define similarity patterns for different variables; numbers indicate probability and magnitude levels (parametric levels were also used but are not labeled for expected value and risk). Bottom: Neural RSA matrices for all neurons (*N* = 483) in different trial epochs. Color code indicates the correlation coefficient for condition pairs, calculated between population-activity vectors. Condition order was preserved across matrices. **b** Multiple regression of neuronal RSA matrices on templates (N = 483 neurons). Prob × cue1/cue2: interaction regressors modeling probability specifically on fractal/sector trials. (*P < 0.01, two-sided t-test compared to null distribution obtained from 10,000 random permutations) Bars show regression coefficients (+/− s.e.m.). **c** Multiple regressions of neuronal RSA matrices on templates for amygdala subdivisions LA (*N* = 113) and BL (*N* = 172), and comparisons of regression coefficients between nuclei (*P < 0.01, two-sided t-test compared to the null distribution obtained from 10,000 random permutations). Bars show regression coefficients (+/− s.e.m.).

magnitude in the second cue period). Using data from all recorded neurons, baseline probability and magnitude decoding (training and testing on the same data) was highly accurate (Fig. 6h, left); however, while cross-decoding from probability to magnitude (i.e., training on first and testing on second cue period) was also highly accurate, cross-decoding in the reverse direction (magnitude to probability, i.e., training on second and testing on first cue period) was much lower (but still significantly above chance, Fig. 6h, right). This pattern of results suggests that at the time of the second, magnitude cue, the amygdala population code is more specialized to reflect the variety of signals that can be decoded at this point in the trial (magnitude, risk, expected value). This complexity of the population code would likely limit the utility of a decoder trained on activity in this trial period to accurately readout probability from the earlier trial period. On the other hand, the finding that training on the probability-cue period and decoding from the magnitude-cue period was possible suggests a more generalized population value code in this early trial period.

Based on previous findings that value codes may differ between amygdala nuclei[9,12], we repeated the above analysis using neurons recorded from different nuclei. Importantly, while probability-to-magnitude cross-decoding was accurate in both LA and BL nuclei (Fig. 6i, right; LA vs. BL: *P* = 1.3 × 10$^{-72}$), the reverse magnitude-to-probability cross-decoding was only accurate in LA but not significantly above chance level in BL (Fig. 6i, right, LA vs. BL: *P* = 2.7 × 10$^{-15}$; Fig. S10). Thus, LA neurons but not BL neurons carried a common (i.e., transferable) value code for probability and magnitude that allowed flexible readout across time periods and cued value components.

Taken together, these results suggested an abstract (largely independent of stimulus type) but transient probability code, highly accurate distinct but simultaneous codes for reward magnitude and risk, and a transferable value code based on probability and magnitude information specifically in LA.

## Structure of population codes for value components and risk in amygdala and its subdivisions

We used representational similarity analysis (RSA) to assess in more detail how our key variables were represented at the population level in the amygdala and in specific nuclei. RSA quantifies the similarity (e.g., correlation) of neuronal responses to different experimental conditions (e.g., specific cue types, probability levels, etc.) to characterize local population codes[28,57].

Similarity patterns in population activity assessed with RSA identified a prominent probability code at the first cue that transitioned into a magnitude code at the second cue (Fig. 7a). In addition, cue type (colored fractal vs. monochrome sector) was strongly reflected in population activity during the first cue period. The probability code in the first cue phase and post-cue phase was abstract as shown by significance of both cue-specific probability regressors (Fig. 7b; fractals: *P* < 1.0 × 10$^{-15}$; sectors: *P* = 2.3 × 10$^{-11}$; non-parametric permutation test). Risk derived purely from probability was also significantly represented in population similarity patterns (Fig. 7b). Importantly, at the second cue, both magnitude and risk regressors were significant in the multiple regression, suggesting that both variables were prominently represented in population activity. By contrast, expected value was not significant.

We also calculated RSA similarity patterns separately for different amygdala nuclei, focusing on the comparison between LA and BL for which we had sufficient numbers of neurons. Notably, an abstract probability code indicated by individually significant cue-specific probability regressors was present in the LA but not in the BL (Fig. 7c, left). By contrast, although magnitude coding was significant in both BL and LA, risk coding was only significant in BL but not in LA (Fig. 7c, right). These results were confirmed in individual animals, although risk coding in animal A was not significant across all neurons but only significant for BL neurons.

Taken together, the patterns of representational similarity in population activity confirmed prominent representations of probability, magnitude, and risk. Furthermore, this analysis identified regional amygdala specializations for an abstract probability code in LA and for a risk code in BL.

## Discussion

Our data show that primate amygdala neurons respond to sequentially viewed visual stimuli by dynamically signaling information about different reward attributes and the related statistics of predicted reward outcomes. Many amygdala neurons had biphasic responses to sequential cues that signaled the value components probability and magnitude in a dynamic (temporally patterned), flexible (reversible) and mutually transferable code. Other neurons were sensitive to either probability or magnitude information but not both, revealed by selective responses to temporally separated cues. Probability coding in single neurons and in the neural population was abstract, as it largely generalized across different visual-cue formats. Finally, a substantial number of neurons integrated probability and magnitude information into risk signals that quantified the variance of predicted rewards. Importantly, neurons were sensitive to both, risk derived from probability alone and to risk derived from integrated probability and magnitude (including when both risk variables competed in the same regression). This diversity of dynamic, selective and integrative neural codes suggests a rich representation of reward attributes and economic decision variables in the primate amygdala.

The animals' well-structured preferences between gambles and risky choice options and relationships between reward variables, motivational engagement and reward expectation (Fig. 1, Tables 1–3) suggest that neuronal codes for reward attributes are suited for informing reward expectations and economic choices between goods. Our aim in this study was to provide evidence on how single amygdala neurons responded to visual stimuli that signaled information about reward probability, magnitude, and their integration into expected value and risk. These variables encoded by amygdala neurons recorded during the Pavlovian task were shown to be behaviorally relevant for the monkeys (trial-abortion and lick rates in the recording task, choices in separate tests). Our data therefore establish a basis for investigating in future studies the separate, important question of how these variables encoded by amygdala neurons are incorporated into neural decision computations.

Many amygdala neurons carried an abstract, stimulus-independent probability code that generalized across different cue types while other neurons showed cue-specific probability coding (Fig. 2). This abstract population code for probability depended on individual neurons' probability-sensitivities across cue types (Fig. 6g). It was prominent in LA, the amygdala's sensory entry point and likely storage site of stimulus-reinforcer associations[58–60], but not in BL, a downstream structure that integrates inputs from LA, prefrontal cortex and hippocampus[61] (Fig. 7c). This result could suggest that signals from cue-specific probability-coding neurons converge onto abstract probability-coding neurons within LA, followed by further differentiation of this information in BL. This interpretation is consistent with previously reported object-value signals in primate LA that generalize over social 'self-other' contexts[9], as well as proposals that BL processing elaborates LA-mediated cue responses[58] and evidence implicating BL in decision computations beyond valuation[12].

Although decoding performance for probability across cues was relatively low, it was significantly above chance. Previous studies showed that different types of population code emphasize either the ability to generalize across conditions or the coding capacity, i.e., the extent to which many different task variables and their combinations can be read from the code[62]. The present result suggests that amygdala neurons in our task may implement a population code that allows significant but limited generalization, though many individual neurons

showed clear cue-invariant probability encoding. Future studies could use more complex designs to determine the structure of the reward code in primate amygdala and its capacity for high-dimensional coding vs. generalization.

Amygdala neurons frequently encoded dynamic probability-to-magnitude transitions that were shown to be flexible (reversible) in a control test with reversed cue presentation (Fig. 3k). Such single-cell sensitivity to probability and magnitude is remarkable as these variables represent fundamentally different sources of reward information, indicating respectively the likelihood and the quantity of reward. Notably, reward magnitude was indicated by a third type of visual stimulus (bar height, different from fractal and sector probability cues), suggesting considerable abstraction from visual cues in the primate amygdala. Importantly, population decoding across cue periods demonstrated that neural codes for probability and magnitude were mutually transferable, particularly in LA (Fig. 6i). This result supports the concept that amygdala neurons dynamically integrate multiple reward attributes into value, which could involve feed-forward convergence from specialized probability- and magnitude-coding neurons onto value-coding neurons. That BL did not exhibit a shared value code suggests more differentiated processing of probability and magnitude in this amygdala subdivision, including the integration of these attributes into risk, as discussed next.

A subset of amygdala neurons directly reflected the integration of probability and magnitude into risk signals that quantified the statistical variance of the current reward distribution (Fig. 4a). Importantly, the animals were not explicitly trained to discern this abstract statistical parameter from the separate probability and magnitude cues. Nor were these two cue types individually sufficient to signal risk but information derived from them had to be 'internally' integrated to represent risk. Behavioral data confirmed that both monkeys used risk information derived from the cues to calibrate their motivational engagement during the Pavlovian task (evidenced by higher trial abortion rates for higher risk, Fig. 1j) and to express their individual risk attitudes in a separate choice task (Fig. 1g). Lick rates depended on a combination of these variables, and reflected risk in a manner that differed between animals, although both animals showed higher lick rates for probability and magnitude, consistent with previous studies[52–54]. Risk coding was particularly clear in a subset of neurons that signaled risk already in response to the probability cue (differentiating high risk, implied by reward probability $P = 0.5$, from low risk, implied by reward probabilities $P = 0.25$ and $P = 0.75$), followed by an integrated risk signal in response to the magnitude cue (Fig. 4g).

Different from previously reported risk signals in response to explicit cues[14,22,24,27,34,35,37], risk signals reported here were constructed internally and dynamically, i.e. from temporally separated probability and magnitude information. One previous study showed that neurons in dorsolateral prefrontal cortex derive risk in the absence of explicit cues from the variance of recently experienced rewards[14]. The present study demonstrates a different mechanism for deriving risk involving the integration of temporally separated probability and magnitude cues that together specified the current reward distribution. The observed diversity of risk signals and dynamic coding transitions (Fig. 4) suggest that some amygdala neurons may have a high threshold for risk signaling that requires joint probability and magnitude information, perhaps to represent an outcome distribution as precisely and unambiguously as possible for guiding economic choices. These neurons may receive information from more flexible risk neurons that signal partial risk information, consistent with the diversity of neurons reported here. Remarkably, although risk neurons were prevalent across amygdala nuclei, BL but not LA carried a prominent population risk code as determined by representational similarity analysis, whereas magnitude was similarly represented in both nuclei (Fig. 7c). Population decoding confirmed that this risk code depended on individual neurons' sensitivities for risk but not

magnitude, providing further support for the simultaneous but separate coding of magnitude and risk in amygdala. Different neurons encoded probability at the first cue with a positive or negative coding scheme, and some of these neurons transitioned to encoding risk at the time of the second cue. Although too few neurons showed probability-to-risk transitions to allow quantitative analyses (13/106), it is possible that neurons with positive and negative coding schemes play different roles in the computation of subsequent risk, which could be investigated in future studies.

Relatively few amygdala neurons directly encoded expected value (defined by the multiplicative integration of probability and magnitude), although we found that the activity of probability-coding and magnitude-coding neurons was suitable for constructing expected value (Fig. 5). This result could suggest that expected value was not strongly represented in amygdala single neurons in this task, which required integrating probability and magnitude across sequential cues (despite monkeys' separately measured choices reflecting expected value). In a previous study using a choice task, we showed that amygdala neurons do encode subjective values that reflected integrated reward probability and magnitude when these reward attributes were cued simultaneously[12]. Perhaps neurons in the prefrontal cortex, including the orbitofrontal cortex, and parietal cortex might be relatively more important in signaling to accumulate decision variables derived from sequential or otherwise complex cues[28,63–66]. Some previous studies found largely similar coding of values and choices in the amygdala and orbitofrontal cortex[13,67], while others emphasized differences in the time courses with which neurons in these structures track changing values[53,68], and in the specificity with which single neurons encode complex, multisensory food rewards[69]. Future studies could combine our task with simultaneous recordings from the amygdala and orbitofrontal cortex to uncover possible differences in how these two structures are able to integrate temporally dissociated value components to guide behavior. Although few individual neurons explicitly encoded expected value (identified with multiple regression), population decoding of expected value was relatively strong. It is possible that individual neurons with subthreshold encoding of expected value or separate probability and magnitude coding contributed to this high decoding accuracy. However, while the decoding approach is suitable for quantifying what information can be decoded from the population, it does not (in its basic form used here) allow for direct competition between different, alternative variables in explaining neural activity, which we instead implemented with our single-neuron GLM approach. Thus, it is possible that correlated variables (e.g., reward magnitude) contributed to significant population decoding of expected value.

Our use of a Pavlovian task with sequential cue presentations enabled us to temporally dissociate probability and magnitude information, to investigate whether individual amygdala neurons responded selectively to one or both of these value components, and whether they integrated these components to risk and expected value. Further, we could examine neuronal responses to individual reward-predictive cues, in the absence of additional cognitive processes that occur during choice tasks, including value comparisons and decision computation. A possible limitation is that our task did not explicitly require integrating reward components for decision-making, although trial-abortion rates and lick responses suggest that the animals used information derived from the cues for reward expectation. Moreover, previous studies showed that amygdala encoding of additional variables such as trial-specific choices, internally planned choice sequences, and internally tracked accumulated rewards was specific to free-choice situations and largely absent during imperative, non-choice tasks[10,11,19]. Thus, it will be interesting to study amygdala neurons in a choice situation with the cues used in the present task.

Our study builds on previous work that documented neuronal value coding from multiple reward attributes, although attribute-specific coding has received less attention so far. One study reported neurons in different frontal-lobe areas that encoded the probability of reward irrespective of cue set, either specifically or together with other decision-relevant variables (magnitude, effort)[23]. An influential series of studies showed that neurons in the parietal and prefrontal cortex respond to sequential cues conveying probabilistic evidence and integrate this information over time into a decision variable that guided monkeys' decisions[63–65]. Similarly, a more recent study showed that when monkeys sequentially inspect probability and magnitude cues for decision-making, neurons in different frontal-lobe areas process cue-derived values that guide the monkeys' choices[28]. Value coding in different brain areas has been shown to reflect the integration of probability, magnitude, reward type, temporal delay, and effort[10,11,13,14,16,24,54,67,70–84]. Amygdala neurons signal the value of visual stimuli in cue-specific or cue-general ways[2], including during decision-making[12] and in social contexts[9,17]. Extending these findings, we show here that amygdala neurons reflect a diversity of response properties: some neurons coded probability in an abstract, cue-independent format while others coded cue-specific probability; some neurons were specialized for probability or magnitude coding while others dynamically processed both probability and magnitude and their integration to value and risk. A recent study reported similarly specialized coding of probability and flavor in the ventrolateral prefrontal and orbitofrontal cortex, respectively[85], suggesting regional differences in coding reward attributes. The co-existence in the amygdala of neurons processing specific reward attributes and neurons with dynamic coding properties reported here identify a single-cell and population basis for integrating multiple reward attributes into value and risk.

Our data add to the emerging evidence of coding differences between primate amygdala nuclei[11,12,14,39,86]. We found that LA carried a prominent abstract probability code that generalized across cue formats (Fig. 7c) and that could be read flexibly from neuronal activity elicited by probability and magnitude cues, irrespective of which period was used for decoder training and testing (Fig. 6i). These results indicate strong separation of different value levels associated with visual stimuli in LA, irrespective of whether value derived from probability or magnitude. This pattern of findings contrasts markedly with population codes in BL, where probability coding and value coding from probability and magnitude were more specific. BL is an integrative structure that receives distinct prefrontal and hippocampal inputs and provides outputs to the prefrontal cortex, striatum, and physiological effector structures in centromedial amygdala[7,87], which could suggest that its population codes are more specifically formatted. Consistent with this notion, BL but not LA showed particularly strong coding of risk, separately from value (Fig. 7c). It is possible that the observed dynamic coding patterns in amygdala nuclei reflect inputs from other brain structures such as orbitofrontal cortex and subgenual cingulate cortex. A recent study[88] reported that basolateral amygdala neurons during reward anticipation signaled reward presence and type with more extended activity patterns than neurons in the subcallosal anterior cingulate cortex and rostromedial striatum. A risk code in BL is consistent with BL inputs from the orbitofrontal cortex[61,89], where neurons also code risk[22,24]. Although our findings should be confirmed using simultaneous recordings across nuclei[86], our data suggest that LA and BL make partly different contributions to processing and integrating value components, with LA carrying more abstract and generalized value codes whereas BL reflects more specific and integrative processing, including the coding of risk.

In summary, our data reveal a diversity of dynamic, specialized, and integrative codes for multiple reward attributes in the primate amygdala. Individual amygdala neurons dynamically coded different reward attributes, expressed as sequential, biphasic reward-probability, and reward-magnitude signals, and their integration over time to value and risk. Such dynamic coding of reward and decision variables could contribute to recently described amygdala activity

patterns coding goal-oriented behavioral sequences[46]. Few individual neurons encoded expected values, though expected values could be recovered by combining probability and magnitude signals. It is possible that neurons in other brain areas such as the orbitofrontal cortex encode expected value more directly in Pavlovian tasks, which may require less integration into expected value compared to instrumental, decision-making tasks. The amygdala's value and risk signals could be distributed widely, including through recently identified wide-spread branching patterns of individual amygdala neurons targeting frontal-subcortical structures[90]. Our findings inform current debates on the relative importance of mixed selectivity compared to specifically tuned neurons[41,42] by revealing how dynamic (temporally-patterned) and integrative coding properties of individual amygdala neurons translate into accurate and flexible (i.e., transferable between reward attributes) population codes. The presence of probability and risk signals in the primate amygdala reported here may also have implications for understanding the amygdala's role in depression and anxiety[43–45]. Specifically, our findings raise the possibility that dysfunctional signaling of the likelihood, amount, and risk of future rewards by amygdala neurons could contribute to compromised mental health and impaired motivation in these conditions.

## Methods

### Animals and ethical approval

Two adult male rhesus monkeys (Macaca mulatta) weighing 10.5 and 12.3 kg were used in the experiments. This number is typical for primate neurophysiology experiments. The animals had free access to a standard diet for laboratory macaques before and after the experiments and received their main liquid intake in the laboratory during recording periods. Blinding was not relevant as no experimental groups were studied. The animals were group-housed and provided with environmental enrichment for foraging activities and cognitive engagement. For daily experiments, the animals were trained to enter a primate chair voluntarily, using positive reinforcement, in which they were transported to the laboratory room for behavioral testing and neurophysiological recordings. All animal procedures conformed to US National Institutes of Health Guidelines. The work has been regulated, ethically reviewed and supervised by the following UK and University of Cambridge (UCam) institutions and individuals: UK Home Office, implementing the Animals (Scientific Procedures) Act 1986, Amendment Regulations 2012, and represented by the local UK Home Office Inspector; UK Animals in Science Committee; UCam Animal Welfare and Ethical Review Body (AWERB); UK National Center for Replacement, Refinement and Reduction of Animal Experiments (NC3Rs); UCam Biomedical Service (UBS) Certificate Holder; UCam Welfare Officer; UCam Governance and Strategy Committee; UCam Named Veterinary Surgeon (NVS); UCam Named Animal Care and Welfare Officer (NACWO).

### Neurophysiological recordings

Experimental procedures for single-neuron recordings from the amygdala in awake, behaving macaque monkeys followed those described previously[9,10]. A titanium head holder and recording chamber (Gray Matter Research) were fixed to the skull under general anesthesia using isoflurane, standard intraoperative monitoring and aseptic conditions. During surgery, normal body temperature was maintained by a homeothermic blanket, liquid was supplemented by physiological saline, and analgesia was provided for as long as necessary in the postoperative period. The anatomical position of the amygdala was located based on bone marks on coronal and sagittal radiographs referenced to the stereotaxically implanted chamber[91]. We recorded the activity of single amygdala neurons from extracellular positions. We used standard electrophysiological techniques including on-line visualization and threshold discrimination of neuronal impulses on oscilloscopes. We aimed to recorded representative samples of neurons from the lateral, basolateral, basomedial and centromedial nuclei of the amygdala. We inserted a stainless-steel tube (0.56 mm outer diameter) to guide a single tungsten microelectrode (0.125 mm diameter; 1- to 5-MΩ impedance, FHC Inc.) through the dura. The microelectrode was advanced vertically in the stereotaxic plane with a hydraulic micromanipulator (MO-90; Narishige, Tokyo, Japan). Neuronal signals were amplified, bandpass filtered (300 Hz to 3 kHz), and monitored online with oscilloscopes. Behavioral data, digital signals from an impulse window discriminator, and analog eye position data were sampled at 2 kHz on a laboratory computer with MATLAB (Mathworks Inc.). Analog impulse waveforms were recorded at 22 kHz with a custom recording system and sorted offline using cluster-cutting and principal component analysis (Offline sorter; Plexon). We used one electrode per recording day and recorded between 1 and 10 neurons per day. We sampled activity from about 1000 amygdala neurons and typically recorded and saved the activity of those neurons that appeared to respond to any task event during online inspection of several trials. We aimed to identify task-responsive neurons but did not preselect based on particular response characteristics. This procedure resulted in a database of 483 neurons that were recorded and analyzed statistically. The number of neurons is similar to those reported in previous studies on primate amygdala.

### Reconstruction of neuronal recording sites

After data collection was completed, the animals received an overdose of pentobarbital sodium (90 mg/kg iv) and were perfused with 4% paraformaldehyde in 0.1 M phosphate buffer through the left ventricle of the heart. We reconstructed the recording positions of neurons from 50-μm-thick, stereotaxically oriented coronal brain sections stained with cresyl violet based on electrolytic lesions (15–20 μA, 20–60 s, made in one animal) and lesions using cannulas placed to demarcate recording areas, by recording coordinates for single neurons noted during experiments, and in reference to brain structures with well-characterized electrophysiological signatures recorded during experiments (internal and external globus pallidus, substantia innominata)[92]. We assigned recorded neurons to amygdala nuclei based on reconstructed recording positions and a stereotaxic atlas[93] at different anterior-posterior positions (figures in the paper show neuron locations collapsed over anterior-posterior levels).

### Pavlovian task used for neurophysiological recordings

Two monkeys performed in a Pavlovian task with sequentially presented conditioned stimuli indicating the probability (as first cue) and magnitude (as the second cue) of predicted liquid rewards under computer control (Fig. 1c). The monkey sat in a primate chair (Crist Instruments) in front of a horizontally mounted touch screen for stimulus display (EloTouch 1522 L 15'; Tyco). Each trial started when the background color on the touch screen changed from black to gray. To initiate the trial, the monkey had to place his hand on an immobile, touch-sensitive key. The presentation of the gray background was followed by the presentation of a central ocular fixation spot (1.3° visual angle). The animal was then required to fixate this spot within 4° for 500 ms and maintain fixation until reward delivery. The fixation spot was followed by the presentation of a visually conditioned stimulus in the center of the screen for 500 ms that indicated reward probability, drawn from a set of six stimuli (Fig. 1d). The probability stimuli were colored fractals or monochrome circular 'sector' stimuli; each sector stimulus consisted of two sectors distinguished by black-white shading at horizontal and oblique orientation with the amount of horizontal shading indicating the probability of obtaining the cued reward magnitude. Each stimulus predicted a forthcoming reward with a specific probability of $P = 0.25$, $P = 0.5$, or $P = 0.75$. This first cue period was followed by a 500-ms inter-stimulus interval which was followed by the second cue period, in which we presented a monochrome bar stimulus with the vertical position of the black horizontal

bar against a white background indicating the magnitude of the predicted reward (drawn from a set of five magnitudes: 0.1, 0.3, 0.5, 0.7, 0.9 ml). A 500-ms inter-stimulus interval followed the second cue period before reward delivery. Reward delivery was followed by a trial-end period of 1000–2000 ms which ended with extinction of the gray background. The next trial started after an inter-trial interval of 2000–4000 ms (drawn from a uniform random distribution). A recording for a given neuron would typically last 90-160 trials.

Reward probabilities and reward magnitudes varied pseudorandomly on a trial-by-trial basis. The specific reward probabilities and magnitudes were chosen based on pre-testing to ensure that the animals maintained high motivation during the task while at the same time providing sufficient variation to study neuronal activities related to probability, magnitude, expected value, and risk. Together, the combination of the reward probability and reward magnitude cue on a given trial specified a probability distribution of possible reward magnitudes that could be delivered on that trial. Accurate reward prediction required the monkeys to combine information about the transiently cued reward probability and magnitude internally. A computer-controlled solenoid valve delivered liquid (juice) reward from a spout in front of the monkeys' mouth. On each completed trial (without fixation breaks), the monkey received one of two outcomes: on 'rewarded' trials, we delivered a liquid reward corresponding to the cued reward magnitude in ml whereas on 'non-rewarded' trials, a small reward of 0.05 ml was delivered. We used small rewards rather than non-reward as we found that a small reward ensured that the animals maintained high motivation during recordings.

We defined risk as follows (Fig. 1a). The variance of a probability distribution is defined as:

$$\text{var} = \sum_{ij} p_i \times \left( m_j - EV_{ij} \right)^2$$

which, when applied to our probability distributions, is:

$$\text{var} = (1 - p_i)\left( 0.05 - EV_{ij} \right)^2 + p_i\left( m_j - EV_{ij} \right)^2$$

where $p_i$ is the probability of reward associated with cue $i$; $m_j$ is the reward magnitude associated with cue $j$; and $EV_{ij}$ is the expected value of composite cue $ij$: [$EV_{ij} = p_i m_j$]. Importantly, the animals received a small reward of 0.05 ml when they did not receive the larger reward ($m_j$).

For a subset of recorded neurons, we repeated the main Pavlovian task but with reversed cue order, i.e., presenting the reward-magnitude cue as the first cue followed by the reward-probability cue to test the flexibility of neuronal responses to these two reward components.

Possible errors in performance included failure to make contact with the touch-sensitive key before the trial, key release before trial completion, failure to fixate the central fixation spot at trial start, or fixation break in the period between initial fixation and reward delivery. Errors led to a brief time out (3000 ms) with a black background followed by trial repetition. We usually interrupted task performance after three consecutive errors. Fixation was continually monitored by the task program during all of these periods and fixation breaks resulted in an error trial. The animals were required to place their hand on a touch-sensitive key to initiate each trial and keep their hand in place on the key until trial completion.

We monitored the animals' licking behavior by interruptions by the animal's tongue of an infrared light beam 4 mm below the spout from which liquid rewards were delivered. The position of the spout relative to the animals' mouth was carefully adjusted before each recording session.

During the periods of neurophysiological recordings and behavioral testing, the animals received their main daily liquid intake during task performance, supplemented by additional liquid after the testing sessions if required. The animals had free access to food in their home cage.

Stimuli and behavior were controlled using MATLAB (The Mathworks) and Psychophysics toolbox (version 3.0.8). The laboratory was interfaced with data acquisition boards (NI 6225; National Instruments) installed on a PC running Microsoft Windows 7.

## Behavioral choice task

We performed a separate choice task using the stimuli from the Pavlovian task to study the monkeys' preferences for reward probabilities, reward magnitudes, and associated expected value and risk levels and to confirm that the monkeys could use the information provided by these stimuli to make meaningful, reward-maximizing choices (preferring higher over lower reward probabilities and higher over lower reward magnitudes). The choice task was performed during the period of neurophysiological recordings but typically on separate testing days. In the choice task, reward magnitude was represented by the same bar stimuli used in the main task, and the probability of reward was conveyed by the same fractal or sector stimuli, presented adjacent to the bar stimulus. On each trial, the animal made a choice between two gambles, one of which was a 'safe' option, or 'degenerate gamble' (reward probability of $P = 1$, trial-by-trial varying reward magnitudes), presented randomly in left-right arrangement on the monitor. The safe option was cued only by a reward-magnitude cue, implying a reward probability of $P = 1$. For risky gambles, the cued reward magnitude could be obtained with the cued probability and a fixed small reward (0.05 ml) could be obtained with $P = 1 -$ cued probability. Each trial started when the background color on the touch screen changed from black to gray. The trial start was similar to the main task. After 500 ms, the two choice options appeared on the touch monitor in a left-right arrangement, followed after 750 ms by the presentation of two blue rectangles below the choice options at the margin of the monitor, close to the position of the touch-sensitive key on which the animal rested its hand. The animal was then required to touch one of the targets within 1500 ms to indicate its choice. Once the animal's choice was registered, the unchosen option disappeared and after a delay of 500 ms, the chosen object also disappeared and a liquid reward was given to the acting animal. Reward delivery was followed by a trial-end period of 1000 ms which ended with extinction of the gray background.

## Task training

Following habituation to the laboratory environment and experimental set-up, we trained the animals over successive steps to drink liquid reward from the spout, place their hands on a touch-sensitive key and hold the touch key for increasingly longer periods to receive reward, to view different visually conditioned stimuli that resulted in reward delivery, to touch and choose between visual stimuli on a touch screen, to choose between visual stimuli based on fixed stimulus-associated reward probability or cued reward magnitude, to choose between visual stimuli under conditions of varying reward probability or magnitude, to choose between stimuli that varied in both reward probability and reward magnitude, to perform the task under head-fixation, to perform the task under gradually increasing visual fixation requirements including saccade choices. We progressed from task training to recording once the animals were implanted with recording chambers and when their performance had reached an asymptotic level. These training periods, including the development of the tasks, lasted approximately 24 and 18 months for animals A and B.

## Behavioral data analysis

**Mixed-effects multinomial logistic regression on choice data and fixation break data.** We used mixed-effects multinomial logistic regression analysis (*fitglme* function, MATLAB) to model the animals' trial-by-trial choices across testing sessions. Specifically, we modeled choices for the right- or left-presented option separately for each animal and specified the categorical session number (*Session*) as the group variable to account for session-by-session variations (random effects). We estimated both main effects and random effects for all relevant regressors. The response variable was the dichotomous right (*RightChosen* = 1) or left (*LeftChosen* = 0) trial-by-trial choice, from $S_k$ sessions in monkey $k$ ($S_k \in \mathbb{N}, k = 1, 2$). Within the generalized linear mixed model framework with logit function as the link function, the logistic regression model is specified as follows:

$$logit\left(\pi_{ij}^L\right) = \log\left(\frac{\pi(RightChosen_{ij} = 1)}{\pi(LeftChosen_{ij} = 0)}\right)$$
$$= \mathbf{x}'_{ij}\boldsymbol{\beta} + \mathbf{z}'_{ij}\mathbf{u}_i + \varepsilon_{ij}, \varepsilon_{ij} \sim Normal(0, \sigma^2)$$

where $\pi_{ij}^L$ denotes the probability of choosing the right option in the $j$ th trial of the session $i$ ($j = 1, 2, \ldots, T_i \in \mathbb{N}; T_i =$ total trial number in session $i$); $\mathbf{x}_{ij}$ is a vector of trial-by-trial predictors (fixed-effect regressors; see below) and $\mathbf{z}_{ij}$ is a vector of trial-by-trial predictors nested in $\mathbf{x}_{ij}$, and the effects of these predictors vary across sessions (random-effect regressors). The model estimated the coefficients of fixed-effect regressors, $\boldsymbol{\beta}$, and the session-wise variations of the random-effect regressors, $\mathbf{u}_i$. The estimated left-right choice responses, $p_{ij}^L$, were derived by reverse logit function conditional on the session-wise random effects ($\mathbf{u}_i$), and the session-wise regression coefficients ($\boldsymbol{\eta}_i$) were derived from the fixed-effect coefficients ($\boldsymbol{\beta}$) and the session-wise calibration terms ($\mathbf{u}_i$).

$$p_{ij}^L = P(RightChosen = 1|\mathbf{u}_i) = \frac{\exp(\mathbf{x}'_{ij}\boldsymbol{\beta} + \mathbf{z}'_{ij}\mathbf{u}_i)}{1 + \exp(\mathbf{x}'_{ij}\boldsymbol{\beta} + \mathbf{z}'_{ij}\mathbf{u}_i)} \in [0, 1], \boldsymbol{\eta}_i = \boldsymbol{\beta} + \mathbf{u}_i$$

In the main model (Table 1), we included the following regressors. Importantly, we specified the categorical session number (*Session*) as the group variable to address session-wise variations of nutrient sensitivities as follows,

$$logit(RightChosen) = \beta_0 + \beta_1 \times (ProbRight - ProbLeft)$$
$$+ \beta_2 \times (MagRight - MagLeft)|Session$$

where *ProbRight* and *ProbLeft* indicated the trial-specific reward probabilities associated with the right and left option; *MagRight* and *MagLeft* indicated the trial-specific reward magnitudes associated with the right and left option. The coefficients from this model are shown in Fig. 1f and Table 1. In a second model, we included the following regressors:

$$logit(RightChosen) = \beta_0 + \beta_1 \times (EVRight - EVLeft)$$
$$+ \beta_2 \times (RiskRight - RiskLeft)|Session$$

where *EVRight* and *EVLeft* indicated the trial-specific expected values associated with the right and left option; *RiskRight* and *RiskLeft* indicated the trial-specific risks (variances) associated with the right and left option. The coefficients from this model are shown in Fig. 1g and Table 1. The logistic regression of choices on expected value and risk was performed separately for low ($p <= 0.25$) and high ($p > 0.25$) probability trials. Subjective values derived from this equation were used to construct the psychometric functions shown in Fig. 1h. In a third model, we included the following regressors to examine the factors related to the animals' eye-fixation breaks (i.e., errors in task performance) and lick responses (in separate regressions):

$$y = \beta_0 + \beta_1 \times (Probability) + \beta_2 \times (Magnitude) + \beta_3 \times (EV)$$
$$+ \beta_4 \times (Risk)|Session$$

where $y$ indicated whether or not the trial was aborted (logistic regression for trial-abortion rates) or the z-normalized lick rate measured from the onset of the first cue to reward onset (linear regression for lick data). The coefficients from this model are shown for trial-abortion rates in Fig. 1j and Table 2 and for licking data in Table 3. Statistical significance of regression coefficients was assessed with two-sided $t$-tests and a criterion for the significance of $P < 0.05$.

**Eye data processing.** We continuously monitored and recorded the animals' eye positions using an infrared eye tracking system at 125 Hz (ETL200; ISCAN) that was placed next to the touchscreen. We calibrated the eye tracker before each test during a fixation task. During recordings, the accuracy of the calibration of the eye tracker was checked and recalibrated if necessary.

**Neuronal data analysis.** We analyzed single-neuron activity by counting neuronal impulses for each neuron on correct trials in fixed time windows relative to different task events focusing on the following non-overlapping task epochs: 500 ms after fixation spot before cues (Fixation), 500 ms after onset of first cue, 500 ms after offset of first cue, 500 ms after onset of second cue, 500 ms after offset of second cue. We used fixed-window and sliding-window linear and multi-linear regression analyses to identify neuronal responses related to specific variables. For fixed-window analyses, we first identified task-related object-evoked responses by comparing activity in the cue and post-cue periods to a baseline control period (before the appearance of fixation spot) using the Wilcoxon test ($P < 0.05$, Bonferroni-corrected for multiple comparisons). We then used multi-linear regression models to test whether neuronal activities were significantly related to specific task variables ($P < 0.05$, t-test on regression coefficient). We also used sliding-window multiple regression analyses with a 200-ms window that we moved in steps of 20 ms across each trial (without pre-selecting task-related responses). Sliding-window analyses tested for dynamic coding of different task-related variables over time within trials and also confirmed that our results did not depend on the pre-selection of task-related responses or the definition of fixed analysis windows. To determine statistical significance of sliding-regression coefficients, we used a permutation-based approach by performing the sliding-window regression 1000 times using trial-shuffled data and determining a false-positive rate by counting the number of consecutive sliding-windows in which a regression was significant with $P < 0.05$. We found that less than five percent of neurons with trial-shuffled data showed more than ten consecutive significant analysis windows. Accordingly, we classified a sliding-window analysis as significant if a neuron showed a significant ($P < 0.05$) effect for more than ten consecutive 20-ms windows. The statistical significance of regression coefficients was determined using t-test; all tests performed were two-sided. Additional population decoding and RSA, described below, examined the independence of our findings from pre-selection of task-related responses and served to assess information about specific task variables contained in the neuronal population.

We performed our regression analysis in the framework of the general linear model (GLM) implemented with the MATLAB function (*glmfit*). Neuronal responses were tested with the following regression models:

GLM 1 (Eq. 1): This GLM served to identify probability-coding neurons and distinguish cue-format specific from abstract, cue-independent probability coding. It also served to derive regression coefficients for Fig. 2c. We adapted a method of classification of

neuronal value responses based on the angle of regression coefficients[55,56]. In our case, this approach identifies probability-coding neurons by testing the statistical significance for a complete model that includes separate regressors for probability cued by fractal and sector stimuli, rather than individual value coefficients, and thus is not biased toward detecting either value code. For the main analysis reported in Fig. 2c, we calculated neuronal activity in two 500-ms fixed windows after the onset of the first cue and for 500 ms following cue offset before the second cue. We used the following GLM:

$$y = \beta_0 + \beta_1(ProbabilityFractal) + \beta_2(ProbabilitySector) + \varepsilon$$

with y as the neuronal activity, *ProbabilityFractal* as reward probability cued by fractal stimuli, and *ProbabilityFractal* as reward probability cued by sector stimuli. Standardized regression coefficients (betas) obtained from this model were used for Fig. 2c and defined as $x_i(s_i/s_y)$, $x_i$ being the raw slope coefficient for regressor $i$, and $s_i$ and $s_y$ the standard deviations of independent variable $i$ and the dependent variable, respectively. Using this method, a neuronal response was categorized as probability-coding if it showed a significant overall model fit ($P < 0.05$, F-test). For responses with significant model fit, we plotted the magnitude of the beta coefficients (standardized slopes) of the two probability regressors on an x-y plane. We followed a previous study[55] and divided the coefficient space into eight equally spaced segments of 45° to categorize neuronal responses based on the polar angle in this space of regression coefficients (Fig. 2c). We categorized responses as coding cue-specific probability if their coefficients fell in the segments pointing toward 0° or 180° (probability on fractal trials) or toward 90° or 270° (probability on sector trials), indicating a relationship to probability for one of the two trial types. We categorized responses as coding abstract, cue-independent probability if their coefficients fell in the segments pointing toward 135° or 315° or in the segments pointing toward 45° or 225°, indicating a relationship to probability on both trial types.

GLM 2 (Eq. 2): This GLM served to identify neurons coding reward probability and/or reward magnitude. It also served to derive partial-$R^2$ values (coefficients of partial determination) for Figs. 3e, 3k, 5b.

$$y = \beta_0 + \beta_1(Probability) + \beta_2(Magnitude) + \varepsilon$$

with y as the neuronal activity in a 200-ms sliding window, aligned to the onset of the first cue and moved in 20-ms steps from 500 ms before the onset of the first cue until 1000 ms after the offset of the second cue, *Probability* the trial-specific probability and *Magnitude* as the trial-specific magnitude.

GLM 3 (Eq. 3): This GLM was identical to GLM2 except that it was calculated on fixed neuronal-response windows rather than on a sliding window. It served to identify neurons coding both probability and magnitude. It also served to derive regression coefficients for Fig. 3g. Classification was based on the angle of regression coefficients as described above[55,56]. For the main analysis reported in Fig. 3g, we calculated neuronal activity in the first, probability-cue period and post-cue period as well as in the second, magnitude-cue period, and post-cue period. We used the following GLM:

$$y = \beta_0 + \beta_1(Probability) + \beta_2(Magnitude) + \varepsilon$$

with y as the neuronal activity, *Probability* as reward probability and *Magnitude* as reward magnitude. (While GLMs 2 and 3 were used to quantify the extent to which probability and magnitude as the most basic variables were encoded by neurons, GLM 4 was used to test whether the more complex integrated variables were also encoded, while controlling for probability and magnitude.) Standardized regression coefficients (betas) obtained from this model were used for Fig. 3g and defined as $x_i(s_i/s_y)$, $x_i$ being the raw slope coefficient for

regressor $i$, and $s_i$ and $s_y$ the standard deviations of independent variable $i$ and the dependent variable, respectively. Using this method, a neuronal response was categorized as coding probability and/or magnitude if it showed a significant overall model fit ($P < 0.05$, F-test). For responses with significant model fit, we plotted the beta coefficients (standardized slopes) of the two regressors on an x-y plane and categorized neuronal responses based on the polar angle in this space of regression coefficients (Fig. 3g). We categorized responses as coding probability only if their coefficients fell in the segments pointing toward 0° or 180°, as coding magnitude only if their coefficients fell in the segment pointing toward 90° or 270°. We categorized responses as coding both probability and magnitude if their coefficients fell in the segments pointing toward 135° or 315° or in the segments pointing toward 45° or 225°, indicating a relationship to both variables.

GLM 4 (Eq. 4): This GLM served to identify neurons coding reward probability, magnitude, expected value or risk. It also served to derive partial-$R^2$ values (coefficients of partial determination) for Fig. 4c.

$$y = \beta_0 + \beta_1(Probability) + \beta_2(Magnitude) + \beta_3(EV) + \beta_4(Risk) + \varepsilon$$

with y as the neuronal activity in a 200-ms sliding window, aligned to the onset of the first cue and moved in 20-ms steps from 500 ms before the onset of the first cue until 1000 ms after the offset of the second cue, *Probability* the trial-specific probability, *Magnitude* as the trial-specific magnitude, *EV* as the trial-specific expected value, and *Risk* as the trial-specific risk. Expected value and risk were defined from the integration of probability and magnitude. Correlations between the regressors in GLM 4 are shown in Table S2.

GLM 5 (Eq. 5): This GLM served to identify neurons coding reward probability, magnitude, expected value, or risk; different from GLM4, the risk was defined dynamically, calculated from probability at the time of the first cue until the onset of the second cue, and calculated from both probability and magnitude from the time of the second cue onward. It also served to derive partial-$R^2$ values (coefficients of partial determination) for Fig. 4e.

$$y = \beta_0 + \beta_1(Probability) + \beta_2(Magnitude) + \beta_3(EV) + \beta_4(Risk) + \varepsilon$$

with all variables as above except for the dynamic risk regressor.

GLM 6 (Eq. 6): This GLM served to identify neurons coding reward probability, magnitude, expected value, the risk from probability or risk from probability and magnitude; different from GLM5, we included two separate regressors for risk from probability and risk from both probability and magnitude so that these variables competed to explain variance in neuronal activity. It also served to derive partial-$R^2$ values (coefficients of partial determination) for Fig. 4f.

$$y = \beta_0 + \beta_1(Probability) + \beta_2(Magnitude) + \beta_3(EV) + \beta_4(RiskProb) + \beta_5(Risk) + \varepsilon$$

with *RiskProb* as risk calculated from probability only and *Risk* as risk calculated from both probability and magnitude.

We performed for each of the 483 neurons a bootstrap analysis with resampling over 1000 iterations and refitting GLM4 for each iteration. We report in Fig. S6 the proportion of significant neurons for a given variable across bootstrap iterations. These histograms show that the bootstrap largely confirms the original proportions of neurons. We also examined whether neurons identified as coding our key correlated variables of interest, magnitude, and risk, in the original analysis were consistently classified in the bootstrap analysis. To do so, we identified neurons that were originally classified as coding magnitude or risk, and then determined the proportion of bootstrap iterations in which this classification was either confirmed or not. For each neuron, we considered a classification as consistent if the probability

of confirmation (across bootstrap iterations) was $p > 0.5$, and a value of 0 if this probability was $p < 0.5$. (We also report numbers for a stricter confirmation threshold of $p > 0.85$.) We then calculate across neurons the proportion of original magnitude and risk neurons that were confirmed or unconfirmed.

**Normalization of population activity.** To normalize activity from different amygdala neurons, we subtracted from the impulse rate in a given task period the mean impulse rate of the pre-fixation control period and divided by the standard deviation of the control period (z-score normalization). We also distinguished neurons that showed positive relationships or negative relationships with a given variable, based on the sign of the regression coefficient, and sign-corrected responses with a negative relationship for plotting population activity. Normalized data were used for Figs. 2g, 3i, 4d, 5c–e, and all decoding and RSA analyses.

**Construction of population expected-value signals.** For the plots in Fig. 5b–e, probability-coding and magnitude-coding neurons were identified by regressing neuronal activity in the first and second cue period on probability and magnitude (Eq. 3). For the plot in Fig. 5c, we then used these neurons to first calculate the mean z-normalized responses across probability-coding neurons to each probability level in the probability-cue period (sign-correcting neurons with negative coding), and correspondingly calculate the mean z-normalized responses across all magnitude-coding neurons to each magnitude level in the magnitude-cue period (sign-correcting neurons with negative coding). Next, we multiplied these probability- and magnitude-level specific responses to calculate hypothetical expected-value signals for each of the 15 probability-magnitude combination defining a specific expected-value level. We then used these 15 'constructed' expected-value activities to calculate activities plotted as a surface in Fig. 5c using interpolation for the oversampled expected-value space shown in Fig. 5c. For the plot in Fig. 5d, we performed linear regression of these constructed (from separate probability and magnitude activities) expected-value signals on the corresponding expected-value levels (separately for signals from neurons with positive coding scheme and for the sign-corrected signals for neurons with negative coding scheme). For Fig. 5e, we first calculated the choice probability that a choice option with a given expected-value level was chosen (using the separately recorded choice-task data, averaging across animals), and regressed these choice probabilities on the neuronal activities corresponding the these expected-value levels.

**Population decoding.** We used nearest-neighbor (NN) and support-vector-machine (SVM) classifiers to quantify the accuracy with which task-related variables were encoded in neuronal population activity in defined task periods, following previous approaches[9,56]. The NN classifier assigned each trial to the group of its nearest single-trial neighbor in a space defined by the distribution of impulse rates for different levels of the grouping variable using the Euclidean distance. The NN classifier can be described as biologically plausible, in the sense that a downstream neuron could perform a similar classification by comparing the input on a given trial, provided by a neuronal population-activity vector, with a stored synaptic-weight vector. The SVM classifier was trained to find a linear hyperplane that best-separated patterns of neuronal population activity defined by a given grouping variable. The different levels of a given grouping variable are referred to as 'groups' in the following text.

We prepared neuronal data for decoding by aggregating z-normalized (to pre-fixation baseline) trial-by-trial impulse rates of the non-simultaneously recorded amygdala neurons from specific task periods into pseudo-populations. We used all recorded neurons that met inclusion criteria for a minimum trial number, without preselecting for coding a specific variable. We only included neurons in

the decoding analyses that had a minimum number of 10 per group for which decoding was performed. We created two $n$ by $m$ matrices with $n$ columns determined by the number of neurons and $m$ rows determined by the number of trials. We defined two matrices, one for each group for which decoding was performed, using the following different groupings. For probability decoding, we grouped data into high ($P = 0.75$) vs. low ($P = 0.25$) probability trials; for magnitude decoding, we grouped data into high (0.9 ml) vs. low (0.1 ml) magnitude trials; for expected value and risk decoding, we grouped trials into high vs. low trials based on terciles of expected value and risk. Accordingly, each cell in a matrix contained the normalized impulse rate from a single neuron on a single trial measured for a given group. In Fig. 6b, we excluded an outlier neuron as its normalized impulse rate exceeded the mean of the other neurons by more than 4 standard deviations. Because neurons were not simultaneously recorded, we randomly matched up trials from different neurons for the same group in the matrix used for decoding, and repeated the decoding analysis with different random trial matching ('within-group trial matching') 500 times for NN and 150 times for SVM. We found in previous studies that these numbers of repetitions produced stable classification results[9,11,12,14,56]. (Our approach likely provides a lower bound for decoding performance because it does not account for potential contributions from cross-correlations between neurons; investigation of cross-correlations would require data from simultaneously recorded neurons.)

We quantified decoding accuracy as the percentage of correctly classified trials, averaged over all decoding analyses for different random within-group trial matchings. We used a leave-one-out cross-validation procedure: a classifier was trained to learn the mapping from impulse rates to groups on all trials except one test trial; this remaining trial was then used for testing the classifier and the procedure was repeated until all trials had been tested. We obtained similar results when splitting data into 80% training trials and 20% test trials. We used a rank-sum test to compare the classification performance against the performance obtained from data in which the group labels were randomly shuffled 1,000 times. We implemented the SVM classifier in Matlab (Version R2018, Mathworks, Natick, MA) with the 'svmtrain' and 'svmclassify' functions using a linear kernel and the default sequential minimal optimization method for finding the hyperplane. We implemented the NN classifier in Matlab with custom code. Statistical significance was determined by comparing vectors of percentage correct decoding accuracy between real data and randomly shuffled data (in which group labels had been shuffled) using the rank-sum test.

For cross-decoding analyses shown in Fig. 6f, h, i, we trained the classifier on data recorded in one particular experimental condition (e.g., probability on fractal trials) and tested the classification performance on data recorded in a different condition (e.g., probability on sector trials).

To investigate how decoding accuracy depended on the number of neurons in the decoding sample in Fig. 6c, d, we randomly selected a given number of neurons at each step (without replacement) and then determined the percentage correct classification. We repeated this procedure 100 for each tested population size. We performed decoding for randomly shuffled data (shuffled group assignment without replacement) with 1000 iterations to test whether decoding on real data differed significantly from chance.

**Representational Similarity Analysis.** We used Representational Similarity Analysis (RSA)[28,94] to examine how activity across the population of recorded neurons in the amygdala and its subdivisions represented task-related variables as quantified by pairwise correlations between condition-specific neuronal population activity vectors. To conduct the RSA analysis, we first calculated, for each recorded neuron, the mean activity related to a specific task event or condition

(e.g., a given probability level cued by fractal stimuli, the same probability level cued by sector stimuli, reward-magnitude levels, etc.). We normalized these condition-specific activities in the same way as for the population decoding analyses described above. For different RSA analyses, we calculated activities in 500-ms fixed time windows (e.g., defined in relation to stimulus presentation). Thus, for a given time window, we calculated the mean activity for a given neuron and condition. This procedure generated a condition-by-neuron matrix for a given time window that we then normalized (by removing the mean and dividing by the standard deviation) and used to calculate pairwise Pearson correlation coefficients between conditions across neurons. These matrices of correlation coefficients between conditions are displayed as color-scaled images in Fig. 7a. Row- and column-ordering of conditions was preserved between all RSA matrix displays.

To interpret the neuronal RSA matrices and evaluate the statistical significance of encoding of particular task-related variables, we generated RSA templates[28] that captured the representational similarity structure related to specific variables (as described in detail below). For statistical analysis, we performed multiple regression using these templates as regressors to explain a given neuronal RSA matrix. To do so, we concatenated all cells of the lower triangular part of the neuronal RSA matrix into a vector and regressed this vector on a regressor matrix defined by the concatenated RSA templates[28]. The statistical significance of coefficients for these RSA regressors was determined using non-parametric permutation tests by shuffling the condition matrix and repeating the regression on the neuronal RSA matrix 10,000 times and then determining the critical t-value corresponding to $P < 0.001$ across the 10,000 shuffled regressions. Similarly, to test whether a particular neuronal RSA coefficient was significantly larger than another coefficient (Fig. 7c), we computed differences in t-values for these regressors based on the shuffled data and determined a critical t-value difference from the shuffled regressions. Analyses were performed using only the unique values from the RSA matrices.

We defined the following RSA templates: an identity matrix to account for the unity correlation between a condition and itself (diagonal of correlation matrix); a cue-format specific matrix that took the value of 1 for condition pairs involving the same probability-cue type (fractals or sectors) and 0 otherwise; a reward-probability matrix that modeled three different mean-centered probability levels and the pairwise similarity between probability levels modeled as the pairwise product of these values, following a previous paper[28]; cue-format specific probability matrices defined by the product of the cue-format matrix and the probability matrix, thus modeling probability similarity only within the same cue type (one matrix each for probability on fractal trials and sector trials); matrices for reward magnitude, expected value, risk from probability, risk from integrated probability and magnitude, each modeling the different mean-centered levels of the respective variable and the pairwise similarity between probability levels modeled as the pairwise product of these values (5 magnitude levels, 15 expected value levels, 9 risk levels). Thus, we used 9 template matrices as regressors in the RSA GLM analyses with the dependent variable consisting of 465 data points (lower triangular part of the across-neuron correlation matrix between 30 ×30 conditions (3 probability levels x 5 magnitude levels x 2 cue sets). The coefficients for these regressors estimated in this way are shown in Fig. 7b, c.

### Reporting summary
Further information on research design is available in the Nature Portfolio Reporting Summary linked to this article.

## Data availability
The data that support the findings of this study are available from the corresponding author upon request. Correspondence and requests for materials should be addressed to fabian.grabenhorst@psy.ox.ac.uk. Source data are provided with this paper.

## Code availability
The custom code that support the findings of this study has been deposited in the Figshare database and can be accessed using the following https://doi.org/10.6084/m9.figshare.28282274.

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

## Acknowledgements

We thank A. David and C. Thompson for animal care, P. Taylor for anesthesia, H. Cousins and A. Stasiak for computer programming, S. Ferrari-Toniolo, F.-Y. Huang, A. Lak, M. O'Neill, and W. Stauffer for support and discussions. We are highly grateful to Wolfram Schultz for the freedom and support to conduct this experiment in his laboratory, and for discussions. This work was funded by the Wellcome Trust and the Royal Society (Wellcome/Royal Society Sir Henry Dale Fellowship grants 206207/Z/17/Z and 206207/Z/17/A to F.G.; Wellcome Trust Principal Research Fellowship and Program Grant 095495 to W. Schultz), the John Fell Oxford University Press Research Fund to F.G., and by the European Union (ERC Starting Grant to R.B.-M., NEUROGROUP, 101041799). Views and opinions expressed are however those of the authors only and do not necessarily reflect those of the European Union or the European Research Council Executive Agency. Neither the European Union nor the granting authority can be held responsible for them. This research was funded in whole, or in part, by the Wellcome Trust. For the purpose of Open Access, the author has applied a CC BY public copyright license to any Author Accepted Manuscript version arising from this submission.

## Author contributions

F.G. initiated the project and conceived the research question. F.G. and R.B.-M. designed and conducted the experiments. F.G. analyzed and interpreted data, and wrote the paper.

## Competing interests

The authors declare no competing interests.
