## [Transparent Peer Review file · Nature Communications]

Dynamic coding and sequential integration of multiple reward attributes by primate amygdala neurons

Corresponding Author: Professor Fabian Grabenhorst

Version 0:

Reviewer comments:

Reviewer #1

(Remarks to the Author)

The authors recorded amygdala neurons while monkeys perform a sequential task with reward-probability and reward-magnitude cues. They examine coding of probability and magnitude, as well as integrated variables, namely expected value and risk. The manuscript nicely demonstrates coding of probability and magnitude as well as coding of the risk of the reward distribution. However, the authors do not control for the part of risk that is only related to the probability signal. Therefore, the manuscript lacks demonstration of an integrated signal that goes beyond function of the individual CSs, a major claim of this paper.

Major concerns:

1. Throughout the manuscript, the authors claim that neurons represent an integrated signal due to coding of the risk signal beyond probability and magnitude. Nonetheless, the authors do not control for a signal that only depends on probability. Importantly, this means that there is no evidence of real integration:
 - a. While the authors show in fig.4e that across neurons risk is independent from magnitude and probability, they do not show that it is independent of the risk that was inferred from the probability signal. Therefore, it is possible that across the population, neurons are sensitive specifically to the uncertainty related to the probability alone and not to the variance that also reflects the magnitude of the reward. One option is to show the results of a model (as in 4e) that considers both the risk from probability alone and the risk from probability and magnitude as separate covariates. If there is coding of risk as variance (that also depends on magnitude), we would expect an increase in partial R^2 even after accounting for probability-related risk. If partial R^2 would only increase for the probability-related risk, this would suggest that the coding is of risk that is only related to the probability.
 - b. How are the magnitude schedule determined? Is there a way to separate risk related to magnitude and probability from risk related to probability (i.e. were their cases of similar risk, with smaller probability and higher magnitude and vice versa)? Can you plot these to demonstrate the combined effect?
 - c. Add risk related to p in fig.7B – across all time points.
 - d. Please show fig.1f for the entire population.
2. The authors claim for a mechanistic difference between risk and expected value coding. The major difference between EV and risk analysis, where in the former the authors partial out both components of the integrated signal – p and m , while in the risk you partial out p and m but not $p(1-p)$ from the integrated signal. Neurons that code p , m and $p(1-p)$ might be sufficient to explain the results in fig.4-5 as well as the risk coding in fig.7. This difference can also emerge from the assumption of their analysis, where they partial out probabilities and magnitude, but do not partial out risk that results from probabilities alone. The idea that p and m are more basic than $p(1-p)$ should be experimentally demonstrated.

Minor concerns:

1. The authors use a scale bar to indicate the timing of the task. This makes it hard to compare temporally specific activity. Please add ticks and numbers relative to cue onset in all temporally aligned figures.
2. Fig.2d, fig.3h, etc.: showing this kind of diagram that is not proportional to the size of the group is misleading and should be avoided. If you use this diagram, areas should be proportional to probabilities. If this cannot be done, avoid using such representation.

3. In the left part of fig.4d the population activity is plotted for the different probabilities. Similar population activity is plotted in fig.2g, separated for the fractal and sectors. The activity does not seem in alignment between these two figures.
4. Fig.1h – right choices always reflect safe? If so, change the wording to safe-risky, the left-right is confusing
5. Fig.3h- this is circular as the subjective values are modeled by choice probabilities and then correlated with choice probabilities.
6. Line 324 – 6% - is this above chance level?
7. The authors describe two methods for neuron selection, based on a fixed or running window. It is not clear which method was used for different presentations (e.g. fig.1f)
8. To demonstrate the integrated signal in the examples, the authors should show some consistency of the integrated signal across levels of a single signal. For example, show fig5A for low and high expected values within the levels of probabilities (plotting high, mid and low EV only in high probability, only in medium probability and only in low probabilities). Same for the different examples and the population level in 4d.
9. Fig. 6f,h,i – what is the gray line?
10. The authors only examine expected value of the integrated signal, but expected value also exist when the first signal appears (with the expected value of the second signal can be calculated). It would be interesting to see such expected value coding.

Reviewer #2

(Remarks to the Author)

Dynamic coding and sequential integration of multiple reward attributes by primate amygdala neurons
 Grabenhorst and Báez-Mendoza

This paper reports interesting new findings on the primate amygdala in experiments that have been carefully performed and analysed.

I have the following comments to help the authors improve the paper.

1. The use of a Pavlovian task (in which the animal can make no choices) is a bit of a surprise, for it could be that the animal may not need to integrate the variables (reward probability and magnitude) as well as if an instrumental choice had to be made between what was on offer and a comparison reward.
 The authors should explain the pros and cons of their choice of a Pavlovian task, and consider this possible limitation.
2. The investigators did run a choice task on some days. Were any recordings of neuronal activity made in the instrumental task? If so, they should be reported in this paper for comparison, and to address the point in (1).
3. Some type of effect size should be reported. The beta in a GLM does not provide an effect size that can be used as a scale-free metric. To help the authors appreciate the point being made, a Cohen's d measure, or a mutual information measure, could provide a scale-free metric.
4. It is somewhat surprising that relatively few neurons encoded expected value, which is presumably what would be useful to the animal at least in an instrumental task. The authors should comment on why they think relatively few amygdala neurons get this far; and on how neurons in the orbitofrontal cortex may compare with these amygdala neurons.
5. The latency of some of these neuronal responses in the amygdala would be of interest, for they might enable assessment of where the inputs that drive these neurons come from.
6. In that context, the authors should consider more how these amygdala neurons' responses compare to those in the orbitofrontal cortex. Could some of these amygdala neurons' responses reflect what is represented in the orbitofrontal cortex?
7. In recordings made from primate amygdala neurons, it has been reported that the responses of the neurons tend to be much more idiosyncratic in how they represent expected value than for the orbitofrontal cortex. Do the authors have any thoughts that are relevant to this issue, and whether the responses of orbitofrontal cortex neurons would be more useful in guiding behaviour because they are more closely related to expected value, which is what some previous investigators have reported?

Reviewer #3

(Remarks to the Author)

The manuscript provides an important demonstration of the encoding of diverse reward attributes in the amygdala, which combined together carry information relative to expected value and risk, likely guiding economical choices. Importantly, neuronal activity discriminated for levels of probability for all cues predicting specific probability, indicating that amygdala neurons generalized across stimuli and encoded probability in an abstract format. Further, authors observed that the same neurones also encoded reward magnitude, which when combined with probability also carried information about the risk associated to the current trial. One of the manuscript's important finding is that these signals are present differentially within the amygdala nuclei: only lateral amygdala supports reward probability and magnitude signals transferrable over time

periods.

The paper provides a very strong and set of analysis to support the conclusions, and rests on a very robust set of data. I have a few points for authors to clarify especially with respect to the concepts of variance, risk and uncertainty.

1) the definition of risk in the intro and throughout the manuscript is unclear.

page 3 line 100, risk is defined as uncertainty (and in the figure 1b), but then the equation in figure 1a defines risk as the variance (probability factoring magnitude and the expected value). The latter takes into account the potential loss (risk is maximal at higher gain with lower probability), while the risk in figure 1b only computes uncertainty (max at $p(\text{reward}) = 0.5$, lower at $p(\text{reward}) = 0.75$ and 0.25). However, variance which is a term that is coined later to describe (line 264) does not really convey the risk computation which takes into account potential reward loss (at same uncertainty of .25 or .75).

The authors should clarify their definition, and maybe compare clarify that "risk" is better than uncertainty alone.

The mix between the different terms (variance in distribution of reward, risk and uncertainty) is confusing.

2) Can authors specify if (reward) = 0.5 were excluded from analysis in Figure 1g, especially since example in fig 1e involved $p(\text{rew})$ of .5.

I also do not understand how the results in Fig 1i (animals prefer the safe reward over the gamble) fit with the risk seeking results of animal B (who prefers options with higher risk in fig 1g).

Can authors define constant on Fig 1j and 1g?

3) The distribution of individual cells in the space is nice, but, the light and blue green are very difficult to differentiate visually on fig 2C and fig3g. Please draw boundaries or change colors.

4) Could the authors measure the latency of the probability or magnitude signal across nuclei?

Figure 1c shows that there are cells that increase activity for lower p (negative beta). Do these cells have the same latency, and how do authors interpret these cells? Are these more likely to be sensitive to risk?

Actually, this seems to be the case, because figure 4d show that higher activity to low p , followed by higher activity for higher risk.

This may have an implication for what is computed by the cells. Maybe authors could comment on this?

5) the sentence line 212, p7: These responses are conceptually important because they could reflect value irrespective of the specific reward parameter and provide a single-neuron basis for integrating probability and magnitude into expected value or risk.

If the cells signal value irrespective of the specific reward, then they already signal expected value. Please clarify the sentence. I suppose that authors mean that the graded signals could provide a single neuron basis to integrate magnitude and probabilities over time into expected value and risk.

6) Line 283: it says that some neurons responded "differently to the highest risk indicated by probability of $P = 0.5$, compared to low risk indicated by $P = 0.25$ and $P = 0.75$ "

I think (cf point 1) that this is uncertainty (highest at .5) , and not risk.

Within any reward magnitude, a p of 0.25 is morerisky than a p of 0.5 according to equation in fig 1.

Next, according to all reward magnitudes and probabilities, isn't risk a continuum of numbers ranging from lowest risk Magnitude of 0.1 with a $p(\text{reward})$ of 0.75 to highest (0.9 at 0.25). How did authors regroup risks into 3 categories ins't clear from main text.

7) which neurons exactly are used to make figure 5b, c, d?

Is it the 69 neurons described earlier that encode probability followed by magnitude?

Can the authors specify the conditions, monkeys and neurons used for figure 5e?

8) I wonder what the decoding on figure 6h implies for decoding of cells in which training was made on probability cues (low vs high), and tests were done on magnitude cues. Were magnitudes also regrouped into two (high and low). Does this imply a one to one relationship between probability and magnitudes such that the state of the population for high probability was the same as for the high magnitude? Please explain how the decoding generalizes across these two periods?

9) The figure 7c, I am not sure whether the colors for BLA and LA were inverted, such that BL should be black! also the figure says magnitude while the text says probability (7c left).

10) This may be the scope of a different paper, but how did amygdala neurons encode reward presence or absence of reward at the time of reward delivery, especially for neurons encoding reward probability?

This may just be interesting to understand how outcome is monitored with respect to probability integration. I understand that animals are over-trained and already known the probabilities, but I was just curious if absence of reward, was signaled differentially given high or low uncertainty.

(Remarks to the Author)

This manuscript by Grabenhorst and Báez-Mendoza describes the encoding properties of neurons in the macaque amygdala in a Pavlovian reward association task. In every trial the monkeys are shown a sequence of visual cues that indicate the probability and magnitude of an upcoming fluid reward. Reward probability refers to the likelihood of receiving the indicated reward magnitude (range 0.1 to 0.9ml) vs. receiving a small nominal reward of 0.05ml. This scheme permits each trial to be parameterized according to reward probability, reward magnitude, the expected value (EV, product of probability and magnitude), and risk, defined as the variance of the distribution of expected outcomes of a given probability-magnitude bundle. Two distinct sets of probability cues were used (fractals and pie-like "sector" stimuli). In the main Pavlovian task, no behavioral measures were analyzed other than trial abort rates; in a separate choice task using the same stimuli, the monkeys' choices indicated that they assigned appropriate EVs to the bundles offered on each trial. The main claims of the paper are: 1) Single amygdala neurons encoded the reward probability and magnitude in serial fashion, consistent with the presentation order of the two cues. 2) Single amygdala neurons usually encoded reward probabilities with a consistent code for the two stimulus sets. 3) Many neurons encoded the risk associated with a given magnitude-probability bundle, whereas very few encoded the EV. However, an accurate EV signal could be reconstructed from the separate probability and magnitude signals encoded at time of the respective cue presentations. 4) The results of pseudopopulation-based decoding analyses and RSA template analyses generally corroborated the single-neurons results, with some differences evident in neurons in the LA and BL subnuclei.

At a technical level, this is a very well-executed study. I am confident in the results, and my comments (see below) are small-to-modest and mostly asking for clarification of technical points. If I were to identify any major shortcoming in this study, it would be that the behavioral significance of the amygdala coding properties is unclear. Because a Pavlovian task was used, and because no Pavlovian responses were recorded (e.g. pupil size or licking), it is not possible to draw strong mechanistic conclusions about how the activity of the amygdala neurons might reflect or contribute to behavior. Therefore, in my opinion this study represents a very solid, but modest, advance in our knowledge of primate amygdala function.

Interpretation comments:

- Fig 5 suggests that by multiplying the probability and magnitude signals recorded at different times, one can recover something that looks like an EV signal. Do the authors think that these two separate signals are integrated downstream of amygdala? If so, where?

- On a related note: In my opinion the statement starting at line 541 seems to be a stretch. First, there is no reward-related behavior measured in this task, so it's difficult to make this conclusion. Second, I think it's highly likely that the monkeys are aware of the EV of the bundles (as indicated by the choice task), but that this variable is just not reflected in single amygdala neurons.

Technical comments:

- It is unclear what distinguishes GLM2 and GLM3, as the equations are the same. Is the difference due to the time windows in which these GLMs were analyzed?

- GLM2 and GLM3 seem to be nested within GLM4. Why not just fit GLM4 and use the coefficients for all analyses? If there's a concern about overfitting by using GLM4, then it might make sense to try stepwise models or some other model selection procedure.

- How many total template matrices are used in the regression analysis? If I read the methods correctly, it seems there could be a large number of matrices needed to capture the combinations of different variables. This could lead to overfitting, which could point to the need for some sort of model selection or regularization procedure.

- In addition to the variables already analyzed (probability, magnitude, etc.) one potential variable that could be encoded is belief confirmation (for details see Hunt et al. Nat Neuro, 2018). In this task, a belief confirmation code would be evident at the magnitude cue, and would signal whether the value of the magnitude cue is congruent with (i.e. confirms) the value of the preceding probability cue.

- A few additional details on the choice task would be welcome. For example, was the "safe" option always the same magnitude in every trial? Or did it vary across trials? Was there an additional fractal or sector cue that indicated $p=1$ for the safe option on each trial?

- The neural screening procedure should be mentioned in the results section, i.e. that the 483 neurons analyzed were out of ~1000 screened for task responses online.

- Pertaining to Figs. 2c,d and related text: The results state that among 148 probability-coding neurons, 84 code probability for both cue formats. But Fig. 2c suggests that a minority of these 84 have opposite encoding signs, e.g. positive for fractals and negative for sectors. Intuitively, it seems that these few opposite-sign neurons do not code for probability in a 'cue-unspecific' format, and should be counted separately from the neurons with significant same-sign encoding.

- Fig. 1b shows the intuitive relationship between probability-based risk and probability alone. It would be helpful to also see an illustration of how risk relates to probability, magnitude, and EV when risk is computed using both probability and magnitude. These relationships are less intuitive because it involves different magnitudes, and because, in this task, the

“non-rewarded” outcomes occurring in the 1-P portion of trials actually involve delivery of a small reward.

- On a related note it would be nice to see in the supplements a table of correlations between regressors in GLM4

- Line 271: Should this read “eq. 4” instead of “eq. 3”?

- A little more clarity on the operation shown in Fig. 5c would be helpful, e.g. in a brief paragraph in the Methods.

- The methods describe both nearest-neighbor and SVM classifiers. Which ones were used for which results?

- I sense a small contradiction between the results of Fig. 6a, showing strong decoding of EV, and the claim that encoding of EV in single neurons was infrequent (6%). How can the strong decoding of EV be explained if single neurons don't encode it? E.g. is it subthreshold encoding distributed over many neurons? Or do the neurons that encode EV do so with high signal to noise?

- In Fig. 6b at trial epochs 3 and 4, there is below-chance classification accuracy for probability. This is surprising, especially for epoch 3, which is presumably before even the magnitude cue is shown. What explains this effect?

- Fig. S3 legend: Should the legend for panel (e) read “Distribution of *risk*-decoding accuracy? . . .”

- In Figure 6F, the low performance of the fractal-vs-sector cross decoding is surprising, given that the high correlation shown in Fig. 2C, i.e. given that single neurons tend to encode fractal and sector cues in the same way. Is there any explanation for this apparent contradiction? This bears on the conclusion stated in line 478-79, i.e. that probability coding is “abstract” and generalized across formats.

- Pertaining to Fig. 6H and 6I: what encoding scheme would result in high cross-classification in one direction but not the other? My intuition, perhaps incorrect, is that cross-classification accuracy should be similar regardless of which stimulus is used for training/testing. Is this perhaps related to the method used (nearest neighbor vs. SVM)? Some clarification on this point would be welcome.

Version 1:

Reviewer comments:

Reviewer #1

(Remarks to the Author)

The authors performed thorough revisions and answered my concerns. Although I am not fully convinced that they can demonstrate conclusively risk coding (rather than integration of probability/uncertainty/magnitude), I notice that they provided most analyses that, under the constraints of the behavioral paradigm, can supply reasonable supporting evidence. Putting risk aside, the behavioral paradigm is interesting and novel, and the results are carefully analyzed and presented. I therefore support publication as-is.

Reviewer #2

(Remarks to the Author)

NCOMMS-24-34111A

Dynamic coding and sequential integration of multiple reward attributes by primate amygdala neurons
Grabenhorst and Báez-Mendoza

I have been through the Responses to all 4 reviewers of the paper, and the revision of this paper, and believe that the authors have made appropriate revisions to this paper for it to be accepted in principle for publication in Nature Communications without further delays caused by sending it out again for further peer review.

The analyses now included in the paper are very detailed, and carefully done.

It is possible that with all this detail, the main conclusions of the paper may not be sufficiently easily assimilated by a reader of the paper. The main conclusion of the paper, it appears, is that in this Pavlovian task providing for separate and combined analysis of the effect of risk in decision-making as influenced by reward probability and magnitude, few amygdala neurons actually encoded the expected outcome, though expected outcome was reflected if the diverse types of neuronal response found in the amygdala to these variables could somehow be combined. The possibility remains that some other brain region, perhaps the orbitofrontal cortex, does represent at the neuronal level expected value as influenced by reward probability and magnitude. In addition, it is worth noting that the Pavlovian task may require less integration of these variables than an instrumental task might.

What I recommend is that the authors might be encouraged to draw out these points somewhere in the paper rather clearly and briefly, with the editors of Nature Communications checking that such an overall summary is easily available to help readers.

Overall, I recommend acceptance in principle, with no further delay caused by further peer review, but possibly some clarification along the lines suggested and assessed by the editors of the Journal.

Reviewer #3

(Remarks to the Author)

I thank the authors for their clarification and the additional analysis which in my opinion makes the paper almost ready for publication.

One last point, relative to their definition of risk and uncertainty early on.

Authors replied that they consider risk throughout the manuscript as “the variance of the statistical distribution of possible outcomes (both positive and negative)”. They also say line 104 “we adopted the frequently considered measure of economic risk as variance²⁹⁻³²: risk defined in this way increased from $P = 0$ to $P = 0.5$ and decreased from $P = 0.5$ to $P = 1$, following an ‘inverted-U’ function ».

However, in the equation presented in figure 1A, var is represented as the sum[$p(\text{magnitude}) \times \text{squared}(\text{magnitude} - \text{expected value})$] which, in effect, takes the expected value as the product between magnitude factoring the probability of magnitude. Doesn't then, this variance, change as a function of expected value, and isn't it higher for low probability of high reward magnitude, and a lower for high probability of high reward?

Then, this is more like risk than just variance?

Thus the risk computed in the integration equation (also labelled as such in the title of figure 1a), is different from the risk computed in figure 1b, the latter being just uncertainty.

Throughout the manuscript, risk is then derived either i) from probability only (then it is uncertainty) or ii) from the integration of probability and magnitude -then call it risk or variance, but it should be for clarity defined differently from risk in figure 1b (black line), which is simply uncertainty.

Then line 289, they say “A substantial number of 289 amygdala neurons (99/483, 20%; eq. 4) carried signals that reflected the risk of expected rewards, defined by the variance of the current-trial statistical reward distribution derived from the integration of probability and magnitude information (cf. Fig. 1a, b; Fig. S3).” Given that there is a reference to two distinct definition of risk, this is confusing, as equation 4 I think refers to 1a, and not 1b.

Terms RiskProb and Risk are clear but only defined line 1060.

I think it would be easier if the same terms would be used throughout the figures and text, and in figure 1A and 1B, they are not clearly labelled early on to avoid confusion;

Reviewer #4

(Remarks to the Author)

My thanks to the authors for their response. While many of the changes and responses have satisfied my concerns, there are still some remaining issues as well as new issues raised by the additional material in the revision. In particular, my original review pointed out some unusual results in the decoding analyses (e.g. below-chance decoding of a variable not yet known to the monkey), but the response to these issues was inadequate. Also, the inclusion of licking results is good, but the actual results themselves raise more questions, and do not provide any new information about the behavioral significance of the neural coding properties documented in the study.

On the whole my opinion of this study remains the same. It is a technically solid effort (with the possible exception of the decoding analyses) that makes a modest advance in our knowledge of primate amygdala function. While the results are compatible with the idea that the amygdala contributes to choices under uncertainty, they do not provide any insight into the mechanisms for this contribution.

1) The analysis of Pavlovian licking behavior in the task is welcome. However, the specific result reported raises new questions. Both monkeys appeared to lick *less* as a function of the anticipated reward magnitude, as indicated by the negative regression estimates for Magnitude in Table 3. Estimates for probability were also negative, though non-significant for both animals. The estimates were obtained using a multiple regression, so one interpretation is that while holding all else equal (i.e., risk and probability) the monkey's overt anticipation of a reward is smallest for higher reward sizes. This is counterintuitive, and contradictory to several prior studies (e.g. Morrison & Salman 2009, Saez & Salzman 2015, McGinty & Newsome 2016). At a minimum, I think it appropriate to comment on this unusual result, and to show these data in the supplements.

2) Related to the above, even if the licking results were more similar to prior studies, their being included in the study does not do much to show how the neural results in the study contribute to behavior. To do that would require additional analyses to explicitly link the behavioral and neural results, e.g., to link licking behavior to the encoding of risk, magnitude, etc. Because of this, I'm still of the opinion that the behavioral significance of the amygdala coding properties is not clear.

3) My thanks for reporting the correlation between the behavioral variables in the new Figure S3 and new table S1. The authors are clearly aware that the high correlations between the variables (e.g., $r=0.943$ for risk and magnitude) could make it difficult to tease apart the unique contribution of each variable to explaining variance in neural activity. The use of partial R^2 (e.g. in Figure 4e) is very appropriate here, and is one way to make the case that risk is a strongly encoded variable over and above magnitude and probability. Nonetheless, when two dependent variables are highly correlated, small, random fluctuations in the data can influence the relative portion of variance attributable to each variable. To quantify this effect in this study, I would like to recommend some sort of repeat reliability analysis. For example, one could split the data into two counterbalanced sets of trials, use GLM4 to identify cells that significantly encode each of the relevant variables in each half of the data, and then construct a confusion matrix showing the consistency of these classifications between the two halves.

4) In the first review I had three comments on the decoding results: Below-chance encoding in Figure 6b, the low cross-condition probability decoding in Figure 6F (despite the very consistent cross-condition coding of probability in Figure 2C), and the asymmetric cross-condition decoding of probability and magnitude. On the whole, I found the response to these issues inadequate. The responses seemed to acknowledge the somewhat unusual results, but only offered vague explanations for them (e.g. outlier cells, or the “complexity” of the population code). At a bare minimum, I think it would be wise to confirm that these results are robust to outliers and to the assumptions of the nearest-neighbor classifier that was used. SVM, which the authors say has already been performed, would probably do the trick.

Beyond confirming the results, a more satisfying response would be to explain the results in terms of concrete features of the underlying data – perhaps corresponding to important features of information coding in the populations. For example, the response regarding Fig. 6b mentioned that outlier data might be responsible for the below-chance decoding, but demurred at the idea of removing these outliers and replacing the figure. This seems like an easy fix, and it’s unclear why it wasn’t done.

In my opinion, one of the strengths of this paper is that the analyses appear well reasoned and technically sound. Not fully investigating these unusual decoding results would undermine the technical merits of the paper.

Version 2:

Reviewer comments:

Reviewer #3

(Remarks to the Author)

The authors have taken all my concerns into account and addressed them appropriately.
The paper will make a great contribution to the field.

Reviewer #4

(Remarks to the Author)

My thanks to the authors for their updates to the manuscript. I have only a few suggested corrections, as well as a general comment.

Licking analysis: Thank you for providing these additional details, in particular the graphs, which help clarify the potentially confusing regression estimates shown in the table (e.g. the negative estimates for magnitude and probability).

Bootstrapping analysis: This new analysis does not fully address my original concern. The original concern was that the variables are highly correlated, making it difficult to consistently designate a neuron as encoding one variable but not the other. This question is made more complicated by the possibility of finding cells that encode two correlated variables (e.g. magnitude and risk), which is evident in the new analysis but was not addressed in the original study.

In contrast, this new analysis mostly addresses the question of sensitivity for detecting significant effects for a single variable, rather than the potential for confusion between them. For example, it’s encouraging that for “risk” neurons there are no cells in the upper right quadrant of the table; but the table entries are thresholded by requiring >500 bootstrapped samples to be significant. A more accurate measure would be to compute the fraction of samples in which significance is detected. Also, Fig. S6 does not separately consider cells that encode Magnitude alone, Risk alone, or both at once, making it difficult to assess the degree of confusion between these two variables.

I’d like to suggest a variant of this new analysis that addresses my concerns: For each cell, determine whether it encodes variable A, variable B, or both, using the original sample of trials. Then, for each cell do the same analysis over 1000 bootstrapped samples, and quantify the fraction of samples for which the cell encodes A, B, and both. Then, among the “original A” cells (cells encoding A in the original analysis), find the mean fraction of bootstrap samples that encoded A, B and both. The expectation is that among “original A” cells, the largest fraction of bootstrap samples should encode A, with fewer encoding B or both. Likewise, among “original B” cells, the largest fraction of samples should encode B, etc. The key outcome here will be how likely it is for an “original A” cell to be mis-classified as a B-cell in the resamples, and for “original B” to be classified as A in the resamples.

Decoding analysis: Thank you for providing the SVM results as a supplement. I could not find the outlier removal procedure

in the methods. Please ensure that this is included in the final version.

General comment: I'd like to offer a more in-depth explanation of my assessment of the mechanistic advance. The main conclusion of the study is that the amygdala encodes the variables that appeared to drive behavior (probability, magnitude, EV, and risk). This makes it possible that the amygdala is involved in the Pavlovian behaviors in this task, and may be involved in other behaviors that depend on these variables. This "existence proof" is an important first step in establishing the mechanisms by which amygdala might contribute to decisions under risk and other behaviors.

However, in my opinion results like this one must be treated with caution until there is additional evidence that links neural coding to behavior. Examples are the study of Stoll & Rudebeck (2024), which shows that within-session changes in juice preference are linked to within-session changes in neural coding; and the study of Ballesta and Padoa-Schioppa, which uses causal methods to show the involvement of OFC in choice.

Without additional evidence, it is quite possible that the amygdala is not involved in the behavior shown here. Many other brain regions encode probability, magnitude, etc., and it's feasible that the signals in these other regions are more important than amygdala for driving behavior. One might argue that the information encoded in amygdala must have some non-trivial influence on downstream circuits and must ultimately affect behavior in some way. But a counterargument which I have recently embraced is the idea that a surprising fraction of neural information coding is in the "null space", defined as activity patterns that do not exert a net influence on downstream circuitry and behavior (for a review of see Churchland & Shenoy, Nat Rev Neuro, 2024). In other words, in any given task it is possible that the information encoded in a given brain region is not directly contributing to the observed behavior.

I grant that this position may make me an outlier, but for these reasons I am circumspect when drawing conclusions about behavioral mechanisms based on results showing only that information is being encoded in a neural population, as is the case here.

REVIEWER COMMENTS

**Reviewer #1 (Remarks to the Author):**

*The authors recorded amygdala neurons while monkeys perform a sequential task with*
*reward-probability and reward-magnitude cues. They examine coding of probability and*
*magnitude, as well as integrated variables, namely expected value and risk. The manuscript*
*nicely demonstrates coding of probability and magnitude as well as coding of the risk of the*
*reward distribution. However, the authors do not control for the part of risk that is only related*
*to the probability signal. Therefore, the manuscript lacks demonstration of an integrated*
*signal that goes beyond function of the individual CSs, a major claim of this paper.*

**Authors' response:** We thank the Reviewer for this positive evaluation of our paper and for
the comments regarding the differentiation of risk from different sources. By addressing
these points, as described in detail below, the paper has been much improved.

*Major concerns:*

*1. Throughout the manuscript, the authors claim that neurons represent an integrated signal*
*due to coding of the risk signal beyond probability and magnitude. Nonetheless, the authors*
*do not control for a signal that only depends on probability. Importantly, this means that there*
*is no evidence of real integration:*

*a. While the authors show in fig.4e that across neurons risk is independent from magnitude*
*and probability, they do not show that it is independent of the risk that was inferred from the*
*probability signal. Therefore, it is possible that across the population, neurons are sensitive*
*specifically to the uncertainty related to the probability alone and not to the variance that also*
*reflects the magnitude of the reward. One option is to show the results of a model (as in 4e)*
*that considers both the risk from probability alone and the risk from probability and*
*magnitude as separate covariates. If there is coding of risk as variance (that also depends*
*on magnitude), we would expect an increase in partial R^2 even after accounting for*
*probability-related risk. If partial R^2 would only increase for the probability-related risk, this*
*would suggest that the coding is of risk that is only related to the probability.*

**Authors' response:** Thank you for raising this important point. We directly followed this
suggestion and now include a GLM in which risk from probability only and risk from both
probability and magnitude compete to explain variance in neuronal activity. We provide the
number of risk neurons from either form of risk in the Results and include a version of Fig. 4e
with partial R^2 s for both regressors. The results support our conclusion that amygdala
neurons encode risk from integrated probability and magnitude.

Lines 314-321: “In a further model, we included two variants of the risk regressor to model risk derived solely from probability and risk derived from the integration of probability and magnitude (in addition to probability, magnitude and expected value). This model identified 103 risk neurons (21%) for risk derived from the integration of probability and magnitude and 65 risk neurons (13%) for risk derived from probability; 27 of these neurons were significant for both regressors. The timecourses of these two types of risk signals showed that risk from probability was coded prominently during the first-cue period but also during the second-cue period, whereas risk from probability and magnitude was coded only during the second-cue period (Fig. 4f).”

“(f) Partial R^2 from a sliding-window multiple regression analysis across all recorded neurons with a separate regressors for risk from probability and risk from integrated probability and magnitude.”

Lines 1050-1059: “GLM 6 (Eq. 6): This GLM served to identify neurons coding reward probability, magnitude, expected value, risk from probability or risk from probability and magnitude; different from GLM5, we included two separate regressors for risk from probability and risk from both probability and magnitude, so that these variables competed to explain variance in neuronal activity. It also served to derive partial- R^2 values (coefficients of partial determination) for Fig. 4f.”

$y = \beta_0 + \beta_1 (Probability) + \beta_2 (Magnitude) + \beta_3 (EV) + \beta_4 (RiskProb) + \beta_5 (Risk) + \varepsilon$

with *RiskProb* as risk calculated from probability only and *Risk* as risk calculated from both
probability and magnitude.”

*b. How are the magnitude schedule determined? Is there a way to separate risk related to*
*magnitude and probability from risk related to probability (i.e. were their cases of similar risk,*
*with smaller probability and higher magnitude and vice versa)? Can you plot these to*
*demonstrate the combined effect?*

**Authors' response:** We designed the probability and magnitude combinations so that we
sampled probabilities avoiding previously demonstrated probability distortions by monkeys
and covered an evenly spaced range of magnitudes; unfortunately, this did not produce
conditions where risk levels were exactly matched under different probability-magnitude
combinations. Producing plots for conditions with similar but not equal risks and different
probability-magnitude combinations would necessarily show some differences in neuronal
responses and it would not be possible to demonstrate absence of response difference that
is not explained by residual risk difference. Nevertheless, we emphasise that the most direct
statistical test of the Reviewer's point relating to combinations of probability and magnitude
producing specific risk levels is our GLM regression approach that tests for quantitative
relationships between neuronal responses and our key variables probability, magnitude, risk
(from probability, and from both probability and magnitude). Please also see our response to
points 2 and the newly included Figs S4 and S5, which plot risk and expected value signals
of our example neurons for all 15 experimental conditions.

**“Fig. S4. Amygdala neuron coding risk. (a)** Activity of the neuron in Fig. 4a, shown
 separately for different probability and magnitude levels. **(b-d)** Activity of the neurons in Fig.
 4i-k, shown separately for different probability and magnitude levels.”

 **“Fig. S5. Amygdala neuron coding expected value.** Activity of the neuron in Fig. 5a,
 shown separately for different probability and magnitude levels.”

*c. Add risk related to p in fig.7B – across all time points.*

 **Authors’ response:** We have added the risk-from-probability regressor to this model. The
 results show that (1) risk was significantly represented in all periods, (2) risk was initially
 represented as risk from probability followed by risk from both probability and magnitude (3)
 integrated risk in later periods was not explained in terms of risk only from probability.

 **“Figure 7. Structure of population codes in amygdala and its subdivisions assessed**
 **with Representational Similarity Analysis (RSA).”**

*d. Please show fig. 1f for the entire population.*

**Authors' response:** We assume the Reviewer was referring to Fig 4f (risk from probability
and risk from probability and magnitude) and we now show the data as suggested in new
panel Fig. 4f (see figure reproduced above) while keeping as a separate display the signal
for this risk-coding subpopulation of amygdala neurons (now Fig. 4g).

*2. The authors claim for a mechanistic difference between risk and expected value coding.*
*The major difference between EV and risk analysis, where in the former the authors partial*
*out both components of the integrated signal – p and m , while in the risk you partial out p*
*and m but not $p(1-p)$ from the integrated signal. Neurons that code p , m and $p(1-p)$ might be*
*sufficient to explain the results in fig.4-5 as well as the risk coding in fig.7.*
*This difference can also emerge from the assumption of their analysis, where they partial out*
*probabilities and magnitude, but do not partial out risk that results from probabilities alone.*
*The idea that p and m are more basic than $p(1-p)$ should be experimentally demonstrated.*

**Authors' response:** We thank the reviewer for raising the point about differentiation of
integrated risk and expected value from their basic components probability, magnitude and
risk-from-probability. We have now addressed this point by including the results of new GLM
and RSA analyses that specifically included all these variables in the same model, so that
the variables competed to explain variance in neural responses. These results show that,
while all variables were significantly encoded at some point, coding of integrated risk (as well
as probability and magnitude) was robust in this stricter model (while expected value
remained relatively weak). We now include the following new text in Results and Discussion
to address this point, in addition to new figure panels Fig. 4f, Fig.7b, Fig. S3, Fig. S4.

Lines 314-326: "In a further model, we included two variants of the risk regressor to model
risk derived solely from probability and risk derived from the integration of probability and
magnitude (in addition to probability, magnitude and expected value). This model identified
103 risk neurons (21%) for risk derived from the integration of probability and magnitude and
65 risk neurons (13%) for risk derived from probability; 27 of these neurons were significant
for both regressors. The timecourses of these two types of risk signals showed that risk from
probability was coded prominently during the first-cue period but also during the second-cue
period, whereas risk from probability and magnitude was coded only during the second-cue
period (**Fig. 4f**). Importantly, a subset of 33 individual amygdala neurons showed a
particularly unambiguous risk-coding pattern (**Fig. 4g**): these neurons showed risk coding
both at the first, probability cue (coding risk derived from probability only, by responding
differently to the highest risk indicated by probability of $P = 0.5$, compared to low risk
indicated by $P = 0.25$ and $P = 0.75$), followed by a risk signal at the second, magnitude cue
(coding risk derived from both probability and magnitude, according to eq. 1)."

Lines 554-556: "Importantly, neurons were sensitive to both risk derived from probability
along and to risk that derived from the explicit integration of probability and magnitude
(including when both variables competed in the same regression)."

*Minor concerns:*
*1. The authors use a scale bar to indicate the timing of the task. This makes it hard to*
*compare temporally specific activity. Please add xticks and numbers relative to cue onset in*
*all temporally aligned figures.*

**Authors' response:** Thank you for this suggestion. We have now included the information
about the latencies of population signals in the text for additional clarity. While we have kept
the current display style with the scale bar to maintain consistency with our previous

publications, we are happy to modify the figures to include xticks and numbers relative to
cue onset if the reviewer felt this was essential.

*2. Fig.2d, fig.3h, etc.: showing this kind of diagram that is not proportional to the size of the*
*group is misleading and should be avoided. If you use this diagram, areas should be*
*proportional to probabilities. If this cannot be done, avoid using such representation.*

**Authors' response:** Thank you for raising this issue. We prefer to keep these plots as they
provide an intuitive summary of the subsets of data and are commonly used by us and other
groups (e.g., Grabenhorst et al., 2019 Cell; Tyree et al., 2024, Science). We understand the
Reviewer's concern and therefore include (in addition to stating the numbers and
percentages in the figure) in each figure legend a note that the size of the display is not
proportional to group size.

*3. In the left part of fig.4d the population activity is plotted for the different probabilities.*
*Similar population activity is plotted in fig.2g, separated for the fractal and sectors. The*
*activity does not seem in alignment between these two figures.*

**Authors' response:** Thank you for pointing this out. The probability-related activity in Fig.
4d is sorted for each neuron based on the sign with which the neuron encoded risk in the
second cue period (i.e., positive risk-coding neurons were also treated as positive
probability-coding neurons). Thus, the data show there was no clear relation in the
population signal between probability coding and subsequent risk coding. We now clarify this
in text and figure legend:

Lines 300-302: "There was no clear relation in the population signal between probability
coding in the first cue period and subsequent risk coding when referencing these signals
based on each neuron's positive vs. negative risk-coding scheme (**Fig. 4d**, left and right
panels)."

*4. Fig.1h – right choices always reflect safe? If so, change the wording to safe-risky, the left-*
*right is confusing*

**Authors' response:** We assume the Reviewer refers to Fig. 1i and thank them for pointing
this out. We have relabelled the axis as P(Safe). (Labels in Fig. 1h are correct as they
stand.)

*5. Fig.3h- this is circular as the subjective values are modeled by choice probabilities and*
*then correlated with choice probabilities.*

**Authors' response:** We assume the Reviewer refers to Fig 1h. These graphs were not
intended as a statistical test but to illustrate the systematic relationship between subjective
value difference and choice probabilities in both animals. We now mention that these values
were derived from the regression in Fig. 1g:

Lines 167-169: "(h) Psychometric functions illustrating the systematic relationship between
subjective values (modelled from logistic regression) and choice probabilities."

*6. Line 324 – 6% - is this above chance level?*

**Authors' response:** A binomial test indicated that this was not above chance. We now
include the following to address this point:

Lines 365-367: "identified 30 amygdala neurons with significant coding of expected value
(6%, eq. 4), which was not different from chance ($P > 0.05$, binomial test)."

7. The authors describe two methods for neuron selection, based on a fixed or running window. It is not clear which method was used for different presentations (e.g. fig. 1f)

**Authors' response:** Thank you. We now explicitly mention in each figure which method was used for selecting neurons.

8. To demonstrate the integrated signal in the examples, the authors should show some consistency of the integrated signal across levels of a single signal. For example, show fig5A
for low and high expected values within the levels of probabilities (plotting high, mid and low
EV only in high probability, only in medium probability and only in low probabilities). Same
for the different examples and the population level in 4d.

**Authors' response:** Thank you for this suggestion. We now show in new supplementary
figures Fig. S3 and S4 the neurons' activities split by probability and magnitude levels.

**“Fig. S3. Amygdala neuron coding risk. (a)** Activity of the neuron in Fig. 4a, shown
separately for different probability and magnitude levels. **(b-d)** Activity of the neurons in Fig.
4i-k, shown separately for different probability and magnitude levels.”

“Fig. S4. Amygdala neuron coding expected value. Activity of the neuron in Fig. 5a, shown separately for different probability and magnitude levels.”

9. Fig. 6f,h,l – what is the gray line?

Authors’ response: We now state in the legend that the gray line represents decoding results based on shuffled data.

10. The authors only examine expected value of the integrated signal, but expected value also exist when the first signal appears (with the expected value of the second signal can be calculated). It would be interesting to see such expected value coding.

Authors’ response: Thank you for the suggestion; however, expected value at the first cue is identical to probability and thus we can cannot separate these two variables in that trial period. We now mention this limitation in the Results:

Lines 369-370: “(Expected value at the first cue was identical to the probability regressor, and these variables could thus not be distinguished in the first-cue period.)”

**Reviewer #2 (Remarks to the Author):**

*Dynamic coding and sequential integration of multiple reward attributes by primate amygdala*
*neurons*

*Grabenhorst and Báez-Mendoza*

*This paper reports interesting new findings on the primate amygdala in experiments that*
*have been carefully performed and analysed.*

**Authors' response:** We thank the Reviewer for this positive evaluation of our work.

*I have the following comments to help the authors improve the paper.*

*1. The use of a Pavlovian task (in which the animal can make no choices) is a bit of a*
*surprise, for it could be that the animal may not need to integrate the variables (reward*
*probability and magnitude) as well as if an instrumental choice had to be made between*
*what was on offer and a comparison reward.*

*The authors should explain the pros and cons of their choice of a Pavlovian task, and*
*consider this possible limitation.*

**Authors' response:** Thank you. We now added a more detailed explanation of our rationale
for using a Pavlovian task and discuss advantages limitations of this design choice in the
Discussion as follows:

Lines 642-653: "Our use of a Pavlovian task with sequential cue presentations enabled us to
temporally dissociate probability and magnitude information, to investigate whether individual
amygdala neurons responded selectively to one or both of these value components, and
whether they integrated these components to risk and expected value. Further, we could
examine neuronal responses to individual reward-predictive cues, in the absence of additional
cognitive processes that occur during choice tasks, including value comparisons and decision
computation. A possible limitation is that our task did not explicitly require integrating reward
components for decision-making, although trial-abortion rates and lick responses suggest that
the animals used information derived from the cues for reward expectation. Moreover,
previous studies showed that amygdala encoding of additional variables such as trial-specific
choices, internally planned choice sequences, and internally tracked accumulated rewards
was specific to free-choice situations and largely absent during imperative, non-choice
tasks^{10,11,19}. Thus, it will be interesting to study amygdala neurons in a choice situation with
the cues used in the present task."

*2. The investigators did run a choice task on some days. Were any recordings of neuronal*
*activity made in the instrumental task? If so, they should be reported in this paper for*
*comparison, and to address the point in (1).*

**Authors' response:** Thank you for this suggestion. Unfortunately, we did not record
neuronal data in this choice task so cannot report these data for comparisons. We agree that
such comparison would be informative but this was not within the scope with which the
present study could be conducted (included in the Discussion as described in our response
to point 1):

*3. Some type of effect size should be reported. The beta in a GLM does not provide an effect*
*size that can be used as a scale-free metric. To help the authors appreciate the point being*
*made, a Cohen's d measure, or a mutual information measure, could provide a scale-free*
*metric.*

**Authors' response:** Thank you for this comment. We do report two different measures of
effect size: the coefficient of partial determination (partial R^2) and the % correct decoding
accuracy for different variables throughout the figures. We appreciate the importance of the
mutual information measure but feel that quantifying the coding properties of amygdala
neurons in our task for the different variables in this way would require a dedicated
manuscript, as this was not the goal of our study.

*4. It is somewhat surprising that relatively few neurons encoded expected value, which is*
*presumably what would be useful to the animal at least in an instrumental task. The authors*
*should comment on why they think relatively few amygdala neurons get this far; and on how*
*neurons in the orbitofrontal cortex may compare with these amygdala neurons.*

**Authors' response:** We agree with this assessment and now include the following in the
Discussion (please also see our responses to points 6 and 7 for comparisons with
orbitofrontal cortex):

Lines 612-627: "This result could suggest that expected value was not strongly represented
in amygdala single neurons in this task, which required integrating probability and magnitude
across sequential cues (despite monkeys' separately measured choices reflecting expected
value). In a previous study using a choice task, we showed that amygdala neurons do
encode subjective values that reflected integrated reward probability and magnitude when
these reward attributes were cued simultaneously¹². Perhaps neurons in prefrontal, including
orbitofrontal, and parietal cortices might be relatively more important in signalling
accumulating decision variables derived from sequential or otherwise complex cues^{28,60-63}."

*5. The latency of some of these neuronal responses in the amygdala would be of interest, for*
*they might enable assessment of where the inputs that drive these neurons come from.*

**Authors' response:** Thank you for this helpful suggestion. We included this information as
follows:

Lines 256-260: "The population signals for probability and magnitude in **Fig 3k** peaked at 161
and 181 ms, respectively, following the onset of the relevant cue (probability in specific nuclei
LA/BL/BM/Ce: 121/161/121/201 ms; magnitude in specific nuclei LA/BL/BM/Ce:
161/201/221/161 ms). (Note that signals from different nuclei were not simultaneously
recorded.)"

Lines 305-307: "The population signal for risk identified in this sliding-window regression
peaked at 221 ms following cue onset (probability in specific nuclei LA/BL/BM/Ce:
121/161/121/201 ms; magnitude in specific nuclei LA/BL/BM/Ce: 241/201/161/161 ms)."

*6. In that context, the authors should consider more how these amygdala neurons'*
*responses compare to those in the orbitofrontal cortex. Could some of these amygdala*
*neurons' responses reflect what is represented in the orbitofrontal cortex?*

**Authors' response:** We now include the following in the Discussion to address this point (in
addition to the new text already mentioned in response to point 4):

Lines 686-691: "It is possible that the observed dynamic coding patterns in amygdala nuclei
reflect inputs from other brain structures such as orbitofrontal cortex and subgenual
cingulate cortex. A recent study⁸⁷ reported that basolateral amygdala neurons during reward
anticipation signalled reward presence and type in with more extended activity patterns than
neurons in subcallosal anterior cingulate cortex and rostromedial striatum. A risk code in BL
is consistent with BL inputs from orbitofrontal cortex^{58,88}, where neurons also code risk^{22,24}."

7. In recordings made from primate amygdala neurons, it has been reported that the
responses of the neurons tend to be much more idiosyncratic in how they represent
expected value than for the orbitofrontal cortex. Do the authors have any thoughts that are
relevant to this issue, and whether the responses of orbitofrontal cortex neurons would be
more useful in guiding behaviour because they are more closely related to expected value,
which is what some previous investigators have reported?

**Authors' response:** Thank you for raising this issue, which is very important and interesting
but too complex to unpack fully in this paper. Nevertheless, we now include the following
new text in the Discussion:

Lines 628-633: "Some previous studies found largely similar coding of values and choices in
amygdala and orbitofrontal cortex^{13,64}, while others emphasised differences in the
timecourses with which neurons in these structures track changing values^{65,66}, and in the
specificity with which single neurons encode complex, multisensory food rewards⁶⁷. Future
studies could combine our task with simultaneous recordings from amygdala and
orbitofrontal cortex to uncover possible differences in how these two structures able to
integrate temporally dissociated value components to guide behaviour."

**Reviewer #3 (Remarks to the Author):**

*The manuscript provides an important demonstration of the encoding of diverse reward*
*attributes in the amygdala, which combined together carry information relative to expected*
*value and risk, likely guiding economical choices. Importantly, neuronal activity discriminated*
*for levels of probability for all cues predicting specific probability, indicating that amygdala*
*neurons generalized across stimuli and encoded probability in an abstract format. Further,*
*authors observed that the same neurones also encoded reward magnitude, which when*
*combined with probability also carried information about the risk associated to the current*
*trial. One of the manuscript's important finding is that these signals are present differentially*
*within the amygdala nuclei: only lateral amygdala supports reward probability and magnitude*
*signals transferrable over time periods.*

*The paper provides a very strong and set of analysis to support the conclusions, and rests*
*on a very robust set of data.*

*I have a few points for authors to clarify especially with respect to the concepts of variance,*
*risk and uncertainty.*

**Authors' response:** We are grateful to the Reviewer for emphasising the relevance of our
findings and robustness of our data set.

*1) the definition of risk in the intro and throughout the manuscript is unclear.*
*page 3 line 100, risk is defined as uncertainty (and in the figure 1b), but then the equation in*
*figure 1a defines risk as the variance (probability factoring magnitude and the expected*
*value). The latter takes into account the potential loss (risk is maximal at higher gain with*
*lower probability), while the risk in figure 1b only computes uncertainty (max at $p(\text{reward})=$*
*0.5, lower at $p(\text{reward})=0.75$ and 0.25). However, variance which is a term that is coined*
*later to describe (line 264) does not really convey the risk computation which takes into*
*account potential reward loss (at same uncertainty of .25 or .75).*

*The authors should clarify their definition, and maybe compare clarify that "risk' is better than*
*uncertainty alone.*

*The mix between the different terms (variance in distribution of reward, risk and uncertainty)*
*is confusing.*

**Authors' response:** Thank you for pointing out the need for more clarity. We now follow the
suggestion and refrain from using the word 'uncertainty' and instead use the more specific
term 'variance' when referring to risk. We also include in the Introduction a statement that
clarifies what information is taken into account in the variance calculation, and that we adopt
the definition of risk as variance (following economic and finance theory) rather than the
definition of risk as 'probability of loss or a negative event' (common in everyday language).
This is the more common approach taken in previous neuroscience studies investigating risk
coding in an economic framework (e.g., Genest et al., 2016, PNAS, O'Neill and Schultz,
2010, Neuron; Preuschoff et al. 2006, Neuron).

Lines 72-75: "We adopt the approach common in economic and finance theory that
measures risk as variance²⁹⁻³², which takes into account the variance of the statistical
distribution of possible outcomes (both positive and negative), which differs from the notion
of risk as 'probability of loss'."

*2) Can authors specify if (reward) =0.5 were excluded from analysis in Figure 1g, especially*
*since example in fg 1e involved $p(\text{rew})$ of .5.*

*I also do not understand how the results in Fig 1i (animals prefer the safe reward over the*
*gamble) fit with the risk seeking results of animal B (who prefers options with higher risk in*
*fig 1g).*

*Can authors define constant on Fig 1j and 1g?*

**Authors' response:** Thank you for pointing out these issues. We now explain how we
defined low- and high-probability trials, and how the test of first-order stochastic dominance
is related to the risk-seeking tendency of the animals. We also define how the word
'constant' is used in the different panels.

Lines 166-167: "(g) Logistic regression of choices on expected value and risk for low (top, p
< 0.25) and high (bottom, $p > 0.25$) probability trials."

Lines 919-921: "The logistic regression of choices on expected value and risk was
performed separately for low ($p < 0.25$) and high ($p > 0.25$) probability trials."

Lines 135-140: "The animals' choices also conformed with first-order stochastic
dominance⁵¹: when choosing between a safe reward and a gamble of equal magnitude (i.e.,
of lower expected value), the animals preferred the safe reward (**Fig. 1i**). (Note that the
animals' risk-seeking tendencies defined in the regression in **Fig. 1g** suggest preferences for
risk when expected value is accounted for (i.e., is included as covariate); by contrast,
following first-order stochastic dominance in **Fig. 1i** indicates preference for higher expected
value.)"

*3) The distribution of individual cells in the space is nice, but, the light and blue green are
very difficult to differentiate visually on fig 2C and fig3g. Please draw boundaries or change
colors.*

**Authors' response:** Thank you for pointing this out. We now use a more saturated green for
better visual discrimination.

*4) Could the authors measure the latency of the probability or magnitude signal across
nuclei?*

*Figure 1c shows that there are cells that increase activity for lower p (negative beta). Do
these cells have the same latency, and how do authors interpret these cells? Are these more
likely to be sensitive to risk?*

*Actually, this seems to be the case, because figure 4d show that higher activity to low p ,
followed by higher activity for higher risk.*

*This may have an implication for what is computed by the cells. Maybe authors could
comment on this?*

**Authors' response:** Thank you for these suggestions. We include the population-coding
latencies derived from sliding window regression. With respect to the negative probability-
coding neurons, a latency comparison was not conclusive and as the neurons were not
simultaneously recorded, and too few risk-coding neurons were preceded by probability
coding (13/106) to allow quantitative comparisons between positive and negative coding
schemes; therefore, we prefer not to include this analysis. However, we do appreciate the
interesting point raised by the Reviewer and included a comment on this in the Discussion,
as shown below.

Lines 256-260: "The population signals for probability and magnitude in **Fig 3k** peaked at 161
and 181 ms, respectively, following the onset of the relevant cue (probability in specific nuclei
LA/BL/BM/Ce: 121/161/121/201 ms; magnitude in specific nuclei LA/BL/BM/Ce:
161/201/221/161 ms). (Note that signals from different nuclei were not simultaneously
recorded.)"

Lines 305-307: "The population signal for risk identified in this sliding-window regression
peaked at 221 ms following cue onset (probability in specific nuclei LA/BL/BM/Ce:
121/161/121/201 ms; magnitude in specific nuclei LA/BL/BM/Ce: 241/201/161/161 ms)."

Lines 612-617: “Different neurons encoded probability at the first cue with a positive or
negative coding scheme, and some of these neurons transitioned to encoding risk at the time
of the second cue. Although too few neurons showed probability-to-risk transitions to allow
quantitative analyses (13/106), it is possible that neurons with positive and negative coding
schemes play different roles in the computation of subsequent risk, which could be
investigated in future studies.”

*5) the sentence line 212, p7: These responses are conceptually important because they*
*could reflect value irrespective of the specific reward parameter and provide a single-neuron*
*basis for integrating probability and magnitude into expected value or risk.*

*If the cells signal value irrespective of the specific reward, then they already signal expected*
*value. Please clarify the sentence. I suppose that authors mean that the graded signals*
*could provide a single neuron basis to integrate magnitude and probabilities over time into*
*expected value and risk.*

**Authors’ response:** Thank you for pointing out the ambiguity in this sentence. We have
rephrased as follows:

Lines 231-233: “These sequential, graded responses in the same neuron are conceptually
important because they could provide a single-neuron basis for integrating probability and
magnitude over time into expected value or risk.”

*6) Line 283: it says that some neurons responded “differently to the highest risk indicated by*
*probability of $P = 0.5$, compared to low risk indicated by $P = 0.25$ and $P = 0.75$ ”*

*I think (cf point 1) that this is uncertainty (highest at .5) , and not risk.*

*Within any reward magnitude, a p of 0.25 is morerisky than a p of 0.5 according to equation*
*in fig 1.*

*Next, according to all reward magnitudes and probabilities, isn’t risk a continuum of numbers*
*ranging from lowest risk Magnitude of 0.1 with a p (reward) of 0.75 to highest (0.9 at 0.25).*

*How did authors regroup risks into 3 categories ins’t clear from main text.*

**Authors’ response:** Thank you for pointing out the need for clearer explanation. We have
rephrased this sentence as follows to indicate that at the first cue, risk derived purely from
probability, whereas at the second cue, risk derived from both probability and magnitude. We
also clarify that for plotting timecourses, we grouped the data according to variance terciles.
In addition, we performed new analyses in which we included two competing risk regressors
in the same GLM: risk derived purely from probability and risk derived from the integration of
probability and magnitude.

Lines 323:326: “these neurons showed risk coding both at the first, probability cue (coding
risk derived from probability only, by responding differently to the highest risk indicated by
probability of $P = 0.5$, compared to low risk indicated by $P = 0.25$ and $P = 0.75$), followed by
a risk signal at the second, magnitude cue (coding risk derived from both probability and
magnitude, according to eq. 1).”

Line 342: “Peri-event time histogram sorted by risk (variance) terciles.”

Lines 314-321: “In a further model, we included two variants of the risk regressor to model
risk derived solely from probability and risk derived from the integration of probability and
magnitude (in addition to probability, magnitude and expected value). This model identified
103 risk neurons (21%) for risk derived from the integration of probability and magnitude and
65 risk neurons (13%) for risk derived from probability; 27 of these neurons were significant
for both regressors. The timecourses of these two types of risk signals showed that risk from
probability was coded prominently during the first-cue period but also during the second-cue

period, whereas risk from probability and magnitude was coded only during the second-cue
 period (Fig. 4f).”

 “(f) Partial R² from a sliding-window multiple regression analysis across all recorded neurons
 with a separate regressors for risk from probability and risk from integrated probability and
 magnitude.”

7) which neurons exactly are used to make figure 5b, c, d?
 Is it the 69 neurons described earlier that encode probability followed by magnitude?
 Can the authors specify the conditions, monkeys and neurons used for figure 5e?

**Authors’ response:** We now provide this information as follows:

Lines 1071-1088: “**Construction of population expected-value signals.** For the plots in Fig.
 5b-e, probability-coding and magnitude-coding neurons were identified by regressing neuronal
 activity in the first and second cue period on probability and magnitude (eq. 3). For the plot in
 Fig. 5c, we then used these neurons to first calculate the mean z-normalized responses across
 probability-coding neurons to each probability level in the probability-cue period (sign-
 correcting neurons with negative coding), and correspondingly calculate the mean z-
 normalized responses across all magnitude-coding neurons to each magnitude level in the
 magnitude-cue period (sign-correcting neurons with negative coding). Next, we multiplied
 these probability- and magnitude-level specific responses to calculate hypothetical expected-

value signals for each of the 15 probability-magnitude combination defining a specific
expected-value level. We then used these 15 'constructed' expected-value activities to
calculate activities plotted as a surface in Fig. 5c using interpolation for the oversampled
expected-value space shown in Fig. 5c. For the plot in Fig. 5d, we performed linear regression
of these constructed (from separate probability and magnitude activities) expected-value
signals on the corresponding expected-value levels (separately for signals from neurons with
positive coding scheme and for the sign-corrected signals for neurons with negative coding
scheme). For Fig. 5e, we first calculated the choice probability that a choice option with a given
expected-value level was chosen (using the separately recorded choice-task data, averaging
across animals), and regressed these choice probabilities on the neuronal activities
corresponding the these expected-value levels.”

*8) I wonder what the decoding on figure 6h implies for decoding of cells in which training was*
*made on probability cues (low vs high), and tests were done on magnitude cues. Were*
*magnitudes also regrouped into two (high and low). Does this imply a one to one relationship*
*between probability and magnitudes such that the state of the population for high probability*
*was the same as for the high magnitude? Please explain how the decoding generalizes*
*across these two periods?*

**Authors' response:** Thank you for raising this interesting point. Our interpretation is that at
the time of the second, magnitude cue, the amygdala population code is more specialized to
reflect the variety of signals that can be decoded at this point in the trial (magnitude, risk,
expected value). This complexity of the population code would likely limit the utility of a
decoder trained on activity in this trial period to accurately readout probability from the earlier
trial period. On the other hand, the finding that training on the probability-cue period and
decoding from the magnitude-cue period was possible suggests a more generalized
population value code in this early trial period. We now include this interpretation as follows:

Lines 453-459: “This pattern of results suggests that at the time of the second, magnitude cue,
the amygdala population code is more specialized to reflect the variety of signals that can be
decoded at this point in the trial (magnitude, risk, expected value). This complexity of the
population code would likely limit the utility of a decoder trained on activity in this trial period
to accurately readout probability from the earlier trial period. On the other hand, the finding
that training on the probability-cue period and decoding from the magnitude-cue period was
possible suggests a more generalized population value code in this early trial period.”

*9) The figure 7c, I am not sure whether the colors for BLA and LA were inverted, such that*
*BL should be black! also the figure says magnitude while the text says probability (7c left).*

**Authors' response:** Thank you – we sincerely apologise for this mistake in the colours and
the confusion it might have caused. We now corrected this, and also clarified the text relating
to magnitude.

*10) This may be the scope of a different paper, but how did amygdala neurons encode*
*reward presence or absence of reward at the time of reward delivery, especially for neurons*
*encoding reward probability?*

*This may just be interesting to understand how outcome is monitored with respect to*
*probability integration. I understand that animals are over-trained and already known the*
*probabilities, but I was just curious if absence of reward, was signaled differentially given*
*high or low uncertainty.*

**Authors' response:** We agree this is an important and interesting issue that also relates to
the unresolved question of whether primate amygdala neurons might code reward prediction
errors. We are planning to address this issue comprehensively in a separate manuscript
using data from the present study and other data sets.

Reviewer #4 (Remarks to the Author):

*This manuscript by Grabenhorst and Báez-Mendoza describes the encoding properties of*
*neurons in the macaque amygdala in a Pavlovian reward association task. In every trial the*
*monkeys are show a sequence of visual cues that indicate the probability and magnitude of*
*an upcoming fluid reward. Reward probability refers to the likelihood of receiving the*
*indicated reward magnitude (range 0.1 to 0.9ml) vs. receiving a small nominal reward of*
*0.05ml. This scheme permits each trial to be parameterized according to reward probability,*
*reward magnitude, the expected value (EV, product of probability and magnitude), and risk,*
*defined as the variance of the distribution of expected outcomes of a given probability-*
*magnitude bundle. Two distinct sets of probability cues were used (fractals and pie-like*
*“sector” stimuli). In the main Pavlovian task, no behavioral measures were analyzed other*
*than trial abort rates; in a separate choice task using the same stimuli, the monkeys’ choices*
*indicated that they assigned appropriate EVs to the bundles offered on each trial. The main*
*claims of the paper are: 1) Single amygdala neurons encoded the reward probability and*
*magnitude in serial fashion, consistent with the presentation order of the two cues. 2) Single*
*amygdala neurons usually encoded reward probabilities with a consistent code for the two*
*stimulus sets. 3) Many neurons encoded the risk associated with a given magnitude-*
*probability bundle, whereas very few encoded the EV. However, an accurate EV signal could*
*be reconstructed from the separate probability and magnitude signals encoded at time of the*
*respective cue presentations. 4) The results of pseudopopulation-based decoding analyses*
*and RSA template analyses generally corroborated the single-neurons results, with some*
*differences evident in neurons in the LA and BL subnuclei.*

*At a technical level, this is a very well-executed study. I am confident in the results, and my*
*comments (see below) are small-to-modest and mostly asking for clarification of technical*
*points. If I were to identify any major shortcoming in this study, it would be that the*
*behavioral significance of the amygdala coding properties is unclear. Because a Pavlovian*
*task was used, and because no Pavlovian responses were recorded (e.g. pupil size or*
*licking), it is not possible to draw strong mechanistic conclusions about how the activity of*
*the amygdala neurons might reflect or contribute to behavior. Therefore, in my opinion this*
*study represents a very solid, but modest, advance in our knowledge of primate amygdala*
*function.*

**Authors’ response:** Thank you for this positive evaluation of our study, and also for raising
this important point regarding behavioural responses during the task, which prompted us to
re-examine the lick data we recorded in the task. We now include in a supplementary table
the results of lick data, to provide additional support for the behavioural relevance of the
recorded neuronal signals during the Pavlovian task. We had not originally included these
data as one of the monkeys seemed to show somewhat variable licking responses (although
we carefully controlled the juice spout position and distance to the animal’s mouth). Upon
analysing these data with mixed-effects regression (allowing for session-dependent variation
in licking), we find that in both animals, magnitude and risk had significant relationships with
licking behaviour. We now include these results in a new Table 3, and included the following
text in Results, Discussion and Methods:

Lines 149-152: “Anticipatory lick rates also reflected the influence of reward variables in
consistent ways across animals: mixed-effects regression showed that higher lick rates
between onset of first cue and reward delivery were associated with lower reward
magnitudes and higher risk levels, respectively (Table 3).”

**Table 3.** Mixed-effects regression of licking behaviour on task variables.

Variable	Estimate	Standard error	t-statistic	Degrees of Freedom	P-value
----------	----------	----------------	-------------	--------------------	---------

Animal A					
Intercept	-0.001	0.007	-0.203	16,037	0.838
Probability	-0.014	0.017	-0.791	16,037	0.428
Magnitude	-0.170	0.370	-4.607	16,037	4.1e-6
Expected value	-0.024	0.029	-0.838	16,037	0.401
Risk	0.074	0.028	2.594	16,037	0.009
Animal B					
Intercept	0.0002	0.005	0.038	31,099	0.968
Probability	-0.027	0.015	-1.774	31,099	0.075
Magnitude	-0.210	0.038	-5.515	31,099	3.5e-8
Expected value	0.085	0.025	3.345	31,099	0.0008
Risk	0.124	0.026	4.648	31,099	3.3e-5

Line 548-551: “The animals’ well-structured preferences between gambles and risky choice options and relationships between reward variables, motivational engagement and reward expectation (**Fig. 1, Tables 1-3**) suggest that these neuronal codes are suited for informing reward expectations and economic choices between goods.”

Lines 919 – 930: “The logistic regression of choices on expected value and risk was performed separately for low ($p < 0.25$) and high ($p > 0.25$) probability trials. Subjective values derived from this equation were used to construct the psychometric functions shown in Fig. 1h. In a third model, we included the following regressors to examine the factors related to the animals’ eye-fixation breaks (i.e., errors in task performance) and lick responses (in separate regressions):

$$y = \beta_0 + \beta_1 \times (Probability) + \beta_2 \times (Magnitude) + \beta_3 \times (EV) + \beta_4 \times (Risk) | Session$$

where y indicated whether or not the trial was aborted (logistic regression for trial-abortion rates) or the z-normalised lick rate measured from onset of first cue to reward onset (linear regression for lick data). The coefficients from this model are shown for trial-abortion rates in Fig. 1j and Table 2 and for licking data in Table 3.”

Interpretation comments:

- Fig 5 suggests that by multiplying the probability and magnitude signals recorded at
different times, one can recover something that looks like an EV signal. Do the authors think
that these two separate signals are integrated downstream of amygdala? If so, where?

**Authors’ response:** Thank you for raising this issue. We now provide more detailed
discussion of this point as follows:

Lines 621-633: “This result could suggest that expected value was not strongly represented
in amygdala single neurons in this task, which required integrating probability and magnitude
across sequential cues (despite monkeys’ separately measured choices reflecting expected
value). In a previous study using a choice task, we showed that amygdala neurons do
encode subjective values that reflected integrated reward probability and magnitude when
these reward attributes were cued simultaneously¹². Perhaps neurons in prefrontal, including
orbitofrontal, and parietal cortices might be relatively more important in signalling
accumulating decision variables derived from sequential or otherwise complex cues^{28,60-63}.”

Some previous studies found largely similar coding of values and choices in amygdala and
orbitofrontal cortex^{13,64}, while others emphasised differences in the timecourses with which
neurons in these structures track changing values^{65,66}, and in the specificity with which single
neurons encode complex, multisensory food rewards⁶⁷. Future studies could combine our
task with simultaneous recordings from amygdala and orbitofrontal cortex to uncover
possible differences in how these two structures able to integrate temporally dissociated
value components to guide behaviour.”

- *On a related note: In my opinion the statement starting at line 541 seems to be a stretch.*
*First, there is no reward-related behavior measured in this task, so it's difficult to make this*
*conclusion. Second, I think it's highly likely that the monkeys are aware of the EV of the*
*bundles (as indicated by the choice task), but that this variable is just not reflected in single*
*amygdala neurons.*

**Authors' response:** Thank you. This has been rephrased and addressed in the revised
paragraph shown just above.

*Technical comments:*

- *It is unclear what distinguishes GLM2 and GLM3, as the equations are the same. Is the*
*difference due to the time windows in which these GLMs were analyzed?*

**Authors' response:** Thank you; we now clarify this as follows:

Lines 1005-1006: “This GLM was identical to GLM2 except that it was calculated on fixed
neuronal-response windows rather than with a sliding window.”

- *GLM2 and GLM3 seem to be nested within GLM4. Why not just fit GLM4 and use the*
*coefficients for all analyses? If there's a concern about overfitting by using GLM4, then it*
*might make sense to try stepwise models or some other model selection procedure.*

**Authors' response:** Thank you for this suggestion. Although the models can be thought of
as nested, we prefer to keep them as separate analyses for the separate purposes
described because of the conceptual difference between explicitly cued and pre-trained,
variables probability and magnitude and the non-cued, integrated variables expected value
and risk. GLMs 2/3 were used to quantify the extent to which probability and magnitude as
the most basic variables were encoded by neurons, whereas GLM4 was used to test
whether the more complex integrated variables were also encoded, while controlling for
probability and magnitude. We now explain this rationale more carefully.

- *How many total template matrices are used in the regression analysis? If I read the*
*methods correctly, it seems there could be a large number of matrices needed to capture the*
*combinations of different variables. This could lead to overfitting, which could point to the*
*need for some sort of model selection or regularization procedure.*

**Authors' response:** We used 9 template matrices as regressors in the RSA GLM analyses
with the dependent variable consisting of 465 data points (lower triangular part of the across-
neuron correlation matrix between 30 x 30 conditions (3 probability levels x 5 magnitude
levels x 2 cue sets). Thus, we had sufficient data points to fit a GLM with 9 regressors. We
now explain the composition of these models in more detail as follows:

Lines 1179-1182: “Thus, we used 9 template matrices as regressors in the RSA GLM
analyses with the dependent variable consisting of 465 data points (lower triangular part of
the across-neuron correlation matrix between 30 x 30 conditions (3 probability levels x 5
magnitude levels x 2 cue sets).”

- In addition to the variables already analyzed (probability, magnitude, etc.) one potential variable that could be encoded is belief confirmation (for details see Hunt et al. Nat Neuro, 2018). In this task, a belief confirmation code would be evident at the magnitude cue, and would signal whether the value of the magnitude cue is congruent with (i.e. confirms) the value of the preceding probability cue.

Authors' response: Thank you for this interesting suggestion. We implemented this idea by performing an additional sliding-window regression in which we included a regressor that encoded whether the expected value at the time of the second cue was higher than the expected value of the first cue (thus modelling sensitivity to belief confirmation). We note that a similar concept has been used to investigate 'state value' in an earlier paper in amygdala neurons. We now cite both papers and include a summary of the results of this analysis as follows:

Lines 380-386: "We also tested whether amygdala neurons at the time of the second cue were sensitive to a change in value relative to the first cue; such value change in a trial has previously been conceptualized as 'belief confirmation'²⁸ or 'state value'¹. We performed a sliding-window regression in which we included a regressor that encoded whether the expected value at the time of the second cue was higher than the expected value of the first cue. We found that 36/483 amygdala neurons (7%) showed a significant effect for this value-change regressor. We interpret these responses as reflecting previously reported state-value coding in amygdala¹."

- A few additional details on the choice task would be welcome. For example, was the "safe" option always the same magnitude in every trial? Or did it vary across trials? Was there an additional fractal or sector cue that indicated $p=1$ for the safe option on each trial?

Authors' response: We now provide this information as follows:

Lines 849-853: "On each trial, the animal made a choice between two gambles, one of which was a 'safe' option, or 'degenerate gamble' (reward probability of $P = 1$, trial-by-trial varying reward magnitudes), presented randomly in left-right arrangement on the monitor. The safe option was cued only by a reward-magnitude cue, implying a reward probability of $P = 1$."

- The neural screening procedure should be mentioned in the results section, i.e. that the 483 neurons analyzed were out of ~1000 screened for task responses online.

Authors' response: We now include the following statement in the Results:

Lines 178-182: "We sampled activity from about 1,000 amygdala neurons and typically recorded and saved the activity of those neurons that appeared to respond to any task event during online inspection of several trials. We aimed to identify task-responsive neurons but did not preselect based on particular response characteristics. This procedure resulted in a database of 483 neurons that were recorded and analyzed statistically."

- Pertaining to Figs. 2c,d and related text: The results state that among 148 probability-coding neurons, 84 code probability for both cue formats. But Fig. 2c suggests that a minority of these 84 have opposite encoding signs, e.g. positive for fractals and negative for sectors. Intuitively, it seems that these few opposite-sign neurons do not code for probability in a 'cue-unspecific' format, and should be counted separately from the neurons with significant same-sign encoding.

Authors' response: Thank you for pointing this out. We now include the following statement in Results:

Lines 200-202: "Of 84 neurons with significant probability coding for both cue formats, 14 neurons coded probability with different signs across cue formats. Thus, 70 of 483 neurons (15%) coded probability irrespective of cue format with matching signs."

- Fig. 1b shows the intuitive relationship between probability-based risk and probability alone. It would be helpful to also see an illustration of how risk relates to probability, magnitude, and EV when risk is computed using both probability and magnitude. These relationships are less intuitive because it involves different magnitudes, and because, in this task, the "non-rewarded" outcomes occurring in the 1-P portion of trials actually involve delivery of a small reward.

Authors' response: Thank you for the suggestion. We now include a schematic as new Fig. S3.

Fig. S3. Relationship of risk to probability, magnitude and expected value. Variables are plotted for a typical testing session.

- On a related note it would be nice to see in the supplements a table of correlations between regressors in GLM4

Authors' response: We now include this table as follows:

Table S1. Correlations between regressors in GLM 4 (Pearson correlation coefficients).

	Probability	Magnitude	Expected value	Risk
Probability	1			
Magnitude	-0.003	1		
Expected value	0.486	0.806	1	
Risk	-0.003	0.943	0.760	1

- Line 271: Should this read "eq. 4" instead of "eq. 3"?

Authors' response: Thank you – corrected.

- A little more clarity on the operation shown in Fig. 5c would be helpful, e.g. in a brief paragraph in the Methods.

Authors' response: We now provide this information in the Methods as follows:

Lines 1071-1088: "**Construction of population expected-value signals.** For the plots in Fig. 5b-e, probability-coding and magnitude-coding neurons were identified by regressing neuronal

activity in the first and second cue period on probability and magnitude (eq. 3). For the plot in
Fig. 5c, we then used these neurons to first calculate the mean z-normalized responses across
probability-coding neurons to each probability level in the probability-cue period (sign-
correcting neurons with negative coding), and correspondingly calculate the mean z-
normalized responses across all magnitude-coding neurons to each magnitude level in the
magnitude-cue period (sign-correcting neurons with negative coding). Next, we multiplied
these probability- and magnitude-level specific responses to calculate hypothetical expected-
value signals for each of the 15 probability-magnitude combination defining a specific
expected-value level. We then used these 15 'constructed' expected-value activities to
calculate activities plotted as a surface in Fig. 5c using interpolation for the oversampled
expected-value space shown in Fig. 5c. For the plot in Fig. 5d, we performed linear regression
of these constructed (from separate probability and magnitude activities) expected-value
signals on the corresponding expected-value levels (separately for signals from neurons with
positive coding scheme and for the sign-corrected signals for neurons with negative coding
scheme). For Fig. 5e, we first calculated the choice probability that a choice option with a given
expected-value level was chosen (using the separately recorded choice-task data, averaging
across animals), and regressed these choice probabilities on the neuronal activities
corresponding the these expected-value levels."

*- The methods describe both nearest-neighbor and SVM classifiers. Which ones were used*
*for which results?*

**Authors' response:** We initially performed analyses using both classifiers but all the results
included are based on nearest-neighbor decoding (SVM decoding accuracies were similar
but typically higher, consistent with the SVM algorithm being more powerful but also less
biologically plausible). We now removed the description of the SVM decoding to focus on the
NN results. Although a systematic comparison between the decoding algorithms would be
interesting, we feel this would require a separate report to not detract from our main results.

*- I sense a small contradiction between the results of Fig. 6a, showing strong decoding of*
*EV, and the claim that encoding of EV in single neurons was infrequent (6%). How can the*
*strong decoding of EV be explained if single neurons don't encode it? E.g. is it subthreshold*
*encoding distributed over many neurons? Or do the neurons that encode EV do so with high*
*signal to noise?*

**Authors' response:** Thank you for raising this interesting point. We now include the
following new text in the Discussion to address this issue:

Lines 633-641: "Although few individual neurons explicitly encoded expected value (identified
with multiple regression), population decoding of expected value was relatively strong. It is
possible that individual neurons with subthreshold encoding of expected value or separate
probability and magnitude coding contributed to this high decoding accuracy. However, while
the decoding approach is suitable for quantifying what information can be decoded from the
population, it does not (in its basic form used here) allow for direct competition between
different, alternative variables in explaining neural activity, which we instead implemented with
our single-neuron GLM approach. Thus, it is possible that correlated variable (e.g., reward
magnitude) contributed to significant population decoding of expected value."

*- In Fig. 6b at trial epochs 3 and 4, there is below-chance classification accuracy for*
*probability. This is surprising, especially for epoch 3, which is presumably before even the*
*magnitude cue is shown. What explains this effect?*

**Authors' response:** Thank you for pointing this out. We agree that this is somewhat
unexpected. We confirmed the effect is not due to uneven trial numbers in decoding groups
(which were matched) and also occurs with SVM decoding. Upon closer inspection we

identified two outlier neurons that had unusually large rates after normalization that seemed
to affect the decoding. Repeating the analysis without these two neurons reduced the below-
chance decoding. We show this result below and include a note in the Methods but chose to
not replace the figure as we did not have a priori criteria for removing outliers.

Lines 1126-1128: "(The decoding performance slightly below chance for probability in Fig. 6b
was due to a few neurons with particularly high impulse rates in these periods.)"

- Fig. S3 legend: Should the legend for panel (e) read "Distribution of *risk*-decoding
accuracy? . . ."

**Authors' response:** Thank you – corrected.

- In Figure 6F, the low performance of the fractal-vs-sector cross decoding is surprising,
given that the high correlation shown in Fig. 2C, i.e. given that single neurons tend to encode
fractal and sector cues in the same way. Is there any explanation for this apparent
contradiction? This bears on the conclusion stated in line 478-79, i.e. that probability coding
is "abstract" and generalized across formats.

**Authors' response:** Thank you for bringing up this issue. Although the across-cues
decoding performance is relatively low, it is significantly above chance, thus allowing some
degree of cross-decoding. We have added the following to the Discussion to address this
point:

Lines 563-571: "Although the across-cues decoding performance for probability was relatively
low, it was significantly above chance, thus allowing some degree of cross-decoding. Previous
studies showed that different types of population code emphasize either the ability to
generalize across conditions or the coding capacity, i.e., the extent to which many different
task variables and their combinations can be read from the code⁵⁹. The present result
suggests that amygdala neurons in our task may have implemented a population code that
allowed significant but limited generalization, though many individual neurons showed clear
cue-invariant probability encoding. Future studies could use more complex designs to
determine the precise structure of the reward code in primate amygdala and its capacity for
high-dimensional coding vs. generalization."

- Pertaining to Fig. 6H and 6I: what encoding scheme would result in high cross-
classification in one direction but not the other? My intuition, perhaps incorrect, is that cross-
classification accuracy should be similar regardless of which stimulus is used for
training/testing. Is this perhaps related to the method used (nearest neighbor vs. SVM)?
Some clarification on this point would be welcome.

**Authors' response:** Thank you for raising this interesting point. We confirmed that this
pattern occurred irrespective of the specific classification method used. Our interpretation is
that at the time of the second, magnitude cue, the amygdala population code is more
specialized to reflect the variety of signals that can be decoded at this point in the trial
(magnitude, risk, expected value). In other words, it is not simply a matter of decoding
across different stimuli, but rather decoding across trial periods that differ in their
computational requirements. This complexity of the population code would likely limit the
utility of a decoder trained on activity in this trial period to accurately readout probability from
the earlier trial period. On the other hand, the finding that training on the probability-cue
period and decoding from the magnitude-cue period was possible suggests a more
generalized population value code in this early trial period. We now include this interpretation
as follows:

Lines 453-459: “This pattern of results suggests that at the time of the second, magnitude cue,
the amygdala population code is more specialized to reflect the variety of signals that can be
decoded at this point in the trial (magnitude, risk, expected value). This complexity of the
population code would likely limit the utility of a decoder trained on activity in this trial period
to accurately readout probability from the earlier trial period. On the other hand, the finding
that training on the probability-cue period and decoding from the magnitude-cue period was
possible suggests a more generalized population value code in this early trial period.”

REVIEWER COMMENTS

Reviewer #1 (Remarks to the Author):

The authors performed thorough revisions and answered my concerns. Although I am not fully convinced that they can demonstrate conclusively risk coding (rather than integration of probability/uncertainty/magnitude), I notice that they provided most analyses that, under the constraints of the behavioral paradigm, can supply reasonable supporting evidence. Putting risk aside, the behavioral paradigm is interesting and novel, and the results are carefully analyzed and presented. I therefore support publication as-is.

Authors' response: We thank the Reviewer for their helpful comments and for supporting the publication of our manuscript.

Reviewer #2 (Remarks to the Author):

NCOMMS-24-34111A
Dynamic coding and sequential integration of multiple reward attributes by primate amygdala neurons
Grabenhorst and Báez-Mendoza

I have been through the Responses to all 4 reviewers of the paper, and the revision of this paper, and believe that the authors have made appropriate revisions to this paper for it to be accepted in principle for publication in Nature Communications without further delays caused by sending it out again for further peer review.

The analyses now included in the paper are very detailed, and carefully done.

It is possible that with all this detail, the main conclusions of the paper may not be sufficiently easily assimilated by a reader of the paper. The main conclusion of the paper, it appears, is that in this Pavlovian task providing for separate and combined analysis of the effect of risk in decision-making as influenced by reward probability and magnitude, few amygdala neurons actually encoded the expected outcome, though expected outcome was reflected if the diverse types of neuronal response found in the amygdala to these variables could somehow be combined. The possibility remains that some other brain region, perhaps the orbitofrontal cortex, does represent at the neuronal level expected value as influenced by reward probability and magnitude. In addition, it is worth noting that the Pavlovian task may

*require less integration of these variables than an instrumental task might.*

*What I recommend is that the authors might be encouraged to draw out these points*
*somewhere in the paper rather clearly and briefly, with the editors of Nature*
*Communications checking that such an overall summary is easily available to help readers.*

*Overall, I recommend acceptance in principle, with no further delay caused by further peer*
*review, but possibly some clarification along the lines suggested and assessed by the*
*editors of the Journal.*

Authors' response:

We thank the Reviewer for this positive evaluation of our revised manuscript and for
endorsing publication in *Nature Communications*. We have followed the recommendation to
include a brief and clear summary of the points mentioned in the final paragraph of the
Discussion:

Line 731: "Few individual neurons encoded expected value, though expected value could be
recovered by combining probability and magnitude signals. It is possible that neurons in
other brain areas such as the orbitofrontal cortex encode expected value more directly in
Pavlovian tasks, which may require less integration into expected value compared to
instrumental, decision-making tasks."

**Reviewer #3 (Remarks to the Author):**

*I thank the authors for their clarification and the additional analysis which in my opinion*
*makes the paper almost ready for publication.*

*One last point, relative to their definition of risk and uncertainty early on.*

*Authors replied that they consider risk throughout the manuscript as "the variance of the*
*statistical distribution of possible outcomes (both positive and negative)". They also say line*
*104 "we adopted the frequently considered measure of economic risk as variance²⁹⁻³²: risk*
*defined in this way increased from $P = 0$ to $P = 0.5$ and decreased from $P = 0.5$ to $P = 1$,*
*following an 'inverted-U' function ». However, in the equation presented in figure 1A, var is*
*represented as the sum[$p(\text{magnitude}) \times \text{squared}(\text{magnitude} - \text{expected value})$] which, in effect,*
*takes the expected value as the product between magnitude factoring the probability of*
*magnitude. Doesn't then, this variance, change as a function of expected value, and isn't it*
*higher for low probability of high reward magnitude, and a lower for high probability of high*
*reward? Then, this is more like risk than just variance?*

*Thus the risk computed in the integration equation (also labelled as such in the title of figure*
*1a), is different from the risk computed in figure 1b, the latter being just uncertainty.*

*Throughout the manuscript, risk is then derived either i) from probability only (then it is*
*uncertainty) or ii) from the integration of probability and magnitude -then call it risk or*
*variance, but it should be for clarity defined differently from risk in figure 1b (black line),*
*which is simply uncertainty.*

*Then line 289, they say "A substantial number of 289 amygdala neurons (99/483, 20%; eq.*
*4) carried signals that reflected the risk of expected rewards, defined by the variance of the*
*current-trial statistical reward distribution derived from the integration of probability and*
*magnitude information (cf. Fig. 1a, b; Fig. S3)." Given that there is a reference to two distinct*
*definition of risk, this is confusing, as equation 4 I think refers to 1a, and not 1b.*

*Terms RiskProb and Risk are clear but only defined line 1060.*

*I think it would be easier if the same terms would be used throughout the figures and text,*
*and in figure 1A and 1B, they are not clearly labelled early on to avoid confusion;*

Authors' response: We are grateful to the Reviewer for supporting the publication of our
manuscript, and for their comments that helped us further improve the manuscript. We
changed the text at the start of the Results to clarify how risk varies as a function of both
probability and magnitude. For clarity, we now avoid using the term 'uncertainty', for clarity.
We now also explicitly refer to the equations used to calculate each form of risk throughout
the Methods, including in the passage mentioned by the Reviewer. We also include a new
figure panel (Fig. 1b) to illustrate how risk varies as a function of probability and magnitude.

Line 297: "The neuron in **Fig. 4a** encoded risk (i.e., variance as calculated in **Fig. 1a**)"

Line 102: "...expected value (calculated as the sum of all probability-weighted values of a
given reward distribution) and risk. Following previous studies, we adopted the frequently
considered measure of economic risk as variance, calculated as the sum of the probability-
weighted differences between outcomes and expected value²⁹⁻³²: risk defined in this way
increased from $P = 0$ to $P = 0.5$ and decreased from $P = 0.5$ to $P = 1$ for a given reward
magnitude, following an 'inverted-U' function, and increased across reward magnitudes for a
given probability level (**Fig. 1b**). Notably, the sequential presentation of probability and
magnitude cues allowed us to test neuronal encoding of risk purely dependent on probability
(at probability cue) and neuronal encoding of risk dependent on both probability and
magnitude (at the magnitude cue, following presentation of the probability cue)."

**“Figure 1. Sequential reward-prediction task and reward-guided behavior. (a) Design.**

Sequential cues for reward probability and reward magnitude test neuronal sensitivity to

multiple reward attributes and integration to expected value and economic risk. (b)

Schematic. Illustration of how expected value and risk depend on both reward probability

and magnitude.”

Line 855:

“We defined risk as follows (Fig. 1a). The variance of a probability distribution is defined as:

$$var = \sum_{ij} p_i \times (m_j - EV_{ij})^2$$

which, when applied to our probability distributions, is:

$$var = (1 - p_i)(0.05 - EV_{ij})^2 + p_i(m_j - EV_{ij})^2$$

where p_i is the probability of reward associated with cue i ; m_j is the reward magnitude151 associated with cue j ; and EV_{ij} is the expected value of composite cue ij : [$EV_{ij} = p_i m_j$].

Importantly, the animals received a small reward of 0.05 ml when they did not receive the

larger reward (m_j).”

**Reviewer #4 (Remarks to the Author):**

*My thanks to the authors for their response. While many of the changes and responses have*
*satisfied my concerns, there are still some remaining issues as well as new issues raised by*
*the additional material in the revision. In particular, my original review pointed out some*
*unusual results in the decoding analyses (e.g. below-chance decoding of a variable not yet*
*known to the monkey), but the response to these issues was inadequate. Also, the inclusion*
*of licking results is good, but the actual results themselves raise more questions, and do not*
*provide any new information about the behavioral significance of the neural coding*
*properties documented in the study.*

*On the whole my opinion of this study remains the same. It is a technically solid effort (with*
*the possible exception of the decoding analyses) that makes a modest advance in our*
*knowledge of primate amygdala function. While the results are compatible with the idea that*
*the amygdala contributes to choices under uncertainty, they do not provide any insight into*
*the mechanisms for this contribution.*

Authors' response: We thank the reviewer for their comments, which helped us further
improve the manuscript by including additional analysis results and figure panels, as
described below.

With respect to mechanistic advance afforded by our data, our focus in this study was to
determine “the role of primate amygdala neurons in the dynamic coding and sequential
integration of distinct reward attributes to value and risk”. Of course, it will be important to
elucidate in future studies how the presently reported signals are processed neuronally in a
choice task. However, even in choice tasks, only some amygdala neurons typically directly
reflect behavioral choices, whereas others signal cue-specific value or other related
variables not directly expressed in behavior. Such neurons may still provide important
information, e.g., as input to decision computations, or subjective valuations. Thus, here we
wanted to establish an initial basis by studying single neurons in a highly controlled
Pavlovian task in which information was revealed sequentially so that neurons coding
particular variables could be identified. We now include a new statement in the Discussion to
clarify the scope of our study.

Discussion (line 572): “Our aim in this study was to provide evidence on how single amygdala
neurons responded to visual stimuli that signalled information about reward probability,
magnitude and their integration into expected value and risk. These variables encoded by
amygdala neurons recorded during the Pavlovian task were shown to be behaviorally relevant
for the monkeys (trial-abortion and lick rates in the recording task, choices in separate tests).
Our data therefore establish a basis for investigating in future studies the separate, important
question of how these variables encoded by amygdala neurons are incorporated into neural
decision computations.”

*1) The analysis of Pavlovian licking behavior in the task is welcome. However, the specific*
*result reported raises new questions. Both monkeys appeared to lick *less* as a function of*
*the anticipated reward magnitude, as indicated by the negative regression estimates for*
*Magnitude in Table 3. Estimates for probability were also negative, though non-significant for*
*both animals. The estimates were obtained using a multiple regression, so one interpretation*
*is that while holding all else equal (i.e., risk and probability) the monkey's overt anticipation*
*of a reward is smallest for higher reward sizes. This is counterintuitive, and contradictory to*
*several prior studies (e.g. Morrison & Salman 2009, Saez & Salzman 2015, McGinty &*
*Newsome 2016). At a minimum, I think it appropriate to comment on this unusual result, and*
*to show these data in the supplements.*

**Authors' response:** We thank the Reviewer for this careful consideration of the licking data.
 These comments prompted us to interrogate these data in more depth. We found that there
 were a substantial number of trials in which the animals did not lick in anticipation (in
 particular, animal B), which were initially included in the regression and affected the results.
 Therefore, in order to explain variance in licking behavior when this behavior occurred, we
 now focus the analysis on trials with non-zero lick rates. We show these lick rates as a
 function of our task variables for both animals in a new supplementary figure. We also
 replaced the previously added regression table with updated results. These results indicate
 that lick rates depended on a combination of our key variables, and how these variables
 affected licking differed between animals. We also show in a new supplementary regression
 table that when only the directly cued variables probability and magnitude are included as
 regressors, both animals showed higher lick rates for probability and magnitude, consistent
 with previous studies. We note that previous studies did not—to our knowledge—investigate
 licking in relation to risk and expected value; thus, our manuscript provides novel evidence
 on this issue. We have updated the description of these findings in the Results, and as
 requested include a new paragraph in the Discussion to discuss these findings in relation to
 prior studies.

 **Fig. S1. Licking data.** Lick rates as a function of reward probability, magnitude, expected
 value and risk. Blue/black: animal A/B.

 **Table 3.** Mixed-effects regression of licking behavior on task variables.

Variable	Estimate	Standard error	t-statistic	Degrees of Freedom	P-value
Animal A					
Intercept	-0.032	0.041	-0.779	12,242	0.435
Probability	-0.137	0.076	-1.798	12,242	0.072
Magnitude	-0.573	0.121	-4.704	12,242	2.5e-6
Expected value	0.791	0.153	-5.146	12,242	2.6e-7
Risk	3.038	0.513	5.923	12,242	3.2e-9
Animal B					
Intercept	-0.259	0.055	-4.689	6,471	2.7e-6
Probability	0.225	0.102	2.195	6,471	0.028
Magnitude	0.609	0.165	3.674	6,471	0.0002
Expected value	-0.123	0.208	-0.591	6,471	0.554
Risk	-2.207	0.713	-3.091	6,471	0.001

234

235 **Table S1.** Mixed-effects regression of licking behavior on task variables.

Variable	Estimate	Standard error	t-statistic	Degrees of Freedom	P-value
Animal A					
Intercept	-0.266	0.028	-9.379	12,245	7.7e-21
Probability	0.188	0.043	4.314	12,245	1.6e-5
Magnitude	0.350	0.131	11.129	12,245	1.2e-28
Animal B					
Intercept	-0.169	0.037	-4.474	6,473	7.7e-6
Probability	0.176	0.059	2.940	6,473	0.003
Magnitude	0.170	0.042	3.991	6,473	6.6e-5

Results (line 154): “Anticipatory lick rates also reflected the influence of a combination of
 these variables, including risk, in a manner that differed between animals (Table 3, Fig. S1).
 We confirmed in a supplementary regression that both animals showed higher lick rates for
 probability and magnitude (Table S1), consistent with previous studies⁶⁰⁻⁶².”

Discussion (line 620): “Lick rates depended on a combination of these variables, and
 reflected risk in a manner that differed between animals, although both animals showed
 higher lick rates for probability and magnitude, consistent with previous studies⁶⁰⁻⁶².”

*2) Related to the above, even if the licking results were more similar to prior studies, their
 being included in the study does not do much to show how the neural results in the study
 contribute to behavior. To do that would require additional analyses to explicitly link the
 behavioral and neural results, e.g., to link licking behavior to the encoding of risk, magnitude,
 etc. Because of this, I'm still of the opinion that the behavioral significance of the amygdala
 coding properties is not clear.*

Authors' response: We appreciate the Reviewer's comment that highlights the potential
 relationship between amygdala neurons and behavior. We have included the following
 statement to clarify the contribution of our present study and to acknowledge the importance
 of building on our results to investigate the relationship between these amygdala signals and
 behavior in future studies.

Discussion (line 573): “Our aim in this study was to provide evidence on how single
 amygdala neurons responded to visual stimuli that signalled information about reward
 probability, magnitude and their integration into expected value and risk. These variables
 encoded by amygdala neurons recorded during the Pavlovian task were shown to be
 behaviorally relevant for the monkeys (trial-abortion and lick rates in the recording task,
 choices in separate tests). Our data therefore establish a basis for investigating in future
 studies the separate, important question of how these variables encoded by amygdala
 neurons are incorporated into neural decision computations.”

*3) My thanks for reporting the correlation between the behavioral variables in the new Figure
 S3 and new table S1. The authors are clearly aware that the high correlations between the
 variables (e.g., $r=0.943$ for risk and magnitude) could make it difficult to tease apart the
 unique contribution of each variable to explaining variance in neural activity. The use of
 partial R^2 (e.g. in Figure 4e) is very appropriate here, and is one way to make the case that
 risk is a strongly encoded variable over and above magnitude and probability. Nonetheless,
 when two dependent variables are highly correlated, small, random fluctuations in the data
 can influence the relative portion of variance attributable to each variable. To quantify this*

*effect in this study, I would like to recommend some sort of repeat reliability analysis. For*
*example, one could split the data into two counterbalanced sets of trials, use GLM4 to*
*identify cells that significantly encode each of the relevant variables in each half of the data,*
*and then construct a confusion matrix showing the consistency of these classifications*
*between the two halves.*

Authors' response: We appreciate the Reviewer's careful consideration of this issue and
thank them for this helpful suggestion. We have implemented new analyses to quantify the
robustness of the neuronal classification as recommended. We performed bootstrap
analyses based on sub-sampling each neuron's trial responses 1000 times (which avoids
reducing statistical power by split-half approaches). We report two new supplementary
results: the distribution of the number of neurons classified as coding each variable when
this bootstrap approach is used, and the distribution of consistent classifications between our
main results (GLM4) and the bootstrap results. These new analyses suggest that
identification of our main variables probability, magnitude and risk was robust in
bootstrapped analyses, whereas identification of neurons coding expected value was less
robust. We have added the following text and figures to the manuscript.

Methods (line 1106): "We performed for each of the 483 neurons a bootstrap analysis with
resampling over 1000 iterations and refitting GLM4 for each iteration. We report in Fig. S6
the proportion of significant neurons for a given variable across bootstrap iterations. These
histograms show that the bootstrap largely confirms the original proportions of neurons. We
also examined whether neurons identified as coding our key correlated variables of interest
in the original analysis, namely magnitude and risk, were consistently classified in the
bootstrap analysis. To do so, we identified neurons that were originally classified as coding
magnitude or risk and then determined the proportion of bootstrap iterations in which this
classification was either confirmed or not. For each neuron, we considered a classification as
consistent if the probability of confirmation (across bootstrap iterations) was $p > 0.5$, and a
value of 0 if this probability was $p < 0.5$. (We also report numbers for a stricter confirmation
threshold of $p > 0.85$.) We then calculate across neurons the proportion of original
magnitude and risk neurons that were confirmed or unconfirmed."

Results (line 340): "To evaluate the robustness of our classification of neurons coding the two
correlated variables magnitude and risk, we performed a bootstrap analysis (Fig. S6). Of 99
risk-coding neurons (eq. 4), 95 showed robust risk-coding in the bootstrap (significant in more
than 500/1,000 bootstrap iterations; 43 neurons were significant in more than 850/1000
iterations); further, 38 of these neurons were significant for risk but insignificant for magnitude
(significant in less than 500/1,000 iterations). Similarly, of 94 magnitude-coding neurons (eq.
4), 81 showed robust magnitude-coding in the bootstrap (significant in more than 500/1,000
bootstrap iterations; 48 neurons significant in more than 850/1000 iterations); 25 of these
neurons were significant for magnitude but insignificant for risk. Thus, although risk and
magnitude were correlated in our dataset, substantial numbers of neurons coded these
variables robustly and distinctly."

**“Fig. S6. Robustness of neuronal classification based on bootstrap.** (a) Distribution of
 neurons with significant regressor in sliding-window regression (eq. 4, GLM4) across 1,000
 iterations from a bootstrap analysis (1,000 iterations per neuron, with resampling). Probability,
 magnitude and risk were more frequently confirmed in bootstrap (number of significant
 bootstrap iterations) than expected value, which was encoded by the lowest number of
 neurons in the original analysis. Further, a minority of neurons encoding the correlated
 variables magnitude and risk showed less reliable coding, indicated by relatively lower number
 of significant iterations. Thus, in these neurons, the distinction between magnitude and risk
 coding was likely less clear; see results in (b). (b) Confusion matrices for key variables. Each
 matrix considers neurons that coded a particular variable in our main model (eq. 4) and
 specifies how many of these neurons were identified in the bootstrap as coding this variable
 but not a control variable (row/column: ‘sig’/‘nsig’; sig: significant, nsig: non-significant),
 neurons that in the bootstrap coded the control variable only but not the originally coded
 variable (‘nsig’/‘sig’), and neurons that in the bootstrap coded both (‘sig’/‘sig’) or neither
 variables (‘nsig’/‘nsig’). Significance defined as a significant regressor in more than 500
 iterations of the bootstrap. For example, of 99 risk-coding neurons in the original analysis
 (upper left matrix in (b)), 38 coded risk but not magnitude in the bootstrap, 57 coded both risk
 and magnitude, 0 coded magnitude but not risk, and 4 coded neither variable.”

*4) In the first review I had three comments on the decoding results: Below-chance encoding*
 *in Figure 6b, the low cross-condition probability decoding in Figure 6F (despite the very*
 *consistent cross-condition coding of probability in Figure 2C), and the asymmetric cross-*
 *condition decoding of probability and magnitude. On the whole, I found the response to*
 *these issues inadequate. The responses seemed to acknowledge the somewhat unusual*
 *results, but only offered vague explanations for them (e.g. outlier cells, or the “complexity”*
 *of the population code). At a bare minimum, I think it would be wise to confirm that these*
 *results are robust to outliers and to the assumptions of the nearest-neighbor classifier that*
 *was used. SVM, which the authors say has already been performed, would probably do the*
 *trick.*

*Beyond confirming the results, a more satisfying response would be to explain the results in*
 *terms of concrete features of the underlying data – perhaps corresponding to important*
 *features of information coding in the populations. For example, the response regarding Fig.*

6b mentioned that outlier data might be responsible for the below-chance decoding, but
 demurred at the idea of removing these outliers and replacing the figure. This seems like an
 easy fix, and it's unclear why it wasn't done.

In my opinion, one of the strengths of this paper is that the analyses appear well reasoned
 and technically sound. Not fully investigating these unusual decoding results would
 undermine the technical merits of the paper.

Authors' response: We thank the Reviewer for their help in guiding us toward further
 robustness checks. We have now replaced Fig. 6b with a version computed without the
 outlier neurons (which we note in the Methods and legend). We also comment on the
 relatively low cross-cue decoding accuracy. We have also included a new supplementary
 figure that shows that the patterns of decoding results in the main results in Fig. 6 are robust
 when assessed with the SVM decoding, as requested.

Results (line 449):

"This reduced accuracy was largely due to low cross-cue decoding between the mid and
 high probability levels. Although these probabilities were distinct behaviorally (Fig. 1),
 discriminability across cue sets was low (52%, compared to 60% and 63% for discrimination
 between low and medium and low and high probabilities, respectively, for nearest-neighbor
 decoder; for support-vector machine: 52% compared to 71% and 62%) perhaps related to
 properties of the images used to cue these probabilities."

**Fig. S8. Population decoding using support-vector machine algorithm.** (a) Accuracy of a
 support-vector machine decoder (mean +/- s.e.m.) classifying high-vs.-low levels of
 probability, magnitude, expected value (EV), and risk from unselected amygdala neurons (N
 = 483) in specific trial epochs. Gray lines: decoding based on data using trial-shuffled group
 labels. (b) Decoding accuracy in control task with reversed cue order. (c) Decoding accuracy
 for probability based on fractal or sector cue trials. Gray lines: decoding based on data using
 trial-shuffled group labels. (d) Decoding accuracy (mean +/- s.e.m.) for probability when
 training and testing within cue set (fractal or sector, black bar) and training and testing across
 cue sets (averaged over fractal-to-sector and sector-to-fractal decoding, white bar). Gray lines:
 decoding based on data using trial-shuffled group labels. (e) Decoding and cross-decoding of
 probability and magnitude information. Decoders were trained to discriminate low-vs.-high
 probability (first cue period, indicated by fractal symbol, though the analysis included both
 fractal and sector trials) or magnitude (second cue period, indicated by bar symbol) and tested
 within or across periods and reward variables. Gray lines: decoding based on data using
 trial-shuffled group labels. (f) Decoding and cross-decoding performed separately for LA (N = 113)

and BL (N = 172) neurons. Gray lines: decoding based on data using trial-shuffled group
labels. *: $P < 0.001$; n.s.: non-significant.

**REVIEWER COMMENTS**

**Reviewer #3 (Remarks to the Author):**

*The authours have taken all my concerns into account and adressed them appropriately.*
*The paper will make a great contribution to the field.*

**Authors' response:** We thank the Reviewer for their helpful comments and for supporting
the publication of our manuscript.

**Reviewer #4 (Remarks to the Author):**

*My thanks to the authors for their updates to the manuscript. I have only a few suggested*
*corrections, as well as a general comment.*

*Licking analysis: Thank you for providing these additional details, in particular the graphs,*
*which help clarify the potentially confusing regression estimates shown in the table (e.g. the*
*negative estimates for magnitude and probability).*

*Bootstrapping analysis: This new analysis does not fully address my original concern. The*
*original concern was that the variables are highly correlated, making it difficult to consistently*
*designate a neuron as encoding one variable but not the other. This question is made more*
*complicated by the possibility of finding cells that encode two correlated variables (e.g.*
*magnitude and risk), which is evident in the new analysis but was not addressed in the*
*original study.*

*In contrast, this new analysis mostly addresses the question of sensitivity for detecting*
*significant effects for a single variable, rather than the potential for confusion between them.*
*For example, it's encouraging that for "risk" neurons there are no cells in the upper right*
*quadrant of the table; but the table entries are thresholded by requiring >500 bootstrapped*
*samples to be significant. A more accurate measure would be to compute the fraction of*
*samples in which significance is detected. Also, Fig. S6 does not separately consider cells*
*that encode Magnitude alone, Risk alone, or both at once, making it difficult to assess the*
*degree of confusion between these two variables.*

***I'd like to suggest a variant of this new analysis that addresses my concerns:*** For each
*cell, determine whether it encodes variable A, variable B, or both, using the original sample*
*of trials. Then, for each cell do the same analysis over 1000 bootstrapped samples, and*
*quantify the fraction of samples for which the cell encodes A, B, and both. Then, among the*
*"original A" cells (cells encoding A in the original analysis), find the mean fraction of*
*bootstrap samples that encoded A, B and both. The expectation is that among "original A"*
*cells, the largest fraction of bootstrap samples should encode A, with fewer encoding B or*
*both. Likewise, among "original B" cells, the largest fraction of samples should encode B,*
*etc. The key outcome here will be how likely it is for an "original A" cell to be mis-classified*
*as a B-cell in the resamples, and for "original B" to be classified as A in the resamples.*

*Decoding analysis: Thank you for providing the SVM results as a supplement. I could not*
*find the outlier removal procedure in the methods. Please ensure that this is included*
*in the final version.*

*General comment: I'd like to offer a more in-depth explanation of my assessment of the*
*mechanistic advance. The main conclusion of the study is that the amygdala encodes the*
*variables that appeared to drive behavior (probability, magnitude, EV, and risk). This makes*
*it possible that the amygdala is involved in the Pavlovian behaviors in this task, and may be*
*involved in other behaviors that depend on these variables. This "existence proof" is an*
*important first step in establishing the mechanisms by which amygdala might contribute to*
*decisions under risk and other behaviors.*

*However, in my opinion results like this one must be treated with caution until there is*
*additional evidence that links neural coding to behavior. Examples are the study of Stoll &*
*Rudebeck (2024), which shows that within-session changes in juice preference are linked to*
*within-session changes in neural coding; and the study of Ballesta and Padoa-Schioppa,*
*which uses causal methods to show the involvement of OFC in choice.*

*Without additional evidence, it is quite possible that the amygdala is not involved in the*
*behavior shown here. Many other brain regions encode probability, magnitude, etc., and it's*
*feasible that the signals in these other regions are more important than amygdala for driving*
*behavior. One might argue that the information encoded in amygdala must have some non-*
*trivial influence on downstream circuits and must ultimately affect behavior in some way. But*
*a counterargument which I have recently embraced is the idea that a surprising fraction of*
*neural information coding is in the "null space", defined as activity patterns that do not exert*
*a net influence on downstream circuitry and behavior (for a review of see Churchland &*
*Shenoy, Nat Rev Neuro, 2024). In other words, in any given task it is possible that the*
*information encoded in a given brain region is not directly contributing to the observed*
*behavior.*

*I grant that this position may make me an outlier, but for these reasons I am circumspect*
*when drawing conclusions about behavioral mechanisms based on results showing only that*
*information is being encoded in a neural population, as is the case here.*

**Authors' response:** We thank the Reviewer for their comments and for supporting the
publication of our manuscript. We fulfilled the specific requests as follows:

We now describe the procedure for outlier removal in the Methods: "In Fig. 6b, we excluded
an outlier neuron as its normalized impulse rate exceeded the mean of the other neurons by
more than 4 standard deviations."

We performed the requested variant of the bootstrapping analysis. The results provide
further support for our original finding that risk-coding and magnitude-coding neurons are
largely distinct. We included the results as new panel in Fig. S6 and a summary in the main
Results

"Further, among neurons classified as risk-coding, the largest fraction of bootstrap samples
was also classified as risk-coding rather than magnitude-coding, and vice versa for
magnitude-coding neurons (**Fig. S6**)."

Fig. S6. Robustness of neuronal classification based on bootstrap. ... (c) Proportion of bootstrap samples classified as magnitude-coding, risk-coding or joint magnitude-and-risk-coding shown separately for neurons that were in the main analysis classified as magnitude-coding (left), risk-coding (middle), or joint magnitude-and-risk-coding (right).